# On the consistent estimation of optimal Receiver Operating Characteristic (ROC) curve

**Renxiong Liu**
Department of Statistics
Ohio State University
Columbus, OH 43210
`liu.6732@buckeyemail.osu.edu`

**Yunzhang Zhu**
Department of Statistics
Ohio State University
Columbus, OH 43210
`zhu.219@osu.edu`

## Abstract

Under a standard binary classification setting with possible model misspecification, we study the problem of estimating general Receiver Operating Characteristic (ROC) curve, which is an arbitrary set of false positive rate (FPR) and true positive rate (TPR) pairs. We formally introduce the notion of *optimal ROC curve* over a general model space. It is argued that any ROC curve estimation methods implemented over the given model space should target the optimal ROC curve over that space. Three popular ROC curve estimation methods are then analyzed at the population level (i.e., when there are infinite number of samples) under both correct and incorrect model specification. Based on our analysis, they are all consistent when the surrogate loss function satisfies certain conditions and the given model space includes all measurable classifiers. Interestingly, some of these conditions are similar to those that are required to ensure classification consistency. When the model space is incorrectly specified, however, we show that only one method leads to consistent estimation of the ROC curve over the chosen model space. We present some numerical results to demonstrate the effects of model misspecification on the performance of various methods in terms of their ROC curve estimates.

## 1 Introduction

One common goal in standard binary classification task is to find a classifier with minimum misclassification error, which is the weighted average of the *false negative rate* (FNR) and *false positive rate* (FPR). This implicitly assumes that the two types of errors are equally important. In many real applications, however, these two types of errors may need to be treated unequally by assigning asymmetric losses to them. For example, in spam email detection, classifying non-spam as spam may incur a higher cost than classifying spam into non-spam. Existing methods to handle this problem can be roughly categorized into three groups. Cost-sensitive learning [Elkan, 2001], which generalizes the empirical risk minimization framework [Vapnik, 1991], assigns unequal class weights to the two types of errors and seeks to find a classifier that minimizes the weighted error. Neyman-Pearson classification [Cannon et al., 2002; Scott and Nowak, 2005; Rigollet and Tong, 2011] is another type of method, which aims to minimize one type of error (FNR) subject to a upper bound constraint on the other type of error (FPR). We call these two types of methods *weighted method* and *constrained method*, respectively. Another popular approach [Fawcett, 2006], which we call *cutoff methods*, directly constructs a set of classifiers based on a discriminant function from some classifier. Specifically, the set of classifiers is obtained through changing the intercept of the given discriminant function. Empirically, the cutoff method is computationally efficient and can work well in some situations.

To study the performance of these methods at the population level, i.e., there are infinite number of samples, we first introduce a general notion of *optimal ROC curve* over an arbitrary model space. This

36th Conference on Neural Information Processing Systems (NeurIPS 2022).

new notion of optimal ROC curve parallels the notion of Bayes error for standard binary classification when the model space is the set of all measurable classifiers. Traditionally, ROC curve is typically defined as the set of FPR-TPR pairs produced by the classifiers obtained from the cutoff method, where TPR denotes *true positive rate* and is defined by TPR $= 1 -$ FNR. An optimal ROC curve [see, e.g., Van Trees, 2004] is defined as the ROC curve generated by $\eta(x) = \mathbb{P}(Y = 1 \mid X = x)$, where $Y$ denotes the binary response and $X$ denotes the feature vector. This curve is optimal in the sense that no other ROC curves can lie above it. Here we generalize the notion of optimal ROC curve given a specific model space. This is done by assigning a partial order to all possible FPR-TPR pairs that can be obtained over the given model space, and defining the optimal ROC curve as the *pareto frontier* of the set of FPR-TPR pairs. This allows us to define the optimal ROC curve under an arbitrary model space, and discuss the general case that model is possibly mis-specified. We show that the newly defined optimal ROC curve coincides with the existing definition when the model space includes all possible measurable classifiers.

In terms of consistency, we find that when the model space contains all measurable classifiers, all three methods target the optimal ROC curve under $0/1$ loss or some surrogate loss under a new notion of correct model specification. Interestingly, in order for a model-loss pair to be correctly specified, a sufficient condition required for the surrogate loss is related to the notion of classification-calibration [Bartlett et al., 2006], which is a sufficient condition required to achieve consistency in binary classification. In this sense, our results generalize that of Bartlett et al. [2006] for binary classification to optimal ROC curve estimation. When the model-loss pair is mis-specified, however, we show that only the constrained method targets the optimal ROC curve over the chosen model space. In this sense, the other two methods may miss some optimal FPR-TPR pairs on the optimal ROC curve. Importantly, this suggests that the constrained method may be preferable to the two other methods in many practical applications where the model space considered does not include the true model.

To the best of our knowledge, our work is the first that formally investigate theoretical conditions under which the commonly used methods could recover the optimal ROC curve in a statistical decision theoretical framework. Moreover, our proposed definition of optimal ROC curve also generalizes the conventional definition, relaxing some unnecessary distributional assumptions on the data. For ROC curve estimation, existing works are either empirical or mostly focus on how to generate a single FPR-TPR pair on the optimal ROC curve using either the weighted method or the constrained method. For example, Cannon et al. [2002] investigated the empirical version of the constrained method, and provided a probabilistic guarantee on simultaneous control of both FPR and FNR. Extensions to structural risk minimization were later considered in Scott and Nowak [2005]. However, they did not address the issues on how to better estimate the whole ROC curve. Scott [2007] proposed two families of performance measures to evaluate classifiers produced by the constrained method. Interestingly, they also showed that the ROC curve is concave under some assumptions on the data generating distributions, part of which are special cases of our theory (see Theorem 2). More recently, Rigollet and Tong [2011] focused on how to obtain a classifier that can control its FPR at the population level. This is done by using a convex surrogate loss and considering a slightly more stringent upper bound than the desired level for the FPR. However, their method may not target the best trade-off between FPR and TPR, which is partly attributed to the use of a convex surrogate loss and possible model mis-specification. Lastly, Bach et al. [2006] proposed to estimate the ROC curve by computing a convex envelope of ROC curves generated by the cutoff method when varying the class weights. They demonstrate that this will usually outperform the ROC curve produced by the cutoff method alone, which is partly consistent with our theory that the weighted and cutoff method may be sub-optimal when the model is mis-specified.

Moreover, our definition of the optimal ROC curve is also new and it extends the old definition considered by Van Trees [2004] and Scott [2007] in two different ways. First, the optimal ROC curve in Van Trees [2004] and Scott [2007] requires some continuity assumptions on the data generating distribution. By contrast, our definition is quite general and requires a much weaker assumption on the data generating distribution. In fact, we show that our definition coincides with the existing definition if the continuity assumption is satisfied. Second, our definition allows model mis-specification, which reveals some interesting differences among the commonly used methods when model mis-specification may be present. Another important contribution is that we formally prove the continuity and concavity of the optimal ROC curve and present some other important properties of the optimal ROC curve, which extends results informally stated in Van Trees [2004] and Scott [2007].

The rest of the article is structured as follows. We define the notion of optimal ROC curve over a general model space, and derive the optimal ROC curve over the entire population analytically in Section 2. Section 3 analyzes the population analogue of three ROC estimation methods when model is correctly specified and mis-specified. We present some numerical examples to illustrate the impact the mis-specification on the performance of three methods in Section 4, and we end with some discussions in Section 5. All proofs are included in the supplementary material.

## 2   Optimal ROC curve over a general model space

In this section, we introduce the notion of optimal ROC curve over a general model space. The notion of optimal ROC curve is parallel to Bayes risk in standard binary classification. We therefore argue that the optimal ROC curve should be the theoretical target for ROC curve estimation, just like Bayes error in standard binary classification.

Before proceeding, we first lay out the basic problem setup, some notations, and assumptions to be used throughout the article. Consider a feature-response pair $(X, Y)$ with joint distribution $\mathbb{P}$, where $X \in \mathbb{R}^p$ is the feature vector and $Y \in \{-1, 1\}$ is the corresponding binary response. Throughout, we assume that the marginal distribution $X$, denoted as $\mathbb{P}_X$, is continuous, and the marginal distribution of $Y$ satisfies $p^+ = \mathbb{P}(Y = 1) \in (0, 1)$ and $p^- = \mathbb{P}(Y = -1) \in (0, 1)$. We use $h : \mathbb{R}^p \to \{-1, 1\}$ to denote a classifier that maps feature vectors to their responses. For many methods, a classifier is induced by the sign of a so-called *discriminant function*. Throughout, we use $\mathcal{H}$ to denote set of classifiers, and $\mathcal{F}$ to denote set of discriminant functions.

Many different metrics can be used to assess the performance of a classifier, of which misclassification error rate $\mathbb{P}(Y \neq h(X))$ is perhaps the most commonly used and well-studied in the literature [Zhang et al., 2004; Bartlett et al., 2006]. Classification methods are often designed to minimize the misclassification error, which is the weighted average of two types of error rate, namely, *false positive rate* $\text{FPR}(h) = \mathbb{P}(h(X) = 1 \mid Y = -1)$ and *false negative rate* $\text{FNR}(h) = \mathbb{P}(h(X) = -1 \mid Y = 1)$. In this article, we focus on the problem of minimizing these two types of error rate simultaneously. This is essentially equivalent to the estimation of optimal ROC curve, which plots the *true positive rate* (TPR) against the false positive rate for a set of classifiers, where the true positive rate is defined to be $\text{TPR}(h) = 1 - \text{FNR}(h)$.

In standard binary classification, the optimal classifier that minimizes the misclassification error is often referred to as the Bayes classifier. For our problem, we have two criterion to optimize. By borrowing ideas from multi-objective optimization literature [Miettinen, 2012], we define the set of optimal classifiers as those that can not be dominated by any other classifiers, and the optimal ROC curve as the set of FPR-TPR pairs that can not be dominated by any other FPR-TPR pairs. Here classifier $h$ dominates classifier $h'$, denoted as $h \succ h'$, means that $\text{TPR}(h) \geq \text{TPR}(h')$, $\text{FPR}(h) \leq \text{FPR}(h')$ and $(\text{FPR}(h), \text{TPR}(h)) \neq (\text{FPR}(h'), \text{TPR}(h'))$. In this sense, a classifier $h$ is an optimal classifier if it can not be dominated by any other classifiers. Similarly, for any two FPR-TPR pairs $(u, v)$ and $(u', v')$, we say that $(u, v)$ dominates $(u', v')$, denoted as $(u, v) \succ (u', v')$, if $u \leq u'$, $v \geq v'$ and $(u, v) \neq (u', v')$. Given this partial ordering, we denote by $PF(A) := \{(u, v) \in A \mid (u', v') \not\succ (u, v) \text{ for any } (u', v') \in A\}$ the pareto frontier of a compact set $A \subseteq \mathbb{R}^2$. Therefore, the set of optimal classifiers constitutes the pareto frontier of the given set of classifiers, while the optimal ROC curve is the pareto frontier of the given set of FPR-TPR pairs.

Formally, we define the optimal ROC curve over an arbitrary set of classifiers $\mathcal{H}$, which allows possible *model mis-specification*.

**Definition 1** *(Optimal ROC curve over model space $\mathcal{H}$) Given a set of classifiers $\mathcal{H}$, denote by $S(\mathcal{H}) = \{(FPR(h), TPR(h)) \mid h \in \mathcal{H}\}$ the set of all FPR-TPR pairs generated by classifiers in $\mathcal{H}$. We call the pareto frontier of $S(\mathcal{H})$, denoted as $\gamma(\mathcal{H}) = PF(S(\mathcal{H}))$, the optimal receiver operating characteristic (ROC) curve over $\mathcal{H}$. When $\mathcal{H} = \mathcal{H}_a$, where $\mathcal{H}_a$ is the class of all measurable classifiers, we simply call $\gamma(\mathcal{H}_a)$ the optimal ROC curve.*

Throughout this article, we assume that $S(\mathcal{H})$ is a closed set so that $PF(S(\mathcal{H}))$ is well-defined. As we can see from Definition 1, classifiers corresponding to any point on the optimal ROC curve cannot be dominated by any other classifier. Note that this is also similar to the notion of admissibility for general decision rules in statistical decision theory. The above definition of optimal ROC curve differs from the existing definition by Scott [2007] in that it allows for model mis-specification. When the model is correctly specified, the optimal ROC curve parallels the notion of Bayes error

rate for regular binary classification, while the corresponding classifiers parallels the notion of Bayes classifier. It is well-known that Bayes error rate is $1/2 - \mathbb{E}|\eta(X) - 1/2|$ [Devroye et al., 2013], where $\eta(x) = \mathbb{P}(Y = 1 \mid X = x)$ is the regression function. In parallel, we derive the optimal ROC curve over the class of all measurable classifiers $\mathcal{H}_a$. Compared with some existing definitions of optimal ROC curve under correct model specification, our definition is more general, requiring less conditions on the distribution of $(X, Y)$ [see, e.g., Scott, 2007]. Specifically, our definition of optimal ROC curve only requires that $X$ is a continuous random variable, while Scott [2007] requires that both $X$ and $\eta(X)$ are continuous random variables.

The following theorem derives an analytic form for the optimal ROC curve $\gamma(\mathcal{H}_a)$ defined in Definition 1.

**Theorem 1** *Let* $\eta(x) = \mathbb{P}(Y = 1 \mid X = x)$, $p^\star = \inf\{p : \mathbb{P}(\eta(X) \le p) = 1\}$, *and let* $\mathbb{I}(\cdot)$ *denote the indicator function. We have that the optimal ROC curve can be represented as*

$$\gamma(\mathcal{H}_a) = \left\{ (\alpha, s(\alpha)) \mid \alpha \in \left[0, 1 - \mathbb{P}(\eta(X) = 0)/p^-\right] \right\},\tag{1}$$

*where*

$$s(\alpha) = \begin{cases} \mathbb{P}(\eta(X) = 1)/p^+ & \text{if } \alpha = 0, \\ \frac{\mathbb{P}(\eta(X) > c(\alpha)) - \alpha p^- - c(\alpha)\mathbb{E}\eta(X)\mathbb{I}(\eta(X) > c(\alpha))}{p^+(1 - c(\alpha))} & \text{if } \alpha \in \left(0, 1 - \frac{\mathbb{P}(\eta(X)=0)}{p^-}\right] \end{cases}\tag{2}$$

*with* $c(\alpha) = \inf\{c \in [0, p^\star] : \mathbb{E}(1 - \eta(X))(\mathbb{I}(\eta(X) > c) - \alpha) \le 0\}$. *Moreover,* $s(\alpha)$ *is continuous, strictly increasing, and concave over its domain* $[0, 1 - \mathbb{P}(\eta(X) = 0)/p^-]$.

Some remarks are in order. First, we can see that the optimal ROC curve is a continuous, strictly increasing, and concave curve defined by the function $s(\alpha)$ over the interval $[0, 1 - \mathbb{P}(\eta(X) = 0)/p^-]$. Second, existing definition of the optimal ROC curve typically relies on the likelihood ratio test [see, e.g., McIntosh and Pepe, 2002; Scott, 2007]. More specifically, it assumes that $X$ has class-conditional density function $f_+(x)$ and $f_-(x)$ conditional on $Y = 1$ and $Y = -1$, and considers a class of classifiers indexed by $\lambda \in [0, \infty]$: $\{h_\lambda(x) = \text{Sign}(f_+(x)/f_-(x) - \lambda), \lambda \in [0, \infty]\}$. Then, under the assumption that

$$\mathbb{P}(f_+(X)/f_-(X) = \lambda \mid Y = -1) = \mathbb{P}(f_+(X)/f_-(X) = \lambda \mid Y = 1) = 0\tag{3}$$

for any $\lambda \in [0, \infty]$, Scott [2007] defined the optimal ROC curve as the set of FPR-TPR pairs generated by the class of classifiers $\{h_\lambda(x); \lambda \in [0, \infty]\}$. In other words, the existing definition of the optimal ROC curve can be expressed as

$$\{(\text{FPR}(h_\lambda), \text{TPR}(h_\lambda)) \mid \lambda \in [0, \infty]\}.\tag{4}$$

Our definition of optimal ROC curve (c.f. Definition 1), however, does not require the likelihood ratio assumption in (3), and hence in this sense is more general than the existing definition. Indeed, assumption(3) may be violated in some common situations, for example, the two class-conditional densities $f_+(x)$ and $f_-(x)$ have different supports. Moreover, under the condition (3), we can show that our proposed definition coincides with the existing definition (4). Furthermore, the optimal ROC curve is the FPR-TPR pairs generated by the set of classifiers $\{\text{Sign}(\eta(X) - c) \mid c \in [0, 1]\}$. We summarize these results in Corollary S1 of the supplementary material.

Lastly, we remark that our proposed notion of the optimal ROC curve (c.f. Definition 1) also allows model misspecification. This allows us to study the statistical consistency of special classes of classification method such as linear classifiers. We will demonstrate in later sections that model specification has a meaningful impact on the consistency of various popular classification methods for ROC curve estimation.

## 3 ROC curve estimation methods

In this section, we review three ROC curve estimation methods that are widely used in practice. We then investigate, from a statistical decision theoretical point of view, whether these methods are targeting the optimal ROC curve defined in the previous section. In particular, we shall analyze their population analogue when the model is mis-specified, that is, the optimal classifier is not included in the set of classifiers considered in the model.

### 3.1 Neyman-Pearson classification

Neyman-Pearson classification [Scott and Nowak, 2005; Rigollet and Tong, 2011], or the constrained method, minimizes one type of error subject to an upper bound constraint on the other type of error. The population analogue of Neyman-Pearson classification is equivalent to finding a classifier that maximizes the true positive rate subject to false positive rate constraint, that is,

$$\underset{h \in \mathcal{H}}{\text{maximize }} \text{TPR}(h) \text{ subject to } \text{FPR}(h) \leq \alpha \,, \tag{5}$$

where $\mathcal{H}$ is a set of classifiers. In this subsection, we will show that under mild conditions the ROC curve generated from the constrained method by varying $\alpha \in [0, 1]$ can recover the optimal ROC curve over general model space $\mathcal{H}$.

Before proceeding, we call the ROC curve generated by the constrained method, defined by

$$\gamma_C(\mathcal{H}) = \{(\text{FPR}(h^\star_{\alpha,\mathcal{H}}), \text{TPR}(h^\star_{\alpha,\mathcal{H}})) \mid \alpha \in [0, 1]\} \,,$$

the population ROC curve of the constrained method, where $h^\star_{\alpha,\mathcal{H}}$ is a solution to (5). Note that the population ROC curve is essentially the set of FPR-TPR pairs generated by the solutions to the constrained method (5) after varying $\alpha \in [0, 1]$.

We first consider the case $\mathcal{H} = \mathcal{H}_a$, where the model space $\mathcal{H}_a$ is the set of all measurable classifiers. In this case, we can derive the analytic solution to the constrained method (5), which then allows us to derive the population ROC curve $\gamma_C(\mathcal{H}_a)$ generated by the constrained method.

**Theorem 2** *Denote $\eta(x) = \mathbb{P}(Y = 1 \mid X = x)$ and $p^\star = \inf\{p : \mathbb{P}(\eta(X) \leq p) = 1\}$. Let $\mathcal{H} = \mathcal{H}_a$ be the set of all measurable classifiers. For any solution $h^\star_\alpha(x)$ of (5), denote by $R^\star_\alpha = \{x : h^\star_\alpha(x) = 1\}$. Then, up to a $\mathbb{P}_X$-null set,*

$$R^\star_\alpha = \begin{cases} \{x : \eta(x) = 1\} & \text{if } \alpha = 0 \,, \\ \{x : \eta(x) > c(\alpha)\} \cup \mathcal{N}_\alpha & \text{if } \alpha \in (0, 1] \,, \end{cases} \tag{6}$$

*where $c(\alpha) = \inf\{c \in [0, p^\star] : \mathbb{E}(1 - \eta(X))(\mathbb{I}(\eta(X) > c) - \alpha) \leq 0\}$ with $c(\alpha) \in [0, 1)$ and $\mathcal{N}_\alpha \subseteq \{x : \eta(x) = c(\alpha)\}$ with*

$$\begin{aligned} \mathbb{P}(X \in \mathcal{N}_\alpha) &= \frac{\mathbb{E}(1-\eta(X))(\alpha - \mathbb{I}(\eta(X) > c(\alpha)))}{1 - c(\alpha)} & \text{if } \alpha \in \left(0, 1 - \frac{\mathbb{P}(\eta(X)=0)}{p^-}\right] \,, \\ \mathbb{P}(X \in \mathcal{N}_\alpha) &\leq \mathbb{E}(1 - \eta(X))(\alpha - \mathbb{I}(\eta(X) > 0)) & \text{if } \alpha \in \left(1 - \frac{\mathbb{P}(\eta(X)=0)}{p^-}, 1\right] \,. \end{aligned} \tag{7}$$

*Moreover, we have that optimal ROC curve $\gamma(\mathcal{H}_a)$ satisfies*

$$\gamma(\mathcal{H}_a) \subseteq \gamma_C(\mathcal{H}_a) \subseteq \gamma(\mathcal{H}_a) \cup \left\{(\alpha, 1) : \alpha \in \left(1 - \mathbb{P}(\eta(X) = 0)/p^-, 1\right]\right\} \,. \tag{8}$$

This theorem provides a complete characterization of the population ROC curve of the constrained method. Combining (8) with Theorem 1, we can see that the constrained method can recover the optimal ROC curve in the sense that it produces a curve that contains the optimal ROC curve. Using this, we can further show that the pareto frontier of the population ROC curve, defined by $\gamma^\star_C(\mathcal{H}_a) = \text{PF}(cl(\gamma_C(\mathcal{H}_a)))$, is identical to the optimal ROC curve. Here we use $cl(A)$ to denote the closure of set $A$. We summarize this result as below.

**Corollary 1** *We have that*

$$\gamma^\star_C(\mathcal{H}_a) = \gamma(\mathcal{H}_a) \subseteq \gamma_C(\mathcal{H}_a) \,. \tag{9}$$

Corollary 1 suggests that the constrained method over $\mathcal{H}_a$ indeed targets the optimal ROC curve $\gamma(\mathcal{H}_a)$. Essentially, by our proof of Corollary 1, the discrepancy between $\gamma_C(\mathcal{H}_a)$ and the optimal ROC curve $\gamma(\mathcal{H}_a)$ occurs only if TPR = 1. Moreover, by our proof if $\mathbb{P}(\eta(X) = 0) = 0$ then we have that $\gamma_C(\mathcal{H}_a) = \gamma(\mathcal{H}_a)$. Notably, Scott [2007] also derived the ROC curve generated by the constrained method $\gamma_C(\mathcal{H}_a)$ and proved that it coincides with the optimal ROC curve they defined under likelihood ratio condition (3). Here, we show that the ROC curve generated by the constrained method coincides with our defined optimal ROC curve (c.f. definition 1) by only requiring $\mathbb{P}(\eta(X) = 0) = 0$.

Note that for a general model space $\mathcal{H}$, it is no longer possible to derive the population ROC curve analytically. However, relations like (9) continue to hold. That is, the pareto frontier of the population ROC curve for the constrained method, defined by $\gamma^\star_C(\mathcal{H}) = \text{PF}(cl(\gamma_C(\mathcal{H})))$, coincides with the optimal ROC curve over $\mathcal{H}$.

**Proposition 1** *Suppose that $S(\mathcal{H}) = \{(FPR(h), TPR(h)) \mid h \in \mathcal{H}\}$ is a closed set. We have that $\gamma_C^\star(\mathcal{H}) = \gamma(\mathcal{H}) \subseteq \gamma_C(\mathcal{H})$.*

Proposition 1 shows that any point on the optimal ROC curve over a general model space $\mathcal{H}$ can be recovered by solutions to the constrained method (5) for some $\alpha \in [0, 1]$. Later, we shall show that this property is unique to the constrained method for a general model space $\mathcal{H}$.

So far all results consider the $0/1$ loss. In practice, a surrogate loss is often used to replace the $0/1$ loss, and a discriminant function is often used as the decision variable in the optimization problem. Here we propose to use the so-called psi loss $\psi(x) = \min(1, \max(0, 1 - x))$ [Shen et al., 2003], which could approximate the $0/1$ loss up to arbitrary precision. The rationale here is that a continuous loss that is close to the $0/1$ loss would enjoy similar nice theoretical properties of the $0/1$ loss. As a final note, we do not discuss convex surrogate loss for the constrained method, partly because if convex surrogate loss and convex set of discriminant function are used, the optimization problem associated with the constrained method is convex, and thus is equivalent to a weighted problem, which we will study subsequently in the next subsection.

### 3.2 Weighted method

Another popular method to estimate the ROC curve, which we call the weighted method, considers a set of classifiers obtained by assigning different class weights to the two types of errors. In practice, instead of directly optimizing the "weighted error rate", some continuous surrogate loss (often convex) is typically used to replace the non-continuous $0/1$ loss for ease of computation, which could introduce additional "model bias" or model mis-specification. This phenomenon has been extensively studied for standard binary classification task [see, e.g., Zhang et al., 2004; Bartlett et al., 2006]. Roughly speaking, the surrogate loss function needs to satisfy certain condition to ensure the consistency of the procedure (in terms of Bayes error). In Bartlett et al. [2006], the surrogate loss satisfying this condition is called to be classification-calibrated [Bartlett et al., 2006], and they further showed that procedures that are consistent using the $0/1$ loss, are also consistent using any classification-calibrated losses.

Here we shall present some parallel results for ROC curve estimation. More specifically, we consider the population analogue of the weighted method under a surrogate loss $V(\cdot)$ over a set of discriminant functions $\mathcal{F}$:

$$\underset{f \in \mathcal{F}}{\text{minimize}} \; \mathbb{E}\big((1 - w)\mathbb{I}(Y = 1) + w\mathbb{I}(Y = -1)\big)V(Yf(X)). \tag{10}$$

We denote by $f_{w,V(\cdot)}^\star$ a solution to (10), which may not necessarily be unique. Given the optimal discriminant function $f_{w,V(\cdot)}^\star$, we define the associated optimal classifier by $h_{w,V(\cdot)}^\star = \text{Sign}(f_{w,V(\cdot)}^\star)$, where $\text{Sign}(z) = 2\mathbb{I}(z \leq 0) - 1$ denotes the sign function. We aim to find conditions for the surrogate loss and the class of discriminant function, under which the population ROC curve of the weighted method using a surrogate loss $V(\cdot)$ over a set of discriminant functions $\mathcal{F}$, defined as

$$\gamma_{W,V(\cdot)}(\mathcal{F}) = \left\{ (\text{FPR}(h_{w,V(\cdot)}^\star), \text{TPR}(h_{w,V(\cdot)}^\star) \mid w \in [0, 1] \right\}, \tag{11}$$

can recover the optimal ROC curve over $\mathcal{H} = \{\text{Sign}(f) \mid f \in \mathcal{F}\}$. We first present a general condition on the class of discriminant functions $\mathcal{F}$ and the surrogate loss function $V(\cdot)$ to ensure consistent estimation of the optimal ROC curve.

**Definition 2** *(Correct model specification for weighted method) We say that a set of discriminant function $\mathcal{F}$ is correctly specified under a surrogate loss $V(\cdot)$ for the weighted method, if for any $0 \leq w \leq 1$ with $\mathbb{P}(\eta(X) = w) = 0$, any solution $f_{w,V(\cdot)}^\star$ to (10) satisfies $\{x : f_{w,V(\cdot)}^\star(x) > 0\} = \{x : \eta(x) > w\}$ up to a $\mathbb{P}_X$-null set.*

Under this notion of correct model specification, we can indeed show that the weighted method using the surrogate loss $V(\cdot)$ recovers the optimal ROC curve if the model space $\mathcal{F}$ is correctly specified under $V(\cdot)$.

**Theorem 3** *If $\mathcal{F}$ is correctly specified under a surrogate loss $V(\cdot)$ for the weighted method, then we have that*

$$\gamma_{W,V(\cdot)}(\mathcal{F}) = \Big\{(\alpha, s(\alpha)) : s(\alpha) \text{ is non-differentiable at } \alpha, \alpha \in (0, 1 - \mathbb{P}(\eta(X) = 0)/p^-)\Big\} \cup$$
$$\Big\{(\alpha, s(\alpha)) : \mathbb{P}(\eta(X) = c(\alpha)) = 0, \alpha \in (0, 1 - \mathbb{P}(\eta(X) = 0)/p^-)\Big\} \cup$$
$$\Big\{(FPR(h^\star_{w,V(\cdot)}), TPR(h^\star_{w,V(\cdot)})) : w \in [0, 1] \text{ and } \mathbb{P}(\eta(X) = w) > 0\Big\} \cup S_0' \cup S_1',$$

$$(12)$$

*where $S_0'$ and $S_1'$ satisfy*

$$S_0' = \begin{cases} \{(0, s(0))\}, & \text{if } c(0) < 1 \text{ or } \mathbb{P}(\eta(X) = 1) = 0 , \\ \emptyset, & \text{otherwise} , \end{cases} \tag{13}$$

*and*

$$S_1' = \begin{cases} \{(1 - \mathbb{P}(\eta(X) = 0)/p^-, 1)\}, & \text{if } \lim_{\alpha \to (1 - \mathbb{P}(\eta(X)=0)/p^-)^-} c(\alpha) > 0 \text{ or } \mathbb{P}(\eta(X) = 0) = 0 , \\ \emptyset, & \text{otherwise} . \end{cases}$$

$$(14)$$

Theorem 3 connects the population ROC curve of the weighted method to the optimal ROC curve when $\mathcal{H} = \mathcal{H}_a$. In Proposition S1 of the supplementary material, we establish that the set of $(\alpha, s(\alpha))$ with $\mathbb{P}(\eta(X) = c(\alpha)) > 0$ and $\alpha > 0$ corresponds to linear pieces on $\gamma(\mathcal{H}_a)$. Combining this with (12) in Theorem 3, it follows that except for some interior points of these linear pieces the other parts over optimal ROC curve can be recovered by weighted method. Therefore, roughly speaking, the set of points on the optimal ROC curve that can be recovered by the weighted method is the union of non-differentiable part and nonlinear part (up to some isolated points).

Although the above result seems to suggest that the weighted method may not recover all these linear pieces on the optimal ROC curve, it can be trivially modified to recover the entire optimal ROC curve by considering the pareto frontier of the convex hull of the ROC curve generated by weighted method. This is established in Corollary S2 of the supplementary material.

Next, we establish sufficient conditions on $V(\cdot)$ under which $\mathcal{F}_a$ is correctly specified under $V(\cdot)$, where $\mathcal{F}_a$ consists of all measurable functions. Interestingly, the requirement on $V(\cdot)$ coincides with the notion of classification-calibrated [Bartlett et al., 2006].

**Definition 3** *(Classification-calibrated loss [Bartlett et al., 2006]) Let $0 \le \eta \le 1$ and define*

$$H_V(\eta) = \inf_{\alpha \in \mathbb{R}} \eta V(\alpha) + (1 - \eta)V(-\alpha) \tag{15}$$

$$H_V^-(\eta) = \inf_{\alpha: \, \alpha(\eta - 1/2) \le 0} \eta V(\alpha) + (1 - \eta)V(-\alpha) \tag{16}$$

*where $V(\cdot)$ is a surrogate loss function. $V(\cdot)$ is said to be classification-calibrated if $H_V(\eta) < H_V^-(\eta)$ for any $\eta \neq 1/2$.*

In Bartlett et al. [2006], it is shown that using a classification-calibrated loss $V(\cdot)$ ensures consistent estimation of the optimal misclassification error, when $\mathcal{F}$ includes all measurable function (i.e., $\mathcal{F} = \mathcal{F}_a$). Now we establish that the class of all measurable discriminant functions $\mathcal{F}_a$ is correctly specified under any classification-calibrated surrogate loss.

**Proposition 2** *The class of all measurable functions $\mathcal{F}_a$ is correctly specified for the weighted method (i.e., in the sense of Definition 2) under the $0/1$ loss, and any classification-calibrated surrogate loss $V(\cdot)$.*

Combining this with Theorem 3, it follows that if $V(\cdot)$ is classification-calibrated and $\mathcal{F} = \mathcal{F}_a$, then the weighted method using the surrogate loss can recover the optimal ROC curve. In other words, for weighted method using classification-calibrated loss over $\mathcal{F} = \mathcal{F}_a$ generates consistent estimation of the optimal ROC curve.

Finally, we give an example in Theorem S1 of the supplementary material to show that in contrast to the constrained method (c.f. Proposition 1), the population ROC curve of the weighted method may miss some components of optimal ROC curve when the model space $\mathcal{F}$ is mis-specified under a surrogate loss $V(\cdot)$. Importantly, these components cannot be recovered even after taking the convex hull transformation.

### 3.3 Cutoff method

Cutoff method is another type of methods commonly used in practice. It first produces a discriminant function using some classification method, and then generates a set of classifiers by thresholding a sequence of cutoffs to the discriminant function. In order to study the population analogue of the cutoff method, we assume that the discriminant function is generated from a given classification method with class weight $w \in [0, 1]$ for negative instances:

$$f^{\star}_{w, V(\cdot)} \in \underset{f \in \mathcal{F}}{\arg\min} \; \mathbb{E}\big((1 - w)\mathbb{I}(Y = 1) + w\mathbb{I}(Y = -1)\big)V(Yf(X)), \tag{17}$$

where $\mathcal{F}$ is the set of discriminant functions. We define the population ROC curve of the cutoff method over $\mathcal{F}$ to be

$$\gamma_{T, V(\cdot)}(\mathcal{F}, w) = \left\{ (\text{FPR}(\text{Sign}(f^{\star}_{w, V(\cdot)} - \delta)), \text{TPR}(\text{Sign}(f^{\star}_{w, V(\cdot)} - \delta))) \mid \delta \in \mathbb{R} \right\}. \tag{18}$$

Next, we establish sufficient conditions on $\mathcal{F}$ and $V(\cdot)$ to ensure that the cutoff method can recover the optimal ROC curve. Similar to the weighted method, we define a notion of correct model specification for the cutoff method.

**Definition 4** *(Correct model specification for cutoff method) We say that a set of discriminant function $\mathcal{F}$ is correctly specified under a surrogate loss function $V(\cdot)$ for the cutoff method for some $w \in (0, 1)$, if any solution to (17) at $w$ is a strictly increasing transformation to $\eta(x)$ almost surely, that is,*

$$f^{\star}_{w, V(\cdot)}(X) = M(\eta(X)) \; \text{almost surely},$$

*where $M(\cdot)$ is a strictly increasing function.*

Under this notion of correct model specification, we show that the cutoff method targets the optimal ROC curve when $\eta(X)$ is a continuous random variable. This is established in the following theorem.

**Theorem 4** *Assume $\eta(X)$ is a continuous random variable. If $\mathcal{F}$ is correctly specified under a surrogate loss $V(\cdot)$ for the cutoff method for some $w \in [0, 1]$, then we have that*

$$\gamma_{T, V(\cdot)}(\mathcal{F}, w) = \gamma(\mathcal{H}_a).$$

For the cutoff method, we can show that the set of all measurable discriminant functions is correctly specified under any surrogate loss $V(\cdot)$ that is differentiable, strictly decreasing, and strictly convex.

**Theorem 5** *Assume $\eta(X)$ is a continuous random variable. Suppose that $V(\cdot)$ is a differentiable, strictly decreasing, proper and strictly convex loss. Then the class of all measurable functions $\mathcal{F}_a$ is correctly specified under $V(\cdot)$ for the cutoff method at any $w \in (0, 1)$.*

An important implication of the above two theorems is that, the cutoff method defined in (18) can recover the optimal ROC curve if $\mathcal{F} = \mathcal{F}_a$ and the surrogate loss $V(\cdot)$ is differentiable, strictly decreasing and strictly convex. Note that the requirement on the surrogate loss for the model to be correctly specified for the cutoff method is more stringent than the classification-calibrated requirement for the weighted method (c.f., Lemma S7 in the supplementary material).

For a general model space $\mathcal{F}$, the requirement of correct model specification for the cutoff method is also generally stronger than those needed for the weighted method. This is because in general we would not know whether $f^{\star}_{w, V(\cdot)}(x)$ is an strictly increasing function of $\eta(x)$, even if $\text{Sign}(f^{\star}_{w, V(\cdot)}(x)) = \text{Sign}(\eta(x) - w)$ and $\eta(X)$ is a continuous random variable. In Theorem S2 of the supplementary material, we give an example to show that in contrast to the constrained method (c.f. Proposition 1), the population ROC curve generated by the cutoff method over $\mathcal{F}$ may be dominated by the optimal ROC curve over $\mathcal{H} = \{\text{Sign}(f - \delta) \mid f \in \mathcal{F}, \delta \in \mathbb{R}\}$, suggesting that the cutoff method may not recover the optimal ROC curve over $\mathcal{H}$ in general.

## 4 Simulations and real data analysis

In this section, we examine the operating characteristics of the three methods analyzed in this article with both simulation studies and a real data example. In our simulation, we investigate the effect of

model mis-specification by considering linear classifiers over two simulated data sets. One dataset is generated under a linear discriminant analysis (LDA) setting, which imitates the scenario where model is correctly specified. The other is based on a quadratic discriminant analysis (QDA) setting, which is used to show the differences across three methods under model mis-specification. We also compare the three methods by using a bank marketing data set [Moro et al., 2014], which allows us to investigate the effect model mis-specification on the performance of ROC curve estimation methods in real problem setting. Throughout all the experiments, we consider linear classification methods including constrained $\psi$-learning, weighted SVM and its cutoff version, and also typical kernel methods that include kernel weighted SVM and its cutoff version

**Example 1 (LDA setting).**   This example considers a linear discriminant analysis model, where $\mathbb{P}(Y = 1) = \mathbb{P}(Y = -1) = 1/2$, and conditioned on $Y$, $X \mid Y \sim N(Y\boldsymbol{\mu}, 4\boldsymbol{I})$, where $\boldsymbol{\mu} = (1, 1)^\top \in \mathbb{R}^2$. The sample size is $n = 2000$.

**Example 2 (QDA setting).**    This example considers a quadratic discriminant analysis model, where $\mathbb{P}(Y = 1) = \mathbb{P}(Y = -1) = 1/2$, and conditioned on $Y$, $X \mid Y \sim N(Y\boldsymbol{\mu}, \boldsymbol{\Sigma}_Y)$, where $\boldsymbol{\mu} = (1, 1)^\top \in \mathbb{R}^2$, and

$$\Sigma_+ = \begin{pmatrix} 8 & 5 \\ 5 & 4 \end{pmatrix}, \ \Sigma_- = \begin{pmatrix} 1 & -3 \\ -3 & 16 \end{pmatrix} .$$

The sample size is $n = 2000$.

**Example 3 (Real data example).**   This example considers a bank marketing dataset [Moro et al., 2014], which records the direct marketing campaigns of a Portuguese banking institution and are available at `https://archive.ics.uci.edu/ml/datasets/Bank+Marketing`. This real dataset consists of $4521$ examples with 17 features and 1 binary response variable.

We visualize the scatter plots of two simulation examples in Figure 1. For two simulation examples, we compare the average estimated ROC curve for the five aforementioned methods over 100 replicates; while for the real data example, we present the estimated ROC curves directly. For all three examples, to generate the estimated ROC curve we vary weight $w$ and the constraint upper bound $\alpha$ from $\{i/500 \mid i = 0, 1, \ldots, 500\}$ for the weighted method and the constrained method, respectively. For the two cutoff methods, we choose $w = 1/2$. The results are summarized in Figure 2 and Figure S1 of the supplementary material.

The left panel of Figure 2 is for the simulations under LDA setting, where the model is correctly specified for all methods. In this case, all methods perform almost the same. The right panel of Figure 2 is the for the simulations under QDA setting. In this case, the model is mis-specified for linear methods. As expected, the two kernel methods perform the best. Moreover, among linear methods, the constrained method performs the best, which corroborates with our theoretical findings that constrained method performs best under possible model mis-specification. For our real data example, we can see in Figure S1 that among all these linear method the constrained method performs best, but is no better than the weighted kernel SVM. These results again corroborate with our theoretical analysis on the goodness of using constrained method when model is mis-specified.

## 5   Discussion

In this article, we define a general notion of optimal ROC curve, which we argue should be the theoretical target of any ROC curve estimation methods. We analyze three popular ROC estimation methods, and show that when the model space contains all measurable discriminant functions all three methods target the optimal ROC curve, although the cutoff method requires a stronger set of conditions on the surrogate loss for it to be consistent. When model mis-specification is present, we show that only the constrained method under the $0/1$ loss targets the optimal ROC curve over the used model space. This together with our simulations and real data studies suggests that the constrained method with a non-convex surrogate loss (e.g. $\psi$ loss) would be the preferred method when the model is possibly mis-specified. Although we have observed good empirical performances using the constrained method with $\psi$ loss, we have not investigated its theoretical guarantees in this article. Further theoretical research on non-convex surrogate loss to achieve guaranteed performance may be necessary.

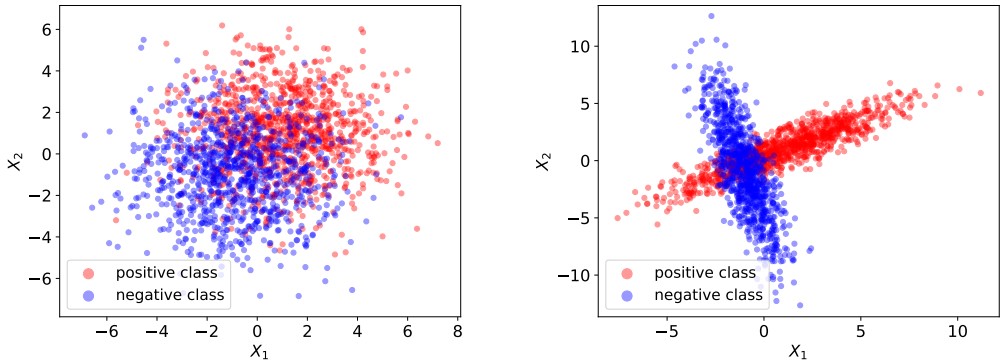

Figure 1: Scatter plots of dataset used in LDA setting (left panel) and QDA setting (right panel).

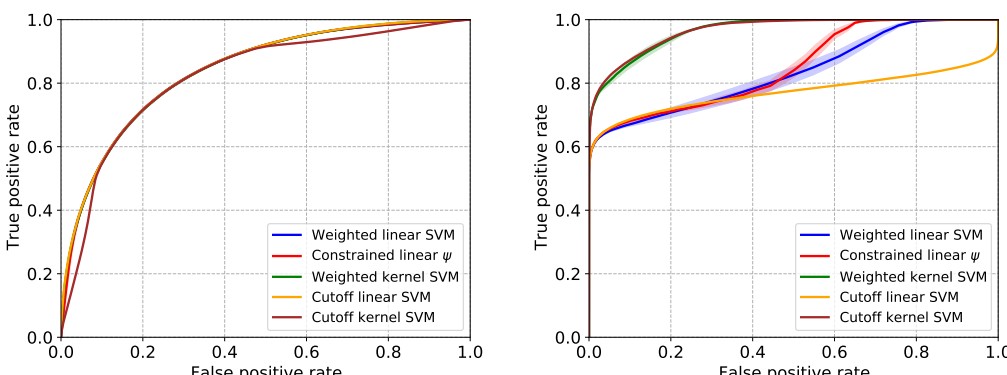

Figure 2: Estimated ROC curves of five different methods under the LDA setting (left panel) and QDA setting (right panel). Here "Constrained linear $\psi$", "Weighted linear SVM", "Cutoff linear SVM", "Weighted kernel SVM" and "Cutoff kernel SVM" denote constrained $\psi$-learning with linear discriminant functions, weighted SVM with linear discriminant functions, cutoff method applied to SVM with linear discriminant functions, weighted SVM with RBF kernel, and cutoff method applied to SVM with RBF kernel.

## Acknowledgments and Disclosure of Funding

The authors are supported in part by the US National Science Foundation (NSF) under grant DMS-20-15490. The authors thank the reviewers for valuable comments and suggestions that improved the article.

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
