# Supplementary Material for "On the consistent estimation of optimal Receiver Operating Characteristic (ROC) curve"

This supplementary material collects the practical issues of using the three methods discussed in the main file, the plot of results by the five methods when applied to a real data example in the main file, some additional corollaries of main results, discussions on the special case of weighted method using 0/1 loss, some additional properties of optimal ROC curve and all the proofs of the theorems and corollaries in the main text.

## A    Practical issues

In this section, we discuss the practical implications of the theoretical results presented in the main file. We also touch on some practical implementation issues of these three methods, and some comments on their relative pros and cons in terms of computation and ease of implementation.

First, the cutoff method is probably the easiest to implement among the three methods, as it only requires running a classification algorithm that outputs a discriminant function. Moreover, it also tends to be the most efficient method in terms of computation as the cutoff step does not incur significant additional computations. This could partly explain why it is so widely used in practice [see, e.g., Fawcett, 2006]. However, as our theory suggested, one should exercise special caution when using the cutoff method, as when a simple model is used, it is not easy to check whether the model is correctly specified or not. Even when the model space is large (e.g., nonparametric method), the requirement on the loss function is quite stringent (c.f. Theorem 5). For example, in

view of Theorem 5 and Proposition 2, we would not recommend using the cutoff method on the discriminant functions produced by the support vector machine (SVM)

The weighted method is also relatively easy to implement as long as it is possible to modify existing software to accommodate unequal class weights. Computationally, since it requires repeated applications of existing classification methods over a set of class weights, it is often less efficient than the cutoff method. On the statistical side, our theory suggests that similar to the cutoff method, the weighted method targets the optimal ROC curve over $\mathcal{H}$ only when the model is correctly specified, although the requirement on the surrogate loss may be less stringent. This happens when one considers a nonparametric classifier with a classification-calibrated loss function. For applications where a simple model has to be used, however, we would not recommend the use of the weighted method as it does not target the optimal ROC curve over the model space.

The constrained method seems to be the most difficult to implement as it is challenging to leverage existing classification software, especially when nonconvex surrogate loss is used. We also note that the use of nonconvex surrogate loss function is essential whenever the constrained method is useful, because if convex surrogate loss is used to replace the $0/1$ loss, the constrained method would be equivalent to the weighted method, which could be suboptimal when the model is mis-specified. As such, when a simple model needs to be used in practice (e.g., for the purpose of better model interpretability), we recommend the constrained method with a nonconvex surrogate loss that approximates the $0/1$ loss. This recommendation will also be corroborated numerically in later sections that using the psi loss $\psi(x) = \min(1, \max(0, 1-x))$ [Shen et al., 2003] in linear classifiers indeed leads to better ROC curve estimation compared with those using a convex surrogate loss or procedures based on weighted or cutoff methods.

# B  Plot of real data example in Section 4

In this section, we include the plot of estimated ROC curves by five different methods on the bank marketing dataset in Figure S1.

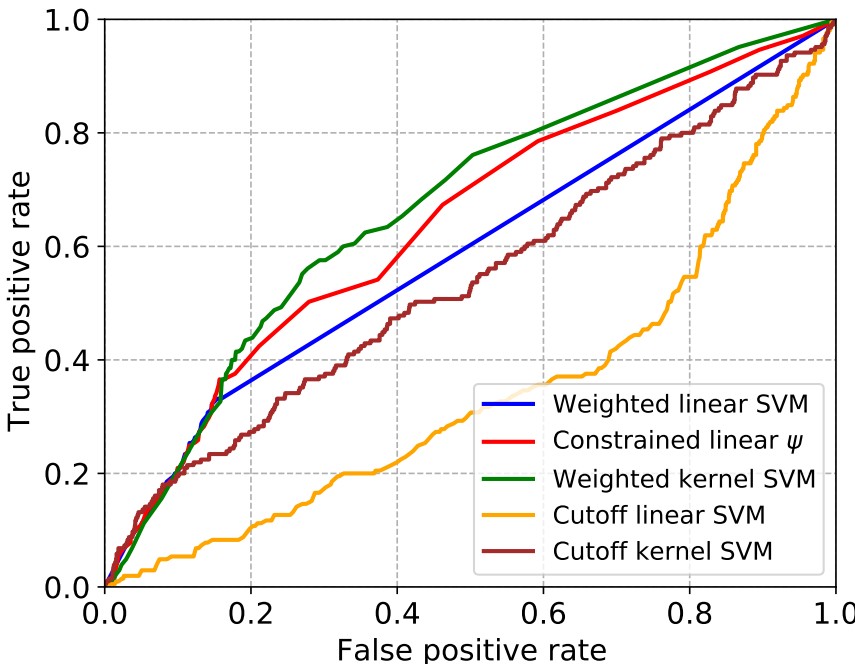

Figure S1: Estimated ROC curves of five different methods on the *bank marketing* dataset. Here "Constrained linear Psi" , "Weighted linear SVM", "Cutoff linear SVM", "Weighted kernel SVM" and "Cutoff kernel SVM" denote constrained $\psi$-learning with linear discriminant functions, weighted SVM with linear discriminant functions, cutoff method applied to SVM with linear discriminant functions, weighted SVM with RBF kernel, and cutoff method applied to SVM with RBF kernel.

# C  Additional theorems and corollaries in the main file

In this section, we include some additional corollaries and theorems of the main file. All the proofs of these results are placed in Section G.

Our first corollary shows that under the condition (3), our proposed definition of optimal ROC curve coincides with the existing definition (4).

**Corollary S1.** *Assume that* (3) *holds. Then*

$$\{(FPR(h_\lambda), TPR(h_\lambda)) \mid \lambda \in [0, \infty]\} = \gamma(\mathcal{H}_a)$$
$$= \{(FPR(Sign(\eta(x) - c)), TPR(Sign(\eta(x) - c))) \mid 0 \le c \le 1\} . \quad \text{(S1)}$$

Our next results show that, although Theorem 3 in the main file seems to suggest that the

weighted method may not recover all these linear pieces on the optimal ROC curve, it can be trivially modified to recover the entire optimal ROC curve by considering the pareto frontier of the convex hull of the ROC curve generated by weighted method.

**Corollary S2.** *If $\mathcal{F}$ is correctly specified under a surrogate loss $V(\cdot)$ for the weighted method, then we have that*

$$PF(\overline{conv}(\gamma_{W,V(\cdot)}(\mathcal{F}))) = \gamma(\mathcal{H}_a)\,, \tag{S2}$$

*where $\overline{conv}(\gamma_{W,V(\cdot)}(\mathcal{F}))$ denotes the closed convex hull of $\gamma_{W,V(\cdot)}(\mathcal{F})$.*

Our next two theorems give some examples on showing that optimal ROC curve may not be obtained by weighted method (Theorem S2) and cutoff method (Theorem S2) when model misspecification is present.

**Theorem S1.** *Denote $\mathcal{H} = \{Sign(f) \mid f \in \mathcal{F}\}$. In general, it is possible that*

$$\gamma(\mathcal{H}) \setminus \gamma^{\star}_{W,V(\cdot)}(\mathcal{F}) \neq \emptyset \text{ and } \gamma(\mathcal{H}) \setminus PF(\overline{conv}(\gamma_{W,V(\cdot)}(\mathcal{F}))) \neq \emptyset\,. \tag{S3}$$

**Theorem S2.** *The population ROC curve of the cutoff method over certain model class $\mathcal{F}$ may be dominated by the optimal ROC curve over $\mathcal{H} = \{Sign(f - \delta) \mid f \in \mathcal{F}, \delta \in \mathbb{R}\}$, in the sense that there exists a situation where for some $(u,v) \in \gamma_{T,V(\cdot)}(\mathcal{F}, w)$, there exists $(u', v') \in \gamma(\mathcal{H})$ such that $(u', v') \succ (u, v)$.*

# D   Weighted method using $0/1$ loss

In this section, we further discuss the special case of weighted method where $0/1$ loss is used. We will first analyze the population analogue of this approach under the $0/1$ loss, which is

$$\underset{h \in \mathcal{H}}{\text{minimize }} \mathbb{E}\big\{((1-w)\mathbb{I}(Y=1) + w\mathbb{I}(Y=-1))\mathbb{I}(h(X) \neq Y)\big\}\,, \tag{S4}$$

where $w \in [0, 1]$ is the class weight for negative instances, and $\mathcal{H}$ is a set of classifiers. Note that when $w = 1/2$, this reduces to the minimization of misclassification error rate. We call the ROC curves generated from solutions to (S4) by varying $w \in [0, 1]$ the population ROC curve of the weighted method, which is defined to be

$$\gamma_W(\mathcal{H}) = \{(\mathrm{FPR}(h_{w,\mathcal{H}}^\star), \mathrm{TPR}(h_{w,\mathcal{H}}^\star)) \mid w \in [0, 1]\}, \tag{S5}$$

where $h_{w,\mathcal{H}}^\star$ is a solution to (S4) and may not be unique. When $\mathcal{H} = \mathcal{H}_a$, we simply use $h_w^\star$ to denote $h_{w,\mathcal{H}_a}^\star$.

Our next result derives the population ROC curve of the weighted method when $\mathcal{H} = \mathcal{H}_a$ includes all measurable classifiers. Unlike the constrained method, it is shown that the population ROC curve of the weighted method may miss some linear pieces on the optimal ROC curve, where those linear pieces correspond to point masses of $\eta(X)$ (c.f. Proposition S1 of supplementary material). However, this would not affect the consistency of weighted method, as one can easily consider a linear extension to achieve consistency (see Figure S2 for an example).

**Theorem S3.** *Suppose that $\mathcal{H} = \mathcal{H}_a$. Then, any solution $h_w^\star$ to the weighted problem (S4) must satisfy*

$$\{x : h_w^\star(x) = 1\} = \{x : \eta(x) > w\} \cup \mathcal{N}_w \tag{S6}$$

*for some $\mathcal{N}_w \subseteq \{x : \eta(x) = w\}$. Moreover, we have that*

$$\begin{aligned}
\gamma_W(\mathcal{H}_a) = &\left\{(\alpha, s(\alpha)) : \mathbb{P}(\eta(X) = c(\alpha)) = 0, \alpha \in (0, 1 - \mathbb{P}(\eta(X) = 0)/p^-)\right\} \cup \\
&\left\{(\alpha, s(\alpha)) : \mathbb{P}(\eta(X) = c(\alpha)) > 0, \alpha = FPR(h_w^\star) \in (0, 1 - \mathbb{P}(\eta(X) = 0)/p^-)\right\} \cup \\
&\left\{(\alpha, s(\alpha)) : s(\alpha) \text{ is non-differentiable at } \alpha, \alpha \in (0, 1 - \mathbb{P}(\eta(X) = 0)/p^-)\right\} \cup \\
&\left\{(0, s_0)\right\} \cup \left\{(s_1, 1)\right\} \cup S_0 \cup S_1
\end{aligned} \tag{S7}$$

*for some $s_0 \in [0, \mathbb{P}(\eta(X) = 1)/p^+]$ and $s_1 \in [1 - \mathbb{P}(\eta(X) = 0)/p^-, 1]$, where $S_0$ and $S_1$ satisfies*

$$S_0 = \begin{cases} \emptyset, & \text{if } c(0) = 1, \\ \{(0, s(0))\}, & \text{otherwise}, \end{cases} \tag{S8}$$

*and*

$$S_1 = \begin{cases} \emptyset, & \text{if } \lim_{\alpha \to (1 - \mathbb{P}(\eta(X) = 0)/p^-)^-} c(\alpha) = 0, \\ \{(1 - \mathbb{P}(\eta(X) = 0)/p^-, 1)\}, & \text{otherwise}. \end{cases} \tag{S9}$$

Theorem S3 connects the population ROC curve of the weighted method to the optimal ROC curve when $\mathcal{H} = \mathcal{H}_a$. Clearly, from (S7), we can see that the population ROC curve of the weighted method is a subset of the optimal ROC curve (except for $\{(0, s_0)\} \cup \{(s_1, 1)\}$). Moreover, we also establish in Proposition S1 of the supplementary material that the set of $(\alpha, s(\alpha))$ with $\mathbb{P}(\eta(X) = c(\alpha)) > 0$ and $\alpha > 0$ corresponds to linear pieces on $\gamma(\mathcal{H}_a)$. Combining this with (S7) in Theorem S3, it follows that the interior of these linear pieces can not be recovered by the weighted method. In fact, only one point in each linear piece can be recovered by the weighted method (i.e., the second set in (S7)). Therefore, roughly speaking, the set of points on the optimal ROC curve that can be recovered by the weighted method is the union of non-differentiable part and nonlinear part (up to some isolated points).

The following result derives the pareto frontier of $\gamma_W(\mathcal{H}_a)$, defined as $\gamma_W^\star(\mathcal{H}_a) = \text{PF}(cl(\gamma_W(\mathcal{H}_a)))$, and relates it to the optimal ROC curve.

**Corollary S3.** *Denote by $I_l = (\alpha_l^-, \alpha_l^+)$ all the disjoint intervals over which $s(\alpha)$ is linear, and $\mathcal{I} = \bigcup_l I_l$. Then*

$$\gamma_W^\star(\mathcal{H}_a) = \cup_l \{(\alpha_l, s(\alpha_l))\} \cup \{(\alpha, s(\alpha)) \mid \alpha \in [0, 1 - \mathbb{P}(\eta(X) = 0)/p^-] \setminus \mathcal{I}\} \subseteq \gamma(\mathcal{H}_a), \tag{S10}$$

*for some $\alpha_l \in [\alpha_l^-, \alpha_l^+]; l = 1, 2, \ldots$.*

The above corollary shows that the pareto frontier of the population ROC curve for the weighted

method is only a subset of the optimal ROC curve. In particular, parts of the linear pieces on the optimal ROC curve may not be recovered by the weighted method.

Although the above result seems to suggest that the weighted method could not recover the linear pieces on the optimal ROC curve, it can be trivially modified to recover the entire optimal ROC curve. In particular, we consider pareto frontier of the convex hull of the ROC curve generated by the weighted method. Formally, this is defined as $\mathrm{PF}(\overline{\mathrm{conv}}(\gamma_W(\mathcal{H}_a)))$, where $\overline{\mathrm{conv}}(\gamma_W(\mathcal{H}_a))$ is the closed convex hull of $\gamma_W(\mathcal{H}_a)$. The following corollary shows that $\mathrm{PF}(\overline{\mathrm{conv}}(\gamma_W(\mathcal{H}_a)))$ is exactly the optimal ROC curve. Also see Figure S2 for an illustration of this in a simple example.

**Corollary S4.** *We have that*

$$PF(\overline{conv}(\gamma_W(\mathcal{H}_a))) = \gamma(\mathcal{H}_a). \tag{S11}$$

To summarize, just like the constrained method, the weighted method can also recover the optimal ROC curve when the model space includes all measurable classifiers, which seems to suggest that there is no real advantage of taking the constrained approach when the goal is to estimate the optimal ROC curve. However, somewhat surprisingly, when the model space $\mathcal{H}$ is not the set of all measurable classifiers, the population ROC curve of the weighted method $\gamma_W(\mathcal{H})$ is only a subset of the optimal ROC curve $\gamma(\mathcal{H})$. More importantly, there exist situations where the pareto frontier of its closed convex hull, defined as $\gamma_W^\star(\mathcal{H}) = \mathrm{PF}(cl(\gamma_W(\mathcal{H})))$, is a proper subset of $\gamma(\mathcal{H})$. This is established in the following theorem.

**Theorem S4.** *In general, we have that*

$$\gamma_W(\mathcal{H}) \setminus \{(FPR(h_{w,\mathcal{H}}^\star), TPR(h_{w,\mathcal{H}}^\star)) : w = 0 \ or \ w = 1\} \subseteq \gamma(\mathcal{H}), \tag{S12}$$

*and $\gamma_W^\star(\mathcal{H}) \subseteq \gamma(\mathcal{H})$ if $S(\mathcal{H}) = \{(FPR(h), TPR(h)) \mid h \in \mathcal{H}\}$ is closed. Moreover, it is possible that*

$$\gamma(\mathcal{H}) \setminus \gamma_W^\star(\mathcal{H}) \neq \emptyset \ and \ \gamma(\mathcal{H}) \setminus PF(\overline{conv}(\gamma_W(\mathcal{H}))) \neq \emptyset. \tag{S13}$$

This above theorem suggests that, unlike the constrained method, the population ROC curve of

the weighted method could be a proper subset of the optimal ROC curve over general model space $\mathcal{H}$. Again, this reveals that the weighted method may miss some FPR-TPR pairs on the optimal ROC curve over a general model space. An example is given in Figure S3, where the weighted method does not recover the optimal ROC curve over a particular model space.

Combining the above theorem with Proposition 1, we have the following general relation

$$\gamma_W^\star(\mathcal{H}) \subseteq \gamma(\mathcal{H}) = \gamma_C^\star(\mathcal{H}) \tag{S14}$$

for a general model space $\mathcal{H}$ as long as $S(\mathcal{H})$ is closed, and it is possible that $\gamma_W^\star(\mathcal{H}) \neq \gamma(\mathcal{H})$. Therefore, the weighted method may be affected by model mis-specification while the constrained method always targets the optimal ROC curve. When the model space contains all measurable classifiers, both methods can recover the optimal ROC curve.

In sum, the weighted method may not be an appropriate approach when the model is clearly mis-specified. When the model is correctly specified, however, the weighted approach may be the preferred choice as it is in general computationally easier to deal with compared with the constrained method.

# E   Additional properties of optimal ROC curve

We first derive some additional properties of the optimal ROC curve under $\mathcal{H}_a$ in this section. First, we establish an one-to-one correspondence between the point masses of $\eta(X)$ and the linear pieces on the optimal ROC curve.

**Proposition S1.** *If $\mathbb{P}(\eta(X) = c(\tilde{\alpha})) > 0$ for some $\tilde{\alpha}$ with $c(\tilde{\alpha}) < 1$, then the optimal ROC curve $s(\alpha)$ as defined in (2) is locally linear over the interval*

$$I(\tilde{\alpha}) := \left[ \frac{\mathbb{E}(1 - \eta(X))\mathbb{I}(\eta(X) > c(\tilde{\alpha}))}{p^-} , \frac{\mathbb{E}(1 - \eta(X))\mathbb{I}(\eta(X) \geq c(\tilde{\alpha}))}{p^-} \right) ,$$

*and $c(\alpha) = c(\tilde{\alpha})$ for any $\alpha \in I(\tilde{\alpha})$. Conversely, if $s(\alpha)$ is locally linear in a neighborhood of $\tilde{\alpha}$,*

*denoted as $\delta_{\tilde{\alpha}}$, then $\mathbb{P}(\eta(X) = c(\tilde{\alpha})) > 0$, and $c(\alpha) = c(\tilde{\alpha})$ for any $\alpha \in \delta_{\tilde{\alpha}}$.*

The above proposition shows that the point masses of $\eta(X)$ correspond to linear pieces on the optimal ROC curve. This is also connected to the likelihood ratio assumption in (3) imposed by Scott [2007], which is basically equivalent to saying that $\eta(X)$ does not have any point mass. In other words, the conventional definition of optimal ROC curve without the likelihood ratio assumption may miss some linear pieces on the optimal ROC curve.

Next, we study the differentiability of the optimal ROC curve $s(\alpha)$ over its domain $[0, 1 - \mathbb{P}(\eta(X) = 0)/p^-]$, which turns out to be useful for analysis of consistency. In Theorem 1, we have already shown that $s(\alpha)$ is concave over its domain $[0, 1 - \mathbb{P}(\eta(X) = 0)/p^-]$. Hence, the left and right derivative of $s(\alpha)$ must exist over the interior of its domain. The following proposition establishes necessary and sufficient conditions under which $s(\alpha)$ is differentiable.

**Proposition S2.** *$s(\alpha)$ is left-differentiable at $\alpha = 1 - \mathbb{P}(\eta(X) = 0)/p^-$; for any $0 < \alpha < 1 - \mathbb{P}(\eta(X) = 0)/p^-$, $s(\alpha)$ is differentiable at $\alpha$ if and only if $c(\alpha)$ is continuous at $\alpha$; and $s(\alpha)$ is right-differentiable at $\alpha = 0$ if and only if $c(0) < 1$.*

Note that for any $\alpha \in (0, 1 - \mathbb{P}(\eta(X) = 0)/p^-)$, the differentiability of $s(\alpha)$ is related to the continuity of $c(\alpha)$. Conditions under which $c(\alpha)$ is continuous is provided in part (iii) of Lemma S3. To the best of our knowledge, these properties about the optimal ROC curve have not been formally established in the literature. We include proofs of Proposition S1 and S2 in Section G.

# F   Supporting Lemmas

In this section, we present some supporting results to be used later in the proofs.

**Lemma S1.** *Assume that $A \subseteq \mathbb{R}^2$ is a bounded and closed set. For any $(u, v) \in A \setminus PF(A)$, there must exist a $(u', v') \in PF(A)$ such that $(u', v') \succ (u, v)$. Moreover, for any compact set $B$ satisfying $PF(A) \subseteq B \subseteq A$, we have that $PF(B) = PF(A)$.*

**Proof of Lemma S1.** We prove the first claim by contradiction. Suppose that there exists $(\tilde{u}, \tilde{v}) \in A \setminus \mathrm{PF}(A)$ such that nothing in $\mathrm{PF}(A)$ dominates $(\tilde{u}, \tilde{v})$. Denote by $B(u, v) = \{(u', v') \mid (u', v') \succ (u, v)\} \cup \{(u, v)\}$. Then we must have $B(\tilde{u}, \tilde{v}) \cap \mathrm{PF}(A) = \emptyset$. Since $A$ is closed and bounded, we have that $B(\tilde{u}, \tilde{v}) \cap A$ is compact. Denote by $(u^\star, v^\star) \in B(\tilde{u}, \tilde{v}) \cap A$ the point in $B(\tilde{u}, \tilde{v}) \cap A$ that maximizes $v - u$, that is,

$$v^\star - u^\star \geq v - u \text{ for any } (u, v) \in B(\tilde{u}, \tilde{v}) \cap A\,.$$

Since $(\tilde{u}, \tilde{v}) \in A \setminus \mathrm{PF}(A)$, it can be dominated by another point $(u', v')$ in $A$. By definition of $B(\tilde{u}, \tilde{v})$, we have that $(u', v') \in B(\tilde{u}, \tilde{v})$, and thus $(u', v') \in B(\tilde{u}, \tilde{v}) \cap A$. Now, we have that $v^\star - u^\star \geq v' - u' > \tilde{v} - \tilde{u}$, which implies that $(u^\star, v^\star) \neq (\tilde{u}, \tilde{v})$. Next, we show that $(u^\star, v^\star) \in \mathrm{PF}(A)$. If not, then there exists $(u'', v'') \in A$ that dominates $(u^\star, v^\star)$, and hence also dominates $(\tilde{u}, \tilde{v})$. This implies that $(u'', v'') \in B(\tilde{u}, \tilde{v}) \cap A$ and $v' - u' > v^\star - u^\star$, which contradicts with the definition of $(u^\star, v^\star)$. This completes the proof of the first claim.

For the second claim, for any $x \in \mathrm{PF}(A)$, we have that there is no point in $A$ that can dominates $x$. Since $\mathrm{PF}(A) \subseteq B \subseteq A$, we have that $x \in B$ and there is no point in $B$ that can dominates $x$. Hence, $x \in \mathrm{PF}(B)$. Therefore, $\mathrm{PF}(A) \subseteq \mathrm{PF}(B)$. Conversely, for any $x \in \mathrm{PF}(B)$, suppose that $x \notin \mathrm{PF}(A)$. Then by using the first claim, we could find $x' \in \mathrm{PF}(A)$ that dominates $x$. But $\mathrm{PF}(A) \subseteq B$ and thus $x' \in B$ and dominates $x$, which contradicts with the fact that $x \in \mathrm{PF}(B)$. Hence, we must have $x \in \mathrm{PF}(A)$, and therefore $\mathrm{PF}(B) \subseteq \mathrm{PF}(A)$. This completes the proof.

**Lemma S2.** *For $\alpha \in [0, 1]$, denote by*

$$\gamma(\alpha) = \{(FPR(h^\star_{\alpha, \mathcal{H}}), TPR(h^\star_{\alpha, \mathcal{H}})) : h^\star_{\alpha, \mathcal{H}} \text{ is a solution to } (5) \text{ at } \alpha\} \tag{S15}$$

*the set of FPR-TPR pairs generated by all possible solutions to (5) at $\alpha$.*

*(i) If $\gamma(\alpha)$ is a nonempty non-singleton, then $v = v'$ for any $(u, v), (u', v') \in \gamma(\alpha)$;*

*(ii) if $S(\mathcal{H})$ is a closed set, then $\gamma(\alpha^\star)$ is a singleton and $\gamma(\alpha^\star) \subseteq \gamma(\alpha)$, where $\alpha^\star = \inf\{u : (u, v) \in$*

$\gamma(\alpha)\}$.

**Proof of Lemma S2.** We first prove claim (i). Denote by $h$ and $h'$ any two solutions to (5) at $\alpha$. By optimality of $h$ and $h'$, it follows that $\text{TPR}(h') = \text{TPR}(h)$, which proves (i).

Next, we prove (ii). Let $v^\star = \text{TPR}(h)$, where $h$ is any solution to (5) at $\alpha$. By (i), we know that $v^\star$ is a constant regardless of the choice of $h$. This, together with the fact that $S(\mathcal{H})$ is a closed set, implies that $(\alpha^\star, v^\star) \in cl(\gamma(\alpha)) \subseteq S(\mathcal{H})$ by definition of $\alpha^\star$. Therefore, there exists $h^\star \in \mathcal{H}$ such that $(\text{FPR}(h^\star), \text{TPR}(h^\star)) = (\alpha^\star, v^\star)$. Moreover, $h^\star$ must be a solution to (5) at $\alpha^\star$, because $\text{TPR}(h) \leq v^\star$ for any $h$ that is feasible for (5) at $\alpha$. This implies that $(\alpha^\star, v^\star) \in \gamma(\alpha^\star)$.

Next, we prove that $\gamma(\alpha^\star)$ must be a singleton. Suppose $\gamma(\alpha^\star)$ is a nonempty non-singleton, then there must exist a solution $h'$ to (5) at $\alpha^\star$ such that $\text{TPR}(h') = \text{TPR}(h^\star) = v^\star$ and $\text{FPR}(h') < \text{FPR}(h^\star) = \alpha^\star$. Therefore, $h'$ must also be a solution to (5) at $\alpha$. This contradicts the definition of $\alpha^\star$. Hence, we have that $\gamma(\alpha^\star)$ is a singleton.

Finally, by using the fact that $\text{TPR}(h^\star) = v^\star = \text{TPR}(h)$, where $h$ is any solution to (5) at $\alpha$, we obtain that $\gamma(\alpha^\star) \subseteq \gamma(\alpha)$. This completes the proof.

**Lemma S3.** *Let* $\eta(X) = \mathbb{P}(Y = 1 \mid X)$ *and* $g(c) = \mathbb{E}(1 - \eta(X))\mathbb{I}(\eta(X) > c)$ *and* $c(\alpha) = \inf\{c \in [0, 1], g(c) \leq \alpha p^-\}$. *Then,*

(i) $c(\alpha) < 1$ *if* $\alpha > 0$ *and* $c(\alpha) > 0$ *if* $\alpha < 1 - \frac{\mathbb{P}(\eta(X)=0)}{p^-}$;

(ii) $g(c(\alpha)) \leq \alpha p^-$ *and* $\mathbb{E}(1 - \eta(X))\mathbb{I}(\eta(X) \geq c(\alpha)) \geq \alpha p^-$ *for all* $\alpha \in [0, 1]$;

(iii) $c(\alpha)$ *is a non-increasing and right-continuous function over* $[0, 1)$, *and it is discontinuous at* $\alpha \in (0, 1 - \frac{\mathbb{P}(\eta(X)=0)}{p^-}]$ *if and only if* $g(c(\alpha)) = \alpha p^-$ *and there exists* $c \in (c(\alpha), 1)$ *such that* $\mathbb{P}(c(\alpha) < \eta(X) < c) = 0$.

**Proof of Lemma S3.** We start by checking claim (i). First note $\lim_{c \to 1^-} g(c) = \mathbb{E}(1 - \eta(X))\mathbb{I}(\eta(X) = 1) = 0 = g(1)$. For $\alpha > 0$, by property of limit, there must exist $\delta > 0$ such that for any $1 - \delta < c < 1$, $g(c) < \alpha p^-$, which means $c(\alpha) < 1$ when $\alpha > 0$. Next, since

$g(0) = \mathbb{E}(1 - \eta(X))\mathbb{I}(\eta(X) > 0) = p^- - \mathbb{P}(\eta(X) = 0)$, we have that $g(0) > \alpha p^-$ if $\alpha < 1 - \frac{\mathbb{P}(\eta(X)=0)}{p^-}$, which means $c(\alpha) > 0$ when $\alpha < 1 - \frac{\mathbb{P}(\eta(X)=0)}{p^-}$. This completes proof of claim (i).

Next, we prove claim (ii). We first verify claim (ii) holds for $\alpha = 0$ and $\alpha = 1$. When $\alpha = 0$, it's easy to see $\mathbb{E}(1 - \eta(X))\mathbb{I}(\eta(X) \geq c(\alpha)) \geq 0$. Moreover, by using definition of $c(0)$ and right-continuity of $g(c)$ we have that $g(c(0)) \leq 0$. Note $g(c) \geq 0$ for any $c \geq 0$, we obtain $g(c(0)) = 0$. When $\alpha = 1$, we must have $c(\alpha) = 0$. Hence $g(0) = \mathbb{E}(1 - \eta(X))\mathbb{I}(\eta(X) > 0) \leq \mathbb{E}(1 - \eta(X)) = p^-$ and $\mathbb{E}(1 - \eta(X))\mathbb{I}(\eta(X) \geq 0) = \mathbb{E}(1 - \eta(X)) = p^-$. This verifies the proof of claim (ii) at $\alpha = 0$ and $\alpha = 1$.

Then, we show claim (ii) holds for $\alpha \in (0, 1)$. It is easy to show that $g(c)$ is right-continuous. By definition of $c(\alpha)$, we have that

$$g(c) \leq \alpha p^- \text{ for any } c \geq c(\alpha),$$

$$g(c) > \alpha p^- \text{ for any } c < c(\alpha).$$

Letting $c \to c(\alpha)$ with $c > c(\alpha)$ in the first equation and using the right continuity of $g(c)$, we obtain that $\alpha p^- \geq \lim_{c \to c(\alpha)+} g(c) = g(c(\alpha))$. Moreover, letting $c \to c(\alpha)$ with $c < c(\alpha)$ in the second equation, we obtain that $\alpha p^- \leq \lim_{c \to c(\alpha)-} g(c) = \mathbb{E}(1 - \eta(X))\mathbb{I}(\eta(X) \geq c(\alpha))$. This proves (ii).

Next, we prove (iii). We first show $c(\alpha)$ is a non-increasing and right-continuous function over $[0, 1)$. Nonincreasingness of $c(\alpha)$ follows easily from the definition of $c(\alpha)$ (we have $c(\alpha) \leq c(\tilde{\alpha})$ for any $\tilde{\alpha} < \alpha$ since $g(c(\tilde{\alpha})) \leq \tilde{\alpha} p^- < \alpha p^-$). Next, we prove that $c(\alpha)$ is right-continuous by contradiction. To this end, suppose that $c(\tilde{\alpha}) \to \bar{c} < c(\alpha)$ as $\tilde{\alpha} \to \alpha$ with $\tilde{\alpha} > \alpha$. By (ii), we have that $g(c(\tilde{\alpha})) \leq \tilde{\alpha} p^-$. Let $\tilde{\alpha} \to \alpha$, we obtain that $c(\tilde{\alpha}) \to \bar{c}$, and thus $g(\bar{c}) = \mathbb{E}(1 - \eta(X))\mathbb{I}(\eta(X) > \bar{c}) \leq \alpha p^-$. On the other hand, since $\bar{c} < c(\alpha)$, by definition of $c(\alpha)$ we must have $g(\bar{c}) > \alpha p^-$. This is a contradiction, and $c(\alpha)$ must be right-continuous over $[0, 1)$.

Finally, we prove the second claim in (iii). For one direction, we suppose that $c(\alpha)$ is discontinuous at $\alpha$, and we shall prove that there must exist $c > c(\alpha)$ such that $\mathbb{P}(c(\alpha) < \eta(X) < c) = 0$.

To this end, since $c(\alpha)$ is right-continuous, we must have that $c(\alpha)$ is not left continuous. By monotomicity of $c(\alpha)$, this means there exists $\bar{c}$, such that $c(\tilde{\alpha}) \to \bar{c}$ as $\tilde{\alpha} \to \alpha$ with $\tilde{\alpha} < \alpha$ and $c(\alpha) < \bar{c} \leq c(\tilde{\alpha})$. By using (ii), we have that $g(c(\alpha)) \leq \alpha p^-$. Moreover, since $c(\alpha) \leq \bar{c} < c(\tilde{\alpha})$, we must have $g(c(\alpha)) > \tilde{\alpha} p^-$, which implies that $g(c(\alpha)) \geq \alpha p^-$. Together, we must have $g(c(\alpha)) = \alpha p^-$. Next, again using (ii), we have $\mathbb{E}(1 - \eta(X))\mathbb{I}(\eta(X) \geq c(\tilde{\alpha})) \geq \tilde{\alpha} p^-$. Letting $\tilde{\alpha} \to \alpha$, this implies that $\mathbb{E}(1 - \eta(X))\mathbb{I}(\eta(X) \geq \bar{c}) \geq \lim_{\tilde{\alpha} \to \alpha} \mathbb{E}(1 - \eta(X))\mathbb{I}(\eta(X) \geq c(\tilde{\alpha})) \geq \lim_{\tilde{\alpha} \to \alpha} \tilde{\alpha} p^- = \alpha p^-$. Combining this with the fact that $g(c(\alpha)) = \alpha p^-$, we obtain that

$$\mathbb{E}((1 - \eta(X))\mathbb{I}(c(\alpha) < \eta(X) < \bar{c})) = g(c(\alpha)) - \mathbb{E}(1 - \eta(X))\mathbb{I}(\eta(X) \geq \bar{c}) \leq \alpha p^- - \alpha p^- = 0,$$

which implies that $\mathbb{E}((1 - \eta(X))\mathbb{I}(c(\alpha) < \eta(X) < \bar{c})) = 0$. This further implies that $\mathbb{P}(c(\alpha) < \eta(X) < \bar{c}) = 0$.

Conversely, we show that if $g(c(\alpha)) = \alpha p^-$ and $\mathbb{P}(c(\alpha) < \eta(X) < c) = 0$ for some $c > c(\alpha)$, then $c(\alpha)$ is not left-continuous. To this end, suppose that $c(\tilde{\alpha}) \to c(\alpha)$ as $\tilde{\alpha} \to \alpha$ with $\tilde{\alpha} < \alpha$ and $c(\tilde{\alpha}) \geq c(\alpha)$. Then, there must exist $\tilde{\alpha}$ such that $c(\tilde{\alpha}) < c$. By (ii), we have $g(c(\tilde{\alpha})) \leq \tilde{\alpha} p^-$. Using this with $\mathbb{P}(c(\alpha) < \eta(X) < c) = 0$, we obtain that $g(c(\alpha)) \leq \tilde{\alpha} p^-$, which contradicts with $g(c(\alpha)) = \alpha p^- > \tilde{\alpha} p^-$. This proves the discontinuity of $c(\alpha)$.

**Lemma S4.** *Denote* $\alpha_w = FPR(h_w^\star)$ *for any* $w \in (0, 1)$ *such that* $FPR(h_w^\star) \in (0, \bar{\alpha})$. *Then* $\alpha_w \leq w$ *and* $s(\alpha_w) = TPR(h_w^\star)$.

**Proof of Lemma S4.** We first show that $\alpha_w \leq w$. By using (S6), we get

$$\alpha_w p^- = \mathbb{E}(1 - \eta(X))(\mathbb{I}(\eta(X) > w) + \mathbb{I}(X \in \mathcal{N}_w)) \geq \mathbb{E}(1 - \eta(X))\mathbb{I}(\eta(X) > w), \qquad \text{(S16)}$$

which implies that $c(\alpha_w) \leq w$ by definition of $c(\alpha_w)$. Next, we consider two cases: (i) $c(\alpha_w) < w$; (ii) $c(\alpha_w) = w$.

For the first case that $c(\alpha_w) < w$, since $\{x : \eta(x) > w\} \cup \mathcal{N}_w \subseteq \{x : \eta(x) > c(\alpha_w)\}$, we have that

$$\alpha_w p^- = \mathbb{E}(1 - \eta(X))(\mathbb{I}(\eta(X) > w) + \mathbb{I}(X \in \mathcal{N}_w)) \leq \mathbb{E}(1 - \eta(X))\mathbb{I}(\eta(X) > c(\alpha_w)) \leq \alpha_w p^-,$$

where the last inequality uses part (ii) of Lemma S3. This implies that

$$\mathbb{E}(1 - \eta(X))(\mathbb{I}(\eta(X) > w) + \mathbb{I}(X \in \mathcal{N}_w)) = \mathbb{E}(1 - \eta(X))\mathbb{I}(\eta(X) > c(\alpha_w)) = \alpha_w p^-. \tag{S17}$$

Since $c(\alpha_w) < w$, we have that $\{x : \eta(x) > c(\alpha_w)\} \setminus \{x : h_w(x) = 1\} \subseteq \{x : c(\alpha_w) < \eta(x) \leq w\}$. Note $\mathbb{E}(1 - \eta(X))\mathbb{I}(\{x : \eta(x) > c(\alpha_w)\} \setminus \{x : h_w(x) = 1\}) = 0$ by using (S17). This further implies $\mathbb{P}(X \in \{x : \eta(x) > c(\alpha_w)\} \setminus \{x : h_w(x) = 1\}) = 0$ since $w < 1$. Therefore we must have $\{x : h_w(x) = 1\} = \{x : \eta(x) > w\} \cup \mathcal{N}_w = \{x : \eta(x) > c(\alpha_w)\}$ up to a $\mathbb{P}_X$ null set. Moreover, by using (7) in Theorem 2, we have

$$\mathbb{P}(X \in \mathcal{N}_{\alpha_w}) = \frac{\mathbb{E}(1 - \eta(X))(\alpha_w - \mathbb{I}(\eta(X) > c(\alpha_w)))}{1 - c(\alpha_w)} = 0.$$

Therefore, we get

$$\mathrm{TPR}(h_w^\star) = \frac{\mathbb{E}\eta(X)\mathbb{I}(h_w^\star = 1)}{p^+} = \frac{\mathbb{E}\eta(X)\mathbb{I}(\eta(X) > c(\alpha_w))}{p^+} = \frac{\mathbb{E}\eta(X)\mathbb{I}(X \in R_{\alpha_w}^\star)}{p^+} = s(\alpha_w).$$

For the second case when $c(\alpha_w) = w \in (0, 1)$, using (S16), we obtain that

$$\mathbb{P}(X \in \mathcal{N}_w) = \frac{\mathbb{E}(1 - \eta(X))(\alpha_w - \mathbb{I}(\eta(X) > w))}{1 - w}.$$

Again by using (7) in Theorem 2, we have that

$$\mathbb{P}(X \in \mathcal{N}_{\alpha_w}) = \frac{\mathbb{E}(1 - \eta(X))(\alpha_w - \mathbb{I}(\eta(X) > c(\alpha_w)))}{1 - c(\alpha_w)} = \mathbb{P}(X \in \mathcal{N}_w),$$

which further implies that

$$
\begin{aligned}
\text{TPR}(h_w^\star) &= \frac{\mathbb{E}\eta(X)(\mathbb{I}(\eta(X) > w) + \mathbb{I}(X \in \mathcal{N}_w))}{p^+} \\
&= \frac{\mathbb{E}\eta(X)(\mathbb{I}(\eta(X) > c(\alpha_w)) + \mathbb{I}(X \in \mathcal{N}_{\alpha_w}))}{p^+} = s(\alpha_w).
\end{aligned}
$$

Therefore we show $s(\alpha_w) = \text{TPR}(h_w^\star)$. This completes the proof of Lemma S4.

**Lemma S5.** *Let $S(\mathcal{H}_a) = \{(FPR(h), TPR(h)) : h \in \mathcal{H}_a\}$ and $\mathcal{I}$ be the union of all open intervals over which $s(\alpha)$ is linear. Then for any $S \subseteq S(\mathcal{H}_a)$ such that $\{(\alpha, s(\alpha)) \mid \alpha \in [0, 1 - \mathbb{P}(\eta(X) = 0)/p^-] \setminus \mathcal{I}\} \subseteq cl(S)$, we have*

$$
PF(\overline{conv}(S)) = \gamma(\mathcal{H}_a). \tag{S18}
$$

**Proof of Lemma S5.** Let $\bar{\alpha} = 1 - \mathbb{P}(\eta(X) = 0)/p^-$. Denote

$$
\bar{s}(\alpha) = \begin{cases} s(\alpha), & \text{if } \alpha \in [0, \bar{\alpha}], \\ 1, & \text{if } \alpha \in (\bar{\alpha}, 1], \\ -\infty, & \text{otherwise.} \end{cases}
$$

We first prove that $\bar{s}(\alpha)$ is concave over $[0, 1]$. To this end, we define

$$
d_1(\alpha) = \begin{cases} s(\alpha), & \text{if } \alpha \in [0, \bar{\alpha}] \\ s(\bar{\alpha}) + s'_-(\bar{\alpha})(\alpha - \bar{\alpha}), & \text{if } \alpha \in (\bar{\alpha}, 1] \\ -\infty, & \text{otherwise} \end{cases} \quad \text{and} \quad d_2(\alpha) = \begin{cases} 1, & \text{if } \alpha \in [0, 1] \\ -\infty, & \text{otherwise} \end{cases}
$$

where $s'_-(\bar{\alpha})$ is defined in (S89). Since $d_1(\alpha)$ linearly extends $s(\alpha)$ over $[\bar{\alpha}, 1]$ with slope $s'_-(\bar{\alpha})$, we have $d_1(\alpha)$ is concave by the fact that $s(\alpha)$ is concave over $[0, \bar{\alpha}]$. Also note $s(\bar{\alpha}) = 1$ and $s'_-(\bar{\alpha}) \geq 0$, we have that $d_1(\alpha) \geq d_2(\alpha)$ when $\alpha \in (\bar{\alpha}, 1]$. Using the facts that $d_2(\alpha)$ is concave and $\bar{s}(\alpha) = \min(d_1(\alpha), d_2(\alpha))$ we have that $\bar{s}(\alpha)$ is also concave, because the minimum of two concave functions is still concave.

Next, for any $S \subseteq S(\mathcal{H}_a)$ such that $\{(\alpha, s(\alpha)) \mid \alpha \in [0, \bar{\alpha}] \setminus \mathcal{I}\} \subseteq \mathrm{cl}(S)$, we shall show that $\overline{\mathrm{conv}}(S) \subseteq \mathrm{hyp}(\bar{s})$, where $\mathrm{hyp}(\bar{s}) = \{(\alpha, \mu) : \mu \leq \bar{s}(\alpha)\}$ is the hypograph of $\bar{s}(\alpha)$. Toward this end, we first show that $S \subseteq \mathrm{hyp}(\bar{s})$. For any $(u, v) \in S$, if $u \in [0, \bar{\alpha}]$, by using definition of optimal ROC curve $s(\alpha)$, we have that $v \leq s(u) = \bar{s}(u)$, implying that $(u, v) \in \mathrm{hyp}(\bar{s})$. If $u \in (\bar{\alpha}, 1]$, we still have that $v \leq s(u) = \bar{s}(u)$ since $\bar{s}(u) = 1$. Again we have $(u, v) \in \mathrm{hyp}(\bar{s})$ in this case. This finishes the proof of $S \subseteq \mathrm{hyp}(\bar{s})$.

Next, since $\bar{s}(\alpha)$ is a continuous function, we have that $\mathrm{hyp}(\bar{s})$ is a closed set. Therefore, we have $\mathrm{cl}(S) \subseteq \mathrm{cl}(\mathrm{hyp}(\bar{s})) = \mathrm{hyp}(\bar{s})$. Moreover, since $\bar{s}(\alpha)$ is concave, we have that $\mathrm{hyp}(\bar{s})$ is convex, which further implies $\mathrm{conv}(\mathrm{cl}(S)) \subseteq \mathrm{hyp}(\bar{s})$. Again by using the closedness of $\mathrm{hyp}(\bar{s})$, we obtain that $\overline{\mathrm{conv}}(S) \subseteq \overline{\mathrm{conv}}(\mathrm{cl}(S)) \subseteq \mathrm{hyp}(\bar{s})$. This completes the proof of $\overline{\mathrm{conv}}(S) \subseteq \mathrm{hyp}(\bar{s})$.

Now we are ready to prove our main results. We first show that $\mathrm{conv}(\mathrm{cl}(S)) \subseteq \overline{\mathrm{conv}}(S)$ for any set $S$. By definition of convex hull, we have that $S \subseteq \mathrm{conv}(S)$, which implies that $\mathrm{cl}(S) \subseteq \overline{\mathrm{conv}}(S)$. Since $\overline{\mathrm{conv}}(S)$ is convex, we obatin that $\mathrm{conv}(\mathrm{cl}(S)) \subseteq \overline{\mathrm{conv}}(S)$.

Next, since $\{(\alpha, s(\alpha)) \mid \alpha \in [0, \bar{\alpha}] \setminus \mathcal{I}\} \subseteq \mathrm{cl}(S)$, we have that $\gamma(\mathcal{H}_a) \subseteq \mathrm{conv}(\mathrm{cl}(S))$ by using the definition of convex hull. Also note that $\gamma(\mathcal{H}_a) = \mathrm{PF}(\mathrm{hyp}(\bar{s}))$. Hence, we have that

$$\mathrm{PF}(\mathrm{hyp}(\bar{s})) = \gamma(\mathcal{H}_a) \subseteq \mathrm{conv}(\mathrm{cl}(S)) \subseteq \overline{\mathrm{conv}}(S) \subseteq \mathrm{hyp}(\bar{s}). \tag{S19}$$

By using Lemma S1, we obtain that $\mathrm{PF}(\overline{\mathrm{conv}}(S)) = \mathrm{PF}(\mathrm{hyp}(\bar{s})) = \gamma(\mathcal{H}_a)$. This completes the proof of Lemma S5.

**Lemma S6.** *For $x \in \mathbb{R}^p$ and $w \in [0, 1]$, denote*

$$\Gamma_w(x) = \arg\min_{a \in \mathbb{R}} (1 - w)\eta(x)V(a) + w(1 - \eta(x))V(-a)$$

*and $\mathcal{F}_w^\star = \{f \mid f(x) \in \Gamma_w(x) \text{ for any } x \in \mathbb{R}^p\}$. Assume $\Gamma_w(x)$ is nonempty for $x \in \mathbb{R}^p$ and $w \in [0, 1]$.*

*Then for any solution $f$ to the following problem,*

$$\underset{f \in \mathcal{F}_a}{\text{minimize}} \ \mathbb{E}\{((1-w)\mathbb{I}(Y=1) + w\mathbb{I}(Y=-1))V(Yf(X))\} \tag{S20}$$

*there must exists $f' \in \mathcal{F}_w^{\star}$, such that $\mathbb{P}(f(X) \neq f'(X)) = 0$.*

**Proof of Lemma S6.** We show this proposition by contradiction. Suppose there exists a solution $f$ to (S20) such that for any $f' \in \mathcal{F}_w^{\star}$, $\mathbb{P}(f(X) \neq f'(X)) > 0$. Denote $B = \{x \in \mathbb{R}^p : f(x) \notin \Gamma_w(x)\}$, by definition of $\Gamma_w(x)$, we have that $\mathbb{P}(X \in B) > 0$. Note

$$\mathbb{E}\{((1-w)\mathbb{I}(Y=1) + w\mathbb{I}(Y=-1))V(Yf(X))\}$$

$$= \ \mathbb{E}\{(1-w)\eta(X)V(f(X)) + w(1-\eta(X))V(-f(X))\}$$

$$= \ \underbrace{\mathbb{E}\{(1-w)\eta(X)V(f(X)) + w(1-\eta(X))V(-f(X))\}\mathbb{I}(X \in B)}_{\text{Part I}}$$

$$+ \ \underbrace{\mathbb{E}\{(1-w)\eta(X)V(f(X)) + w(1-\eta(X))V(-f(X))\}\mathbb{I}(X \notin B)}_{\text{Part II}}$$

For part I, note $\mathbb{P}(X \in B) > 0$ and for any $x \in B$,

$$\{(1-w)\eta(x)V(f(x)) + w(1-\eta(x))V(-f(x))\} > \underset{a \in \mathbb{R}}{\min}(1-w)\eta(x)V(a) + w(1-\eta(x))V(-a)$$

we have that for any $f' \in \mathcal{F}_w^{\star}$,

$$\text{Part I} \ > \ \mathbb{E}\{\underset{a \in \mathbb{R}}{\min}(1-w)\eta(x)V(a) + w(1-\eta(x))V(-a)\}\mathbb{I}(X \in B)$$

$$= \ \mathbb{E}\{(1-w)\eta(X)V(f'(X)) + w(1-\eta(X))V(-f'(X))\}\mathbb{I}(X \in B)$$

For part II, it's easy to see for any $f' \in \mathcal{F}_w^{\star}$,

$$\mathbb{E}\{(1-w)\eta(X)V(f(X)) + w(1-\eta(X))V(-f(X))\}\mathbb{I}(X \notin B)$$

$$= \ \mathbb{E}\{(1-w)\eta(X)V(f'(X)) + w(1-\eta(X))V(-f'(X))\}\mathbb{I}(X \notin B)$$

Hence we obtain that

$$\mathbb{E}\left\{(1-w)\eta(X)V(f(X)) + w(1-\eta(X))V(-f(X))\right\}$$

$$> \; \mathbb{E}\left\{(1-w)\eta(X)V(f'(X)) + w(1-\eta(X))V(-f'(X))\right\}$$

for any $f' \in \mathcal{F}_w^\star$, which contradiction the fact that $f$ is a solution to (S20). This completes the proof.

**Lemma S7.** *Let $V(\cdot)$ be a differentiable, strictly decreasing, proper and strictly convex loss. Then $V(\cdot)$ is classification-calibrated.*

**Proof of Lemma S7.** Denote $G(\eta, a) = \eta V(a) + (1-\eta)V(-a)$. Since $V(\cdot)$ is proper and strictly convex, we have that the minimizer $a^\star(\eta)$ of $G(\eta, \cdot)$ is unique for any fixed $\eta \in [0,1]$. Recall $H_V(\eta) = \inf_{a \in \mathbb{R}} G(\eta, \alpha)$ and $H_V^-(\eta) = \inf_{\alpha : \alpha(\eta-1/2) \leq 0} G(\eta, \alpha)$. In order to prove $H_V(\eta) < H_V^-(\eta)$ for any $\eta \neq 1/2$, it suffices to show $a^\star(\eta)(\eta - 1/2) > 0$ for any $\eta \neq 1/2$.

Note $a^\star(\eta)$ is also the solution to (S56) at $w = 1/2$. By the proof of Theorem 5, we have $a^\star(\eta)$ is a strictly increasing function, meaning that $(a^\star(\eta) - a^\star(1/2))(\eta - 1/2) > 0$ for any $\eta \neq 1/2$. Moreover, since $G(1/2, a) = (V(a) + V(-a))/2$ is strictly convex and symmetric around 0, we obtain that $a^\star(1/2) = 0$, which further implies $a^\star(\eta)(\eta - 1/2) > 0$ for any $\eta \neq 1/2$. This completes the proof of Lemma S7.

# G    Proofs

**Proof of Theorem 1.** We first prove that $s(\alpha)$, as defined in (2), is continuous, strictly increasing, and concave. For continuity, we start by showing $s(\alpha)$ is continuous on $\left(0, 1 - \frac{\mathbb{P}(\eta(X)=0)}{p^-}\right]$. Let $R_\alpha^\star$ be the solution to (S41) at $\alpha$. From the proof of Theorem 2, we know that $R_\alpha^\star = \{x : \eta(x) > c(\alpha)\} \cup \mathcal{N}_\alpha$ and the constraint is tight at the solution when $\alpha \in \left(0, 1 - \frac{\mathbb{P}(\eta(X)=0)}{p^-}\right)$, which implies that

$$\mathbb{E}(1-\eta(X))(\mathbb{I}(X \in R_\alpha^\star) - \alpha) = 0, \text{ or } \mathbb{E}(1-\eta(X))\mathbb{I}(X \in R_\alpha^\star) = \alpha p^- . \tag{S21}$$

Moreover, it follows from (S42) and (S43) that when $\alpha = 1 - \frac{\mathbb{P}(\eta(X)=0)}{p^-}$, we have $c(\alpha) = 0$ and $\mathbb{P}(\mathcal{N}_\alpha) = 0$, which further implies $R_\alpha^\star = \{x : \eta(x) > c(\alpha)\}$ and (S21) also holds when $\alpha = 1 - \frac{\mathbb{P}(\eta(X)=0)}{p^-}$. Using (S21), we have that for any $\alpha \in \left(0, 1 - \frac{\mathbb{P}(\eta(X)=0)}{p^-}\right]$

$$s(\alpha) = \text{TPR}(h_\alpha^\star) = \frac{\mathbb{E}(\eta(X)\mathbb{I}(X \in R_\alpha^\star))}{p^+} = \frac{\mathbb{P}(X \in R_\alpha^\star) - \alpha p^-}{p^+} . \tag{S22}$$

Hence, it suffices to show continuity of $\mathbb{P}(X \in R_\alpha^\star)$ as a function of $\alpha$. Using (S21), note that

$$
\begin{aligned}
(\alpha_2 - \alpha_1)p^- &= \mathbb{E}(1 - \eta(X))(\mathbb{I}(X \in R_{\alpha_2}^\star) - \mathbb{I}(X \in R_{\alpha_1}^\star)) \\
&= \mathbb{E}(1 - \eta(X))\mathbb{I}(X \in R_{\alpha_2}^\star \setminus R_{\alpha_1}^\star) - \mathbb{E}(1 - \eta(X))\mathbb{I}(X \in R_{\alpha_1}^\star \setminus R_{\alpha_2}^\star) \\
&\geq (1 - c(\alpha_1))\mathbb{P}(X \in R_{\alpha_2}^\star \setminus R_{\alpha_1}^\star) - (1 - c(\alpha_1))\mathbb{P}(X \in R_{\alpha_1}^\star \setminus R_{\alpha_2}^\star) \\
&= (1 - c(\alpha_1))(\mathbb{P}(X \in R_{\alpha_2}^\star) - \mathbb{P}(X \in R_{\alpha_1}^\star))
\end{aligned}
\tag{S23}
$$

for any $\alpha_1, \alpha_2 \in \left(0, 1 - \frac{\mathbb{P}(\eta(X)=0)}{p^-}\right]$ with $\alpha_1 < \alpha_2$. This, together with the fact that $c(\alpha) < 1$ when $\alpha \in (0,1]$ (c.f. (i) of Lemma S3), implies that $\mathbb{P}(X \in R_\alpha^\star)$ is continuous in $\alpha$. This completes the proof of continuity over $\left(0, 1 - \frac{\mathbb{P}(\eta(X)=0)}{p^-}\right]$.

Next, we show $s(\alpha)$ is right-continuous at $\alpha = 0$. We first use Theorem 2 to show that there exists solutions $R_\alpha^\star$ to (5) such that we have $R_\alpha^\star$ is monotone in $\alpha$, that is, $R_{\alpha_1}^\star \subseteq R_{\alpha_2}^\star$ for any $\alpha_1 < \alpha_2$. To this end, by the proof of Theorem 2, we know that we can choose

$$R_0^\star = \{x : \eta(x) = 1\} \text{ and } R_\alpha^\star = \{\eta(x) > c(\alpha)\} \cup \mathcal{N}_\alpha \tag{S24}$$

where $N_\alpha \subseteq \{x : \eta(x) = c(\alpha)\}$ with $\mathbb{P}(\mathcal{N}_\alpha) = \frac{\mathbb{E}(1-\eta(X))(\alpha - \mathbb{I}(\eta(X)>c(\alpha)))}{1-c(\alpha)}$ when $\alpha \in \left(0, 1 - \frac{\mathbb{P}(\eta(X)=0)}{p^-}\right)$. Clearly, we can always choose $R_\alpha^\star$ so that $R_0^\star \subseteq R_\alpha^\star$, because $R_\alpha^\star \cup R_0^\star$ would have the same constraint with no smaller objective function for problem (5). Moreover, by definition of $c(\alpha)$, it is easy to see that it is nonincreasing. Then, for any $0 < \alpha_1 < \alpha_2 < 1 - \frac{\mathbb{P}(\eta(X)=0)}{p^-}$, we have that $R_{\alpha_1}^\star \subseteq R_{\alpha_2}^\star$ if $c(\alpha_1) > c(\alpha_2)$, and we can choose $R_{\alpha_2}^\star$ so that $R_{\alpha_1}^\star \subseteq R_{\alpha_2}^\star$ if $c(\alpha_1) = c(\alpha_2)$, where the latter claim is

because $\mathbb{P}(X \in \mathcal{N}_{\alpha_2}) \geq \mathbb{P}(X \in \mathcal{N}_{\alpha_1})$ when $c(\alpha_1) = c(\alpha_2)$. Therefore, we can define $\bar{R} = \lim_{\alpha \to 0} R_\alpha^\star$, which satisfies $R_0^\star \subseteq \bar{R}$. Letting $\alpha \to 0$ on both sides of the constraint of (5), we obtain that

$$0 \leq \mathbb{E}(1 - \eta(X))\mathbb{I}(X \in \bar{R}) = \lim_{\alpha \to 0} \mathbb{E}(1 - \eta(X))\mathbb{I}(X \in R_\alpha^\star) \leq \lim_{\alpha \to 0} \alpha p^- = 0, \qquad \text{(S25)}$$

which implies that $\mathbb{E}(1 - \eta(X))\mathbb{I}(X \in \bar{R}) = 0$. Therefore, $\bar{R} \subseteq \{x : \eta(x) = 1\} = R_0^\star$. Combining, we get $\bar{R} = R_0^\star$. Hence,

$$\lim_{\alpha \to 0} s(\alpha) = \frac{\lim_{\alpha \to 0} \mathbb{E}\eta(X)\mathbb{I}(X \in R_\alpha^\star)}{p^+} = \frac{\mathbb{E}\eta(X)\mathbb{I}(X \in R_0^\star)}{p^+} = s(0). \qquad \text{(S26)}$$

This completes the proof of continuity at $\alpha = 0$. In sum, we have that $s(\alpha)$ is a continuous function when $\alpha \in \left[0, 1 - \frac{\mathbb{P}(\eta(X)=0)}{p^-}\right]$.

Next, we show that $s(\alpha)$ is concave. Similar to (S23), we also have that

$$\begin{aligned}
(\alpha_2 - \alpha_1)p^- &= \mathbb{E}(1 - \eta(X))(\mathbb{I}(X \in R_{\alpha_2}^\star) - \mathbb{I}(X \in R_{\alpha_1}^\star)) \\
&= \mathbb{E}(1 - \eta(X))\mathbb{I}(X \in R_{\alpha_2}^\star \setminus R_{\alpha_1}^\star) - \mathbb{E}(1 - \eta(X))\mathbb{I}(X \in R_{\alpha_1}^\star \setminus R_{\alpha_2}^\star) \\
&\leq (1 - c(\alpha_2))\mathbb{P}(X \in R_{\alpha_2}^\star \setminus R_{\alpha_1}^\star) - (1 - c(\alpha_2))\mathbb{P}(X \in R_{\alpha_1}^\star \setminus R_{\alpha_2}^\star) \\
&= (1 - c(\alpha_2))(\mathbb{P}(X \in R_{\alpha_2}^\star) - \mathbb{P}(X \in R_{\alpha_1}^\star)) \qquad \text{(S27)}
\end{aligned}$$

for any $\alpha_1, \alpha_2 \in \left(0, 1 - \frac{\mathbb{P}(\eta(X)=0)}{p^-}\right]$ with $\alpha_1 < \alpha_2$. Combining this with (S23), we have that

$$\frac{(\alpha_2 - \alpha_1)p^-}{1 - c(\alpha_2)} \leq \mathbb{P}(X \in R_{\alpha_2}^\star) - \mathbb{P}(X \in R_{\alpha_1}^\star) \leq \frac{(\alpha_2 - \alpha_1)p^-}{1 - c(\alpha_1)}, \qquad \text{(S28)}$$

which in turn implies that

$$\frac{p^- c(\alpha_2)}{p^+(1 - c(\alpha_2))}(\alpha_2 - \alpha_1) \leq s(\alpha_2) - s(\alpha_1) \leq \frac{p^- c(\alpha_1)}{p^+(1 - c(\alpha_1))}(\alpha_2 - \alpha_1). \qquad \text{(S29)}$$

for any $\alpha_1, \alpha_2 \in \left(0, 1 - \frac{\mathbb{P}(\eta(X)=0)}{p^-}\right]$ with $\alpha_1 < \alpha_2$. Using this, we obtain for any $0 < \alpha_1 < \alpha_2 \leq$

$1 - \frac{\mathbb{P}(\eta(X)=0)}{p^-}$, that

$$
\begin{aligned}
2s(\bar{\alpha}) - s(\alpha_1) - s(\alpha_2) &= s(\bar{\alpha}) - s(\alpha_1) - (s(\alpha_2) - s(\bar{\alpha})) \\
&\geq \frac{p^- c_{\bar{\alpha}}^\star}{p^+(1 - c_{\bar{\alpha}}^\star)}(\bar{\alpha} - \alpha_1) - \frac{p^- c_{\bar{\alpha}}^\star}{p^+(1 - c_{\bar{\alpha}}^\star)}(\alpha_2 - \bar{\alpha}) \\
&= \frac{p^- c_{\bar{\alpha}}^\star}{p^+(1 - c_{\bar{\alpha}}^\star)}\frac{\alpha_2 - \alpha_1}{2} - \frac{p^- c_{\bar{\alpha}}^\star}{p^+(1 - c_{\bar{\alpha}}^\star)}\frac{\alpha_2 - \alpha_1}{2} = 0 \, ,
\end{aligned}
$$

where $\bar{\alpha} = (\alpha_1 + \alpha_2)/2$. This proves the concavity of $s(\alpha)$.

We next prove that $s(\alpha)$ is strictly increasing. By definition, we have that $s(\alpha)$ must be non-decreasing. Moreover, from (S29), we can see that $s(\alpha)$ is a strictly increasing function of $\alpha$ over the interval $(0, 1 - \mathbb{P}(\eta(X) = 0)/p^-)$. Therefore, $s(\alpha)$ must be a strictly increasing function over $[0, 1 - \mathbb{P}(\eta(X) = 0)/p^-]$.

Finally, we prove (1). We first prove that

$$
\gamma(\mathcal{H}_a) \subseteq \{(\alpha, s(\alpha)) \mid \alpha \in [0, 1]\} \, , \tag{S30}
$$

for $s(\alpha)$ as defined in Theorem 2. Using (8) of Theorem 2, it suffices to show that, for any classifier $h$ satisfying $(\mathrm{FPR}(h), \mathrm{TPR}(h)) \in \gamma(\mathcal{H}_a)$, $h$ must be the solution to (5) with $\alpha = \mathrm{FPR}(h)$ and $\mathcal{H} = \mathcal{H}_a$. First, $h$ is feasible. Moreover, if $h$ is not a solution to (5), then there must exist $h' \in \mathcal{H}_a$, such that $\mathrm{TPR}(h') > \mathrm{TPR}(h)$ and $\mathrm{FPR}(h') \leq \mathrm{FPR}(h)$, which means that $h'$ dominates $h$. This contradicts with the fact that $h$ is on the pareto frontier (c.f. Definition 1). Therefore, $h$ must be a solution of (5) with $\alpha = \mathrm{FPR}(h)$ and $\mathcal{H} = \mathcal{H}_a$.

Now let $\tilde{\gamma} = \{(\alpha, s(\alpha)) \mid \alpha \in [0, 1]\}$. Since $s(\alpha)$ is continuous, we have that $\tilde{\gamma}$ is a closed set. By using Lemma S1, we know that $\gamma(\mathcal{H}_a) = PF(\tilde{\gamma})$, which is is the the pareto frontier of $\tilde{\gamma}$. Next we derive the pareto frontier of $\tilde{\gamma}$. First, it is easy to see that any point on $\{\alpha, s(\alpha) \mid \alpha \in (1 - \mathbb{P}(\eta(X) = 0)/p^-, 1]\}$ can be dominated by $(1 - \mathbb{P}(\eta(X) = 0)/p^-, 1)$. Moreover, no pairs in $\{(\alpha, s(\alpha)) \mid \alpha \in [0, 1 - \mathbb{P}(\eta(X) = 0)/p^-]\}$ could dominate each other since $s(\alpha)$ is strictly increasing. Consequently, we have that the pareto frontier of $\tilde{\gamma}$ is $s(\alpha)$ defined in (2).

**Proof of Corollary S1.** We start by showing that

$$\gamma(\mathcal{H}_a) = \{(\text{FPR}(\text{Sign}(\eta(x) - c(\alpha))), \text{TPR}(\text{Sign}(\eta(x) - c(\alpha)))) \mid 0 \le \alpha \le 1\} \quad \text{(S31)}$$

$$= \{(\text{FPR}(\text{Sign}(\eta(x) - c)), \text{TPR}(\text{Sign}(\eta(x) - c))) \mid 0 \le c \le 1\}, \quad \text{(S32)}$$

under (3). We first show that $\alpha = \text{FPR}(\text{Sign}(\eta(x) - c(\alpha)))$. Since $\eta(x) = p^+ f_+(x)/(p^+ f_+(x) + p^- f_-(x))$, for any $c \in [0, 1]$ we have that

$$\mathbb{P}(\eta(X) = c \mid Y = 1) = \mathbb{P}(\eta(X) = c \mid Y = -1) = 0, \quad \text{(S33)}$$

which implies

$$\mathbb{P}(\eta(X) = c) = p^+ \mathbb{P}(\eta(X) = c \mid Y = 1) + p^- \mathbb{P}(\eta(X) = c \mid Y = -1) = 0. \quad \text{(S34)}$$

By using (i) and (ii) in Lemma S3, we have that

$$\mathbb{E}(1 - \eta(X))\mathbb{I}(\eta(X) > c(\alpha)) = \mathbb{E}(1 - \eta(X))\mathbb{I}(\eta(X) \ge c(\alpha)) = \alpha p^-, \quad \text{(S35)}$$

for all $\alpha \in [0, 1]$, which can be rewritten as

$$\alpha = \frac{1}{p^-}\mathbb{E}(1 - \eta(X))\mathbb{I}(\eta(X) > c(\alpha)) = \mathbb{P}(\eta(X) > c(\alpha) \mid Y = -1) = \text{FPR}(\text{Sign}(\eta(x) - c(\alpha))) \quad \text{(S36)}$$

Next, we show $s(\alpha) = \text{TPR}(\text{Sign}(\eta(x) - c(\alpha)))$, When $\alpha \in (0, 1]$, $s(\alpha)$ can be represented by

$$\begin{aligned}
s(\alpha) &= \frac{\mathbb{P}(\eta(X) > c(\alpha)) - c(\alpha)\mathbb{E}\eta(X)\mathbb{I}(\eta(X) > c(\alpha)) - \alpha p^-}{p^+(1 - c(\alpha))} \\
&= \frac{\mathbb{P}(\eta(X) > c(\alpha)) - c(\alpha)\mathbb{E}\eta(X)\mathbb{I}(\eta(X) > c(\alpha)) - \mathbb{E}(1 - \eta(X))\mathbb{I}(\eta(X) > c(\alpha))}{p^+(1 - c(\alpha))} \\
&= \frac{(1 - c(\alpha))\mathbb{E}\eta(X)\mathbb{I}(\eta(X) > c(\alpha))}{p^+(1 - c(\alpha))} = \frac{1}{p^+}\mathbb{E}\eta(X)\mathbb{I}(\eta(X) > c(\alpha)) \\
&= \mathbb{P}(\eta(X) > c(\alpha) \mid Y = 1) = \text{TPR}(\text{Sign}(\eta(x) - c(\alpha))),
\end{aligned}$$

where we use the fact that $c(\alpha) < 1$ when $\alpha > 0$ (c.f. (i) in Lemma S3). When $\alpha = 0$, by using (S35) and the fact that $\mathbb{P}(\eta(X) = 1) = 0$, we obtain that

$$\mathbb{E}\eta(X)\mathbb{I}(\eta(X) > c(0)) = \mathbb{P}(\eta(X) > c(0)) = \mathbb{P}(c(0) < \eta(X) < 1) = 0\,. \tag{S37}$$

This combined with (2) implies

$$s(0) = 0 = \frac{1}{p^+}\mathbb{E}\eta(X)\mathbb{I}(\eta(X) > c(0)) = \mathbb{P}(\eta(X) > c(0) \mid Y = 1) = \text{TPR}(\text{Sign}(\eta(x) - c(0)))\,.$$

Therefore for all $0 \leq \alpha \leq 1$, we have that

$$s(\alpha) = \frac{1}{p^+}\mathbb{E}\eta(X)\mathbb{I}(\eta(X) > c(\alpha)) = \text{TPR}(\text{Sign}(\eta(x) - c(\alpha)))\,. \tag{S38}$$

This finishes the proof of (S31).

Next, we check (S32). For any $\alpha \in [0,1]$, using definition of $c(\alpha)$, we have that $c(\alpha) \in [0,1]$, which implies

$$\{(\text{FPR}(\text{Sign}(\eta(x) - c(\alpha))), \text{TPR}(\text{Sign}(\eta(x) - c(\alpha)))) \mid 0 \leq \alpha \leq 1\}$$
$$\subseteq \{(\text{FPR}(\text{Sign}(\eta(x) - c)), \text{TPR}(\text{Sign}(\eta(x) - c))) \mid 0 \leq c \leq 1\}\,.$$

Moreover, for any $c \in [0,1]$, let $\alpha = \text{FPR}(\text{Sign}(\eta(x) - c)) \in [0,1]$, by using (S36) we have that

$$\alpha = \text{FPR}(\text{Sign}(\eta(x) - c(\alpha))) = \frac{1}{p^-}\mathbb{E}(1 - \eta(X))\mathbb{I}(\eta(X) > c(\alpha)) = \frac{1}{p^-}\mathbb{E}(1 - \eta(X))\mathbb{I}(\eta(X) > c)\,.$$

Also note $\mathbb{P}(\eta(X) = c(\alpha)) = 0$, we obtain that $\mathbb{P}(c < \eta(X) \leq c(\alpha)) = 0$. This together with (S38) implies

$$\text{TPR}(\text{Sign}(\eta(x) - c(\alpha))) = \frac{1}{p^+}\mathbb{E}\eta(X)\mathbb{I}(\eta(X) > c(\alpha)) = \frac{1}{p^+}\mathbb{E}\eta(X)\mathbb{I}(\eta(X) > c) = \text{TPR}(\text{Sign}(\eta(x) - c))$$

Therefore we have that

$$\{(\text{FPR}(\text{Sign}(\eta(x) - c)), \text{TPR}(\text{Sign}(\eta(x) - c))) \mid 0 \le c \le 1\}$$

$$\subseteq \{(\text{FPR}(\text{Sign}(\eta(x) - c(\alpha))), \text{TPR}(\text{Sign}(\eta(x) - c(\alpha)))) \mid 0 \le \alpha \le 1\}.$$

This completes the proof of (S32).

Next, we derive the ROC curve defined in (4). Let $\alpha = \text{FPR}(h_\lambda)$ for any $\lambda \in [0, \infty]$. We first show that the range of $\alpha$ is $[0, 1]$. Again by using the fact that $\eta(x) = p^+ f_+(x)/(p^+ f_+(x) + p^- f_-(x))$, we obtain that

$$\alpha = \mathbb{P}\left(f_+(X)/f_-(X) > \lambda \mid Y = -1\right) = \mathbb{P}\left(\eta(X) > \frac{p^+ \lambda}{p^+ \lambda + p^-} \mid Y = -1\right). \qquad \text{(S39)}$$

Using this, we obtain that

$$\alpha \le \lim_{\lambda \to 0^+} \mathbb{P}\left(\eta(X) > \frac{p^+ \lambda}{p^+ \lambda + p^-} \mid Y = -1\right) = \mathbb{P}(\eta(X) > 0 \mid Y = -1) = 1,$$

and

$$\alpha \ge \lim_{\lambda \to \infty} \mathbb{P}\left(\eta(X) > \frac{p^+ \lambda}{p^+ \lambda + p^-} \mid Y = -1\right) = \mathbb{P}(\eta(X) = 1 \mid Y = -1) = 0.$$

Also note when $\lambda = 0$ and $\lambda = \infty$, we have that $\alpha = 1$ and $\alpha = 0$ respectively. Since by assumption $\mathbb{P}(\eta(X) = c) = 0$ for any $c \in [0, 1]$, we have that $\mathbb{P}(\eta(X) > c \mid Y = -1)$ is a continuous function for $c \in (0, 1)$. Therefore the range of $\alpha$ must be $[0, 1]$.

Now we are ready to compute $\text{TPR}(h_\lambda)$. Note that (S39) can reduce to

$$\mathbb{E}(1 - \eta(X))\left(\mathbb{I}\left(\eta(X) > \frac{p^+ \lambda}{p^+ \lambda + p^-}\right) - \alpha\right) = \mathbb{E}(1 - \eta(X))\left(\mathbb{I}\left(\eta(X) \ge \frac{p^+ \lambda}{p^+ \lambda + p^-}\right) - \alpha\right) = 0.$$

By using (ii) in Lemma S3, we have that $c(\alpha) \le \frac{p^+ \lambda}{p^+ \lambda + p^-} < 1$,

$$\mathbb{E}(1 - \eta(X))\left(\mathbb{I}\left(\eta(X) > c(\alpha)\right) - \alpha\right) = 0 \text{ and } \mathbb{P}\left(c(\alpha) < \eta(X) \le \frac{p^+ \lambda}{p^+ \lambda + p^-}\right) = 0. \qquad \text{(S40)}$$

This implies

$$
\begin{aligned}
\mathrm{TPR}(h_\lambda) &= \mathbb{P}(f_+(X) > \lambda f_-(X) \mid Y = 1) = \mathbb{P}\left(\eta(X) > \frac{p^+\lambda}{p^+\lambda + p^-} \mid Y = 1\right) \\
&= \frac{1}{p^+}\mathbb{E}\eta(X)\mathbb{I}\left(\eta(X) > \frac{p^+\lambda}{p^+\lambda + p^-}\right) = \frac{1}{p^+}\mathbb{E}\eta(X)\mathbb{I}(\eta(X) > c(\alpha)).
\end{aligned}
$$

When $\alpha \in (0, 1)$, this is exactly $s(\alpha)$. When $\alpha = 1$, we have $c(\alpha) = 0$. Therefore,

$$
\mathrm{TPR}(h_\lambda) = \frac{1}{p^+}\mathbb{E}\eta(X)\mathbb{I}(\eta(X) > 0) = 1 = s(1).
$$

When $\alpha = 0$, by using (S40) and the fact that $c(\alpha) < 1$, we have that $\mathbb{P}(c(\alpha) < \eta(X) < 1) = 0$. This implies that

$$
\mathrm{TPR}(h_\lambda) = \frac{1}{p^+}\mathbb{E}\eta(X)\mathbb{I}(\eta(X) = 1) = 0 = s(0).
$$

Therefore, we have that

$$
\{(\mathrm{FPR}(h_\lambda), \mathrm{TPR}(h_\lambda)) \mid \lambda \in [0, \infty]\} = \gamma(\mathcal{H}_a).
$$

Combining this with (S32), we complete the proof of Corollary S1.

**Proof of Theorem 2.** For any measurable classifier $h$, define $R_h = \{x : h(x) = 1\}$. By iterated expectation, we have

$$
\begin{aligned}
\mathrm{TPR}(h) &= \mathbb{P}(h(X) = 1 \mid Y = 1) = \frac{\mathbb{E}(\eta(X)\mathbb{I}(X \in R_h))}{p^+} \\
\mathrm{FPR}(h) &= \mathbb{P}(h(X) = 1 \mid Y = -1) = \frac{\mathbb{E}((1 - \eta(X))\mathbb{I}(X \in R_h))}{p^-}.
\end{aligned}
$$

Note that $\mathbb{E}(1 - \eta(X)) = 1 - \mathbb{E}(\mathbb{P}(Y = 1 \mid X)) = 1 - p^+ = p^-$. This implies that $\mathrm{FPR}(h) \le \alpha$ is

equivalent to $\mathbb{E}(1 - \eta(X))(\mathbb{I}(X \in R_h) - \alpha) \le 0$. Hence, we can rewrite (5) as

$$
\begin{aligned}
&\text{maximize}_{h \in \mathcal{H}} \quad \mathbb{E}\{\eta(X)\mathbb{I}(X \in R_h)\} \\
&\text{subject to} \quad \mathbb{E}(1 - \eta(X))(\mathbb{I}(X \in R_h) - \alpha) \le 0\,.
\end{aligned}
\tag{S41}
$$

Note that both the objective and constraint of the above problem depend on $h$ through $R_h$. Hence, we can view $R_h$ as the decision variable.

Next, we consider two cases: (i) $\alpha = 0$; (ii) $\alpha \in (0, 1]$. When $\alpha = 0$, for any feasible $R$ for (S41), we have that

$$
\mathbb{E}(1 - \eta(X))\mathbb{I}(X \in R) \le 0
$$

Since $(1 - \eta(X))\mathbb{I}(X \in R)$ is nonnegative, we have that $R \subseteq \{x : \eta(x) = 1\}$ up to $\mathbb{P}_X$ measure zero set. Hence, the objective function at $R$ is

$$
\mathbb{E}\eta(X)\mathbb{I}(X \in R) = \mathbb{P}(X \in R) \le \mathbb{P}(\eta(X) = 1)
$$

which implies that the optimal solution must be $R_\alpha^\star = \{x : \eta(x) = 1\}$ up to a $\mathbb{P}_X$-null set. Moreover, at the solution $R_\alpha^\star$, we have that

$$
\begin{aligned}
\text{TPR}^\star &= \mathbb{P}(X \in R_\alpha^\star \mid Y = 1) = \frac{\mathbb{E}(\eta(X)\mathbb{I}(\eta(X) = 1))}{p^+} = \frac{\mathbb{P}(\eta(X) = 1)}{p^+}\,, \\
\text{FPR}^\star &= \mathbb{P}(X \in R_\alpha^\star \mid Y = -1) = \frac{\mathbb{E}((1 - \eta(X))\mathbb{I}(\eta(X) = 1))}{p^-} = 0\,.
\end{aligned}
$$

This completes the proof of the first case.

Next, we consider the case $\alpha \in (0, 1]$. Define $g(c) = \mathbb{E}(1 - \eta(X))\mathbb{I}(\eta(X) > c)$ for $c \in [0, 1]$. It is easy to show that $g(c)$ is a right-continuous and non-increasing function on $[0, 1]$. Let $c(\alpha) = \inf\{c \in [0, p^\star] : g(c) \le \alpha p^-\}$, which is well-defined because $g(p^\star) = 0 < \alpha p^-$. Moreover, it is easy to verify that $g(c(\alpha)) \le \alpha p^-$ and $g(c) > \alpha p^-$ for any $c < c(\alpha)$. By Lemma S3, we also have $c(\alpha) < 1$

when $\alpha \in (0,1]$. Moreover, since

$$g(0) = \mathbb{E}(1 - \eta(X))\mathbb{I}(\eta(X) \geq 0) - \mathbb{P}(\eta(X) = 0) = p^- - \mathbb{P}(\eta(X) = 0),$$

we have that $g(0) \leq \alpha p^-$ if and only if $\alpha \geq 1 - \frac{\mathbb{P}(\eta(X)=0)}{p^-}$. By the right-continuity and monotonicity of $g(c)$, we have that

$$c(\alpha) = 0 \text{ if and only if } \alpha \geq 1 - \frac{\mathbb{P}(\eta(X) = 0)}{p^-}. \tag{S42}$$

Next, we show that any solution to (5), up to a $\mathbb{P}_X$-null set, must be in the following form

$$R_\alpha^\star = \{x : \eta(x) > c(\alpha)\} \cup \mathcal{N}_\alpha$$

with $\mathcal{N}_\alpha \subseteq \{x : \eta(x) = c(\alpha)\}$ and its probability satisfying

$$\begin{aligned}
\mathbb{E}(1 - \eta(X))(\mathbb{I}(\eta(X) > c(\alpha)) - \alpha) + (1 - c(\alpha))\mathbb{P}(X \in \mathcal{N}_\alpha) \leq 0 &\quad \text{if } \alpha \in \left(1 - \frac{\mathbb{P}(\eta(X)=0)}{p^-}, 1\right], \\
\mathbb{E}(1 - \eta(X))(\mathbb{I}(\eta(X) > c(\alpha)) - \alpha) + (1 - c(\alpha))\mathbb{P}(X \in \mathcal{N}_\alpha) = 0 &\quad \text{if } \alpha \in \left(0, 1 - \frac{\mathbb{P}(\eta(X)=0)}{p^-}\right].
\end{aligned} \tag{S43}$$

Toward this end, we first verify the existence of $\mathcal{N}_\alpha$. The existence of $\mathcal{N}_\alpha$ for the first case in (S43) can be ensured by setting $\mathcal{N}_\alpha = \emptyset$. For the second case, the existence is ensured by noting that the left hand side equals $g(c(\alpha)) - \alpha p^- \leq 0$ when $\mathcal{N}_\alpha = \emptyset$, and it equals $\mathbb{E}(1 - \eta(X))(\mathbb{I}(\eta(X) \geq c(\alpha)) - \alpha) = \mathbb{E}(1 - \eta(X))\mathbb{I}(\eta(X) \geq c(\alpha)) - \alpha p^- \geq 0$ when $\mathcal{N}_\alpha = \{x : \eta(x) = c(\alpha)\}$ (c.f. (ii) of Lemma S3), and the fact that $X$ is a continuous random variable.

We next consider the scenario $\alpha \in \left(0, 1 - \frac{\mathbb{P}(\eta(X)=0)}{p^-}\right]$. In this case, $R_\alpha^\star$ satisfies the constraint, because

$$\mathbb{E}(1 - \eta(X))(\mathbb{I}(X \in R_\alpha^\star) - \alpha) = \mathbb{E}(1 - \eta(X))(\mathbb{I}(\eta(X) > c(\alpha)) - \alpha) + (1 - c(\alpha))\mathbb{P}(X \in \mathcal{N}_\alpha) = 0.$$

Moreover, for any solution $R$ of (S41), since $R$ satisfies the constraint of (S41), we have that

$$\mathbb{E}(1 - \eta(X))(\mathbb{I}(X \in R) - \alpha) \leq 0 = \mathbb{E}(1 - \eta(X))(\mathbb{I}(X \in R_\alpha^\star) - \alpha)$$

Combining this with the fact that $\eta(x) \geq c(\alpha)$ when $x \in R_\alpha^\star \setminus R$ and $\eta(x) \leq c(\alpha)$ when $x \in R \setminus R_\alpha^\star$, we obtain that

$$\mathbb{E}(1 - c(\alpha))\mathbb{I}(X \in R \setminus R_\alpha^\star) \leq \mathbb{E}(1 - \eta(X))\mathbb{I}(X \in R \setminus R_\alpha^\star)$$
$$\leq \quad \mathbb{E}(1 - \eta(X))\mathbb{I}(X \in R_\alpha^\star \setminus R) \leq \mathbb{E}(1 - c(\alpha))\mathbb{I}(X \in R_\alpha^\star \setminus R) \,.$$

Using the fact that $c(\alpha) < 1$, we obtain that

$$\mathbb{E}\mathbb{I}(X \in R \setminus R_\alpha^\star) \leq \mathbb{E}\mathbb{I}(X \in R_\alpha^\star \setminus R) \,. \tag{S44}$$

Consequently, the optimality of $R_\alpha^\star$ follows from

$$\mathbb{E}\left\{\eta(X)\mathbb{I}(X \in R_\alpha^\star)\right\} - \mathbb{E}\left\{\eta(X)\mathbb{I}(X \in R)\right\}$$
$$= \quad \mathbb{E}\{\eta(X)(\mathbb{I}(X \in R_\alpha^\star \setminus R) - \mathbb{I}(X \in R \setminus R_\alpha^\star))\}$$
$$\geq \quad c(\alpha)(\mathbb{E}\mathbb{I}(X \in R_\alpha^\star \setminus R) - \mathbb{E}\mathbb{I}(X \in R \setminus R_\alpha^\star)) \geq 0$$

Since $R$ is also a solution, we must have $\mathbb{E}\left\{\eta(X)\mathbb{I}(X \in R_\alpha^\star)\right\} = \mathbb{E}\left\{\eta(X)\mathbb{I}(X \in R)\right\}$. As a result,

$$\mathbb{E}\{\eta(X)(\mathbb{I}(X \in R_\alpha^\star \setminus R) - \mathbb{I}(X \in R \setminus R_\alpha^\star))\} = c(\alpha)(\mathbb{E}\mathbb{I}(X \in R_\alpha^\star \setminus R) - \mathbb{E}\mathbb{I}(X \in R \setminus R_\alpha^\star)) = 0 \,.$$

Hence, we must have that up to a $\mathbb{P}_X$-null set, $R_\alpha^\star \setminus R$ and $R \setminus R_\alpha^\star$ are both subsets of $\{x : \eta(x) = c(\alpha)\}$ and $\mathbb{P}(X \in R_\alpha^\star \setminus R) = \mathbb{P}(X \in R \setminus R_\alpha^\star)$. Thus, the solution $R$ must be in the form of $R = \{x : \eta(x) > c(\alpha)\} \cup \tilde{\mathcal{N}}_\alpha$, where $\tilde{\mathcal{N}}_\alpha \subseteq \{x : \eta(x) = c(\alpha)\}$ and $\mathbb{P}(X \in \tilde{\mathcal{N}}_\alpha) = \mathbb{P}(X \in \mathcal{N}_\alpha)$.

Next, we consider the scenario that $\alpha \in \left(1 - \frac{\mathbb{P}(\eta(X)=0)}{p^-}, 1\right]$. In this scenario, we have $c(\alpha) = 0$.

Moreover, note that $R_\alpha^\star$ satisfies the constraint

$$\mathbb{E}(1 - \eta(X))(\mathbb{I}(X \in R_\alpha^\star) - \alpha) = \mathbb{E}(1 - \eta(X))(\mathbb{I}(\eta(X) > 0) - \alpha) + \mathbb{P}(X \in \mathcal{N}_\alpha) \leq 0.$$

For any solution $R$, we have that

$$\mathbb{E}\eta(X)\mathbb{I}(X \in R) \geq \mathbb{E}\eta(X)\mathbb{I}(X \in R_\alpha^\star) = \mathbb{E}\eta(X)\mathbb{I}(\eta(X) > 0).$$

On the other hand,

$$\mathbb{E}\eta(X)\mathbb{I}(X \in R) = \mathbb{E}\eta(X)\mathbb{I}(X \in R \cap \{x : \eta(x) > 0\}) \leq \mathbb{E}\eta(X)\mathbb{I}(\eta(X) > 0).$$

Therefore, we must have

$$\mathbb{E}\eta(X)\mathbb{I}(X \in R \cap \{x : \eta(x) > 0\}) = \mathbb{E}\eta(X)\mathbb{I}(\eta(X) > 0),$$

which implies that, up to a $\mathbb{P}_X$-null set, we must have $\{x : \eta(x) > 0\} \subseteq R$. Denote $R = \{x : \eta(x) > 0\} \cup \tilde{\mathcal{N}}_\alpha$, since $R$ satisfies constraint $\mathbb{E}(1 - \eta(X))(\mathbb{I}(X \in R) - \alpha) \leq 0$, which implies that

$$\mathbb{E}(1 - \eta(X))(\mathbb{I}(\eta(X) > 0) - \alpha) + \mathbb{P}(\tilde{\mathcal{N}}_\alpha) \leq 0$$

which is the first condition in (S43). In sum, any solution $R$ must be of the form $R = \{x : \eta(x) > 0\} \cup \mathcal{N}_\alpha$, where $\mathcal{N}_\alpha \subseteq \{x : \eta(x) = 0\}$ and

$$\mathbb{E}(1 - \eta(X))(\mathbb{I}(\eta(X) > 0) - \alpha) + \mathbb{P}(X \in \mathcal{N}_\alpha) \leq 0.$$

At the solution, when $\alpha \in \left(1 - \frac{\mathbb{P}(\eta(X)=0)}{p^-}, 1\right]$, we have that

$$
\begin{aligned}
\text{TPR}^\star &= \mathbb{P}(X \in R_\alpha^\star \mid Y = 1) = \frac{\mathbb{E}\eta(X)\mathbb{I}(X \in R_\alpha^\star)}{p^+} = 1 \\
\text{FPR}^\star &= \mathbb{P}(X \in R_\alpha^\star \mid Y = -1) = \frac{\mathbb{E}(1 - \eta(X))\mathbb{I}(X \in R_\alpha^\star)}{p^-} \\
&= \frac{\mathbb{E}(1 - \eta(X))\mathbb{I}(\eta(X) > 0)}{p^-} + \frac{\mathbb{P}(X \in \mathcal{N}_\alpha)}{p^-} \\
&= 1 - \frac{\mathbb{P}(\eta(X) = 0)}{p^-} + \frac{\mathbb{P}(X \in \mathcal{N}_\alpha)}{p^-} \in \left[1 - \frac{\mathbb{P}(\eta(X) = 0)}{p^-}, \alpha\right]
\end{aligned}
$$

When $\alpha \in \left(0, 1 - \frac{\mathbb{P}(\eta(X)=0)}{p^-}\right]$, we have that

$$
\begin{aligned}
\text{TPR}^\star &= \mathbb{P}(X \in R_\alpha^\star \mid Y = 1) = \frac{\mathbb{E}\eta(X)\mathbb{I}(X \in R_\alpha^\star)}{p^+} = \frac{\mathbb{E}\eta(X)\mathbb{I}(\eta(X) > c(\alpha)) + \mathbb{P}(X \in \mathcal{N}_\alpha)}{p^+} \\
&= \frac{\mathbb{P}(\eta(X) > c(\alpha)) - \alpha p^- - c(\alpha)\mathbb{E}\eta(X)\mathbb{I}(\eta(X) > c(\alpha))}{p^+(1 - c(\alpha))}, \\
\text{FPR}^\star &= \mathbb{P}(X \in R_\alpha^\star \mid Y = -1) = \frac{\mathbb{E}(1 - \eta(X))\mathbb{I}(X \in R_\alpha^\star)}{p^-} = \alpha,
\end{aligned}
$$

where we have used the fact that

$$
\mathbb{E}(1 - \eta(X))\mathbb{I}(\eta(X) > c(\alpha)) + (1 - c(\alpha))\mathbb{P}(X \in \mathcal{N}_\alpha) = \alpha p^- .
$$

This completes the proof of (8).

**Proof of Corollary 1.** The claim follows immediately from applying Theorem 1, equation (8) in Theorem 2, and Lemma S1.

**Proof of Proposition 1.** Denote by

$$
\tilde{\gamma}_C(\mathcal{H}) = \{(\text{FPR}(h_{\alpha,\mathcal{H}}^\star), \text{TPR}(h_{\alpha,\mathcal{H}}^\star)) \mid \alpha \in [0, 1], h_{\alpha,\mathcal{H}}^\star \text{ is a solution to } (5)\}
$$

the set of FPR-TPR pairs generated by all possible solutions to (5) and let $\tilde{\gamma}_C^\star(\mathcal{H}) = \text{PF}(cl(\tilde{\gamma}_C(\mathcal{H})))$

denote its pareto frontier. We first show that

$$\gamma(\mathcal{H}) = \tilde{\gamma}_C^\star(\mathcal{H}) \,. \tag{S45}$$

To this end, we first prove that $\gamma(\mathcal{H}) \subseteq \tilde{\gamma}_C(\mathcal{H})$. Let $h \in \mathcal{H}$ be a classifier satisfying $(\mathrm{FPR}(h), \mathrm{TPR}(h)) \in \gamma(\mathcal{H})$. We shall prove that $h$ is a solution to (5) with $\alpha = \mathrm{FPR}(h)$. First, $h$ is feasible. Moreover, if $h$ is not a solution to (5), then there must exist $h' \in \mathcal{H}$, such that $\mathrm{TPR}(h') > \mathrm{TPR}(h)$ and $\mathrm{FPR}(h') \leq \alpha = \mathrm{FPR}(h)$, which means that $h'$ dominates $h$. This contradicts with the fact that $h$ is on the pareto frontier (c.f. Definition 1). Therefore, $h$ must be a solution of (5) with $\alpha = \mathrm{FPR}(h)$. This completes the proof of $\gamma(\mathcal{H}) \subseteq \tilde{\gamma}_C(\mathcal{H})$. Next, note that $\gamma(\mathcal{H}) = \mathrm{PF}(S(\mathcal{H})) \subseteq [0,1] \times [0,1]$, and $\gamma(\mathcal{H}) \subseteq cl(\tilde{\gamma}_C(\mathcal{H})) \subseteq S(\mathcal{H})$, because $S(\mathcal{H})$ is closed. Then applying Lemma S1, we have that $\gamma(\mathcal{H}) = \tilde{\gamma}_C^\star(\mathcal{H})$.

Next, we show that $\gamma_C^\star(\mathcal{H}) = \gamma(\mathcal{H})$. For any given $\alpha \in [0,1]$, we define

$$\gamma(\alpha) = \{(\mathrm{FPR}(h_{\alpha,\mathcal{H}}^\star), \mathrm{TPR}(h_{\alpha,\mathcal{H}}^\star)) : h_{\alpha,\mathcal{H}}^\star \text{ is a solution to (5) at } \alpha\}$$

to be the set of FPR-TPR pairs generated by all possible solutions to (5) at $\alpha$.

We first show that if $\gamma(\alpha)$ is a nonempty non-singleton, then there must exist $\alpha^\star \in [0,1]$ such that $\gamma(\alpha^\star) = \{(\alpha^\star, v^\star)\}$ is a singleton with $\gamma(\alpha^\star) \subseteq \gamma(\alpha)$ and $(\alpha^\star, v^\star) \succ (u,v)$ for any $(u,v) \in \gamma(\alpha)$, $(u,v) \neq (\alpha^\star, v^\star)$. To this end, let $\alpha^\star = \inf\{u : (u,v) \in \gamma(\alpha)\}$, by using part (ii) of Lemma S2, we have that $\gamma(\alpha^\star)$ is a singleton and $\gamma(\alpha^\star) \subseteq \gamma(\alpha)$. Moreover, by (i) in Lemma S2, we have that $v^\star = v$ for any $(u,v) \in \gamma(\alpha)$, which implies $(\alpha^\star, v^\star) \succ (u,v)$ for any $(u,v) \in \gamma(\alpha)$ and $(\alpha^\star, v^\star) \neq (u,v)$, because $\alpha^\star < u$.

Now, we are ready to prove that $\gamma_C^\star(\mathcal{H}) = \gamma(\mathcal{H})$. Let $A = \{\alpha : \gamma(\alpha) \text{ is empty or a singleton}\}$ and $B = \{\alpha : \gamma(\alpha) \text{ is a nonempty non-singleton}\}$. We first prove $\tilde{\gamma}_C^\star(\mathcal{H}) \subseteq \bigcup_{\alpha \in A} \gamma(\alpha)$ by contradiction. Suppose there exists $(u^\star, v^\star) \in \tilde{\gamma}_C^\star(\mathcal{H})$ such that $(u^\star, v^\star) \notin \bigcup_{\alpha \in A} \gamma(\alpha)$. Note $\tilde{\gamma}_C^\star(\mathcal{H}) \subseteq \tilde{\gamma}_C(\mathcal{H}) = \bigcup_{\alpha \in A \cup B} \gamma(\alpha)$, we must have $(u^\star, v^\star) \in \bigcup_{\alpha \in B} \gamma(\alpha)$. That is, there exists $\alpha \in B$, such

that $(u^\star, v^\star) \in \gamma(\alpha)$. Moreover, by using fact that $(u^\star, v^\star) \in \tilde{\gamma}_C^\star(\mathcal{H})$, we must have $u^\star = \inf\{u : (u, v) \in \gamma(\alpha)\}$, because it can not be strictly dominated by any points in $\gamma(\alpha)$. This combined with (ii) in Lemma S2 implies $\gamma(u^\star)$ is a singleton, which contradicts with the assumption that $(u^\star, v^\star) \notin \bigcup_{\alpha \in A} \gamma(\alpha)$. This proves $\tilde{\gamma}_C^\star(\mathcal{H}) \subseteq \bigcup_{\alpha \in A} \gamma(\alpha)$. Next, by definition of $\gamma_C(\mathcal{H})$, we have that $\bigcup_{\alpha \in A} \gamma(\alpha) \subseteq \gamma_C(\mathcal{H})$. Combining this with (S45) and the fact that $\tilde{\gamma}_C^\star(\mathcal{H}) \subseteq \bigcup_{\alpha \in A} \gamma(\alpha)$, we obtain that $\gamma(\mathcal{H}) \subseteq \bigcup_{\alpha \in A} \gamma(\alpha) \subseteq \gamma_C(\mathcal{H})$. Note that $\gamma_C(\mathcal{H}) \subseteq S(\mathcal{H})$, by applying Lemma S1, we have that $\gamma_C^\star(\mathcal{H}) = \gamma(\mathcal{H}) \subseteq \gamma_C(\mathcal{H})$. This completes the proof.

**Proof of Theorem 3.** We first write

$$\gamma_{W, V(\cdot)}(\mathcal{F}) = \gamma_{W_0, V(\cdot)}(\mathcal{F}) \cup \{(\mathrm{FPR}(h^\star_{w, V(\cdot)}), \mathrm{TPR}(h^\star_{w, V(\cdot)})) \mid w \in [0, 1] \text{ and } \mathbb{P}(\eta(X) = w) > 0\},$$

where $h^\star_{w, V(\cdot)} = \mathrm{Sign}(f^\star_{w, V(\cdot)})$ and

$$\gamma_{W_0, V(\cdot)}(\mathcal{F}) = \{(\mathrm{FPR}(h^\star_{w, V(\cdot)}), \mathrm{TPR}(h^\star_{w, V(\cdot)})) \mid w \in [0, 1] \text{ and } \mathbb{P}(\eta(X) = w) = 0\},$$

It suffices to show that

$$\gamma_{W_0, V(\cdot)}(\mathcal{F}) = B_1 \cup B_2 \cup S_0' \cup S_1', \tag{S46}$$

where $B_1$ and $B_2$ are defined in (S63).

Since $\mathcal{F}$ is correctly specified under $V(\cdot)$, it follows from Definition 2 that $h^\star_{w, V(\cdot)}(x) = \mathrm{Sign}(f^\star_{w, V(\cdot)}(x)) = \mathrm{Sign}(\eta(X) - w)$ for any $w \in [0, 1]$ with $\mathbb{P}(\eta(X) = w) = 0$, which implies that

$$\gamma_{W_0, V(\cdot)}(\mathcal{F}) = \{(\mathrm{FPR}(h^\star_w), \mathrm{TPR}(h^\star_w)) \mid w \in [0, 1] \text{ and } \mathbb{P}(\eta(X) = w) = 0\},$$

where $h^\star_w$ is a solution to (S4) with $\mathcal{H} = \mathcal{H}_a$. Moreover, note that we can further decompose $\gamma_{W_0, V(\cdot)}(\mathcal{F})$ as

$$\gamma_{W_0, V(\cdot)}(\mathcal{F}) = A_1 \cup C_1 \cup C_2, \tag{S47}$$

where $A_1$ is defined in (S64) and

$$
\begin{aligned}
C_1 &= \left\{(\mathrm{FPR}(h_w^\star), \mathrm{TPR}(h_w^\star)) \mid w \in (0,1), \mathrm{FPR}(h_w^\star) = 0 \text{ or } \bar{\alpha}, \mathbb{P}(\eta(X) = w) = 0\right\}, \\
C_2 &= \left\{(\mathrm{FPR}(h_w^\star), \mathrm{TPR}(h_w^\star)) \mid w = 0 \text{ or } 1, \mathbb{P}(\eta(X) = w) = 0\right\}.
\end{aligned}
\tag{S48}
$$

In view of (S65), we have $A_1 = B_1 \cup B_2$. Therefore, to prove (S46) it remains to show that $C_1 \cup C_2 = S_0' \cup S_1'$.

To this end, we first show that $C_1 = S_0 \cup S_1$, where $S_0$ and $S_1$ are defined in (S8) and (S9). By using (S61), it suffices to show that

$$
C_1 = \left\{(\mathrm{FPR}(h_w^\star), \mathrm{TPR}(h_w^\star)) \mid w \in (0,1), \mathrm{FPR}(h_w^\star) = 0 \text{ or } \bar{\alpha}\right\}.
$$

By definition of $C_1$, we have

$$
C_1 \subseteq \left\{(\mathrm{FPR}(h_w^\star), \mathrm{TPR}(h_w^\star)) \mid w \in (0,1), \mathrm{FPR}(h_w^\star) = 0 \text{ or } \bar{\alpha}\right\}.
$$

Next, we show that

$$
C_1 \supseteq \left\{(\mathrm{FPR}(h_w^\star), \mathrm{TPR}(h_w^\star)) \mid w \in (0,1), \mathrm{FPR}(h_w^\star) = 0 \text{ or } \bar{\alpha}\right\}.
\tag{S49}
$$

First, it is easy to check that $\mathrm{TPR}(h_w^\star) = s(0)$ when $\mathrm{FPR}(h_w^\star) = 0$ for some $w \in (0,1)$, and $\mathrm{TPR}(h_w^\star) = s(\bar{\alpha})$ when $\mathrm{FPR}(h_w^\star) = \bar{\alpha}$ for some $w \in (0,1)$. Therefore, we only need to show that for any $w \in (0,1)$ with $\mathrm{FPR}(h_w^\star) = 0$ (or $\bar{\alpha}$), there exists $w' \in (0,1)$ with $\mathbb{P}(\eta(X) = w') = 0$ such that $\mathrm{FPR}(h_{w'}^\star) = 0$ (or $\bar{\alpha}$). If $\mathrm{FPR}(h_w^\star) = 0$ for some $w \in (0,1)$, by using (S4) we have that $0 \leq \mathbb{E}(1 - \eta(X))\mathbb{I}(w < \eta(X) < 1) \leq \mathbb{E}(1 - \eta(X))\mathbb{I}(h_w^\star(X) = 1) = 0$, which implies that $\mathbb{P}(w < \eta(X) < 1) = 0$. Now we pick any $w' \in (w, 1)$, we have that $\mathbb{P}(\eta(X) = w') = 0$ and $0 \leq \mathrm{FPR}(h_{w'}^\star) \leq \mathbb{P}(\eta(X) \geq w' \mid Y = -1) \leq \mathbb{P}(w < \eta(X) < 1)/p^- + \mathbb{P}(\eta(X) = 1 \mid Y = -1) = \mathbb{P}(w < \eta(X) < 1)/p^- + \mathbb{E}(1 - \eta(X))\mathbb{I}(\eta(X) = 1)/p^- = 0$, which implies $\mathrm{FPR}(h_{w'}^\star) = 0$. Similarly,

if $\text{FPR}(h_w^\star) = \bar{\alpha}$ for some $w \in (0,1)$, we have $p^- - \mathbb{P}(\eta(X) = 0) = \mathbb{E}(1 - \eta(X))\mathbb{I}(h_w^\star(X) = 1) \leq$

$\mathbb{E}(1 - \eta(X))\mathbb{I}(0 < \eta(X) < 1) = p^- - \mathbb{P}(\eta(X) = 0) = p^-\bar{\alpha}$, which implies that $\mathbb{P}(0 < \eta(X) < w) = 0$.

Again we pick any $w' \in (0, w)$, we have that $\mathbb{P}(\eta(X) = w') = 0$ and $\bar{\alpha} = \text{FPR}(h_w^\star) \leq \text{FPR}(h_{w'}^\star) \leq$

$\mathbb{P}(\eta(X) > 0 \mid Y = -1) = \bar{\alpha}$, which implies $\text{FPR}(h_{w'}^\star) = \bar{\alpha}$. This proves (S49), and completes the

proof of $C_1 = S_0 \cup S_1$.

Since $(\text{FPR}(h_w^\star), \text{TPR}(h_w^\star)) = (\mathbb{P}(\eta(X) > 0 \mid Y = -1), \mathbb{P}(\eta(X) > 0 \mid Y = 1)) = (1,1) =$

$(1 - \mathbb{P}(\eta(X) = 0)/p^-, 1)$ at $w = 0$ if $\mathbb{P}(\eta(X) = 0) = 0$, and $(\text{FPR}(h_w^\star), \text{TPR}(h_w^\star)) = (\mathbb{P}(\eta(X) > 1 \mid$

$Y = -1), \mathbb{P}(\eta(X) > 1 \mid Y = 1)) = (0,0) = (0, s(0))$ at $w = 1$ if $\mathbb{P}(\eta(X) = 1) = 0$, we have that

$C_2 = S_0'' \cup S_1''$, where

$$S_0'' = \begin{cases} \{(0, s(0))\}, & \text{if } \mathbb{P}(\eta(X) = 1) = 0 \;, \\ \emptyset, & \text{otherwise}\,, \end{cases} \tag{S50}$$

and

$$S_1'' = \begin{cases} \{(1 - \mathbb{P}(\eta(X) = 0)/p^-, 1)\}, & \text{if } \mathbb{P}(\eta(X) = 0) = 0\,, \\ \emptyset, & \text{otherwise}\,. \end{cases} \tag{S51}$$

Therefore, we have $C_1 \cup C_2 = S_0 \cup S_1 \cup S_0'' \cup S_1'' = S_0' \cup S_1'$. This completes the proof of Theorem 3.

Next we show Corollary S2. To this end, we first present a similar result to Corollary S3, which

reveals that all nonlinear parts of the optimal ROC curve can be recovered by the weighted method,

when $\mathcal{F}$ is correctly specified under a surrogate loss $V(\cdot)$.

**Corollary S5.** *Denote by $I_l = (\alpha_l^-, \alpha_l^+)$ all the disjoint intervals over which $s(\alpha)$ is linear, and*

*$\mathcal{I} = \bigcup_l I_l$. If $\mathcal{F}$ is correctly specified under a surrogate loss $V(\cdot)$ for the weighted method, then*

$$\gamma_{W,V(\cdot)}^\star(\mathcal{F}) \supseteq \{(\alpha, s(\alpha)) \mid \alpha \in [0, 1 - \mathbb{P}(\eta(X) = 0)/p^-] \setminus \mathcal{I}\}, \tag{S52}$$

*where $\gamma_{W,V(\cdot)}^\star(\mathcal{F}) = PF(cl(\gamma_{W,V(\cdot)}(\mathcal{F})))$ is the pareto frontier of $\gamma_{W,V(\cdot)}(\mathcal{F})$.*

**Proof of Corollary S5.** First using (S67) and the fact that $S'_j \supseteq S_j; j = 0, 1$, we obtain that

$$\{(\alpha, s(\alpha)) \mid \alpha \in [0, \bar{\alpha}] \setminus \mathcal{I}\} = \mathrm{cl}(A_1) \cup S_0 \cup S_1 \subseteq \mathrm{cl}(A_1) \cup S'_0 \cup S'_1. \tag{S53}$$

Next, by using Theorem 3 we have $A_1 \cup S'_0 \cup S'_1 \subseteq \gamma_{W,V(\cdot)}(\mathcal{H}_a)$, which implies

$$\mathrm{cl}(A_1) \cup S'_0 \cup S'_1 = \mathrm{cl}(A_1 \cup S'_0 \cup S'_1) \subseteq \mathrm{cl}(\gamma_{W,V(\cdot)}(\mathcal{H}_a)). \tag{S54}$$

Combining (S53) and (S54), we have that $\{(\alpha, s(\alpha)) \mid \alpha \in [0, \bar{\alpha}] \setminus \mathcal{I}\} \subseteq \mathrm{cl}(\gamma_{W,V(\cdot)}(\mathcal{H}_a))$. Using this and the definition of optimal ROC curve $\gamma(\mathcal{H}_a)$, we obtain (S52). This completes the proof of Corollary S5.

**Proof of Corollary S2.** Recall in the proof of Corollary S5, we show that $\{(\alpha, s(\alpha)) \mid \alpha \in [0, \bar{\alpha}] \setminus \mathcal{I}\} \subseteq \mathrm{cl}(\gamma_{W,V(\cdot)}(\mathcal{H}_a))$. By combining this with Lemma S5, we obtain (S2). This completes proof of Corollary S2.

**Proof of Proposition 2.** By Theorem S3, for any $w \in [0, 1]$ with $\mathbb{P}(\eta(X) = w) = 0$, the solution to the weighted problem over $\mathcal{H}_a$ must be of the form $\{x : f^\star_w(x) > 0\} = \{x : h^\star_w(x) = 1\} = \{x : \eta(x) > w\}$ up to a $\mathbb{P}_X$-null set. This implies that the set of all measurable functions $\mathcal{F}_a$ is correctly-specified under the 0/1 loss according to Definition 2.

Next, we verify its correct specification for any classification-calibrated loss $V(\cdot)$. To proceed, for any $x \in \mathbb{R}^p$ and $w \in [0, 1]$, define

$$\Gamma_w(x) = \arg\min_{a \in \mathbb{R}} (1 - w)\eta(x)V(a) + w(1 - \eta(x))V(-a)$$

and $\mathcal{F}^\star_w = \{f \mid f(x) \in \Gamma_w(x) \text{ for any } x \in \mathbb{R}^p\}$. We consider three cases: $w = 0$, $w = 1$ and $w \in (0, 1)$.

For the case that $w = 0$ and $\mathbb{P}(\eta(X) = 0) = 0$, we have $\Gamma_0(x) = \arg\min_{a \in \mathbb{R}} \eta(x)V(a)$. Note when $\eta(x) > 0$, this reduces into $\Gamma_0(x) = \arg\min_{a \in \mathbb{R}} V(a)$. By definition of classification-calibrated loss, we obtain that $H_V(1) < H^-_V(1)$, where $H_V(\cdot)$ and $H^-_V(\cdot)$ are defined in (15) and (16). This means $a^\star_0 > 0$ for any $a^\star_0 \in \Gamma_0(x)$ when $\eta(x) > 0$, which further implies for any $f' \in \mathcal{F}^\star_0$, $f'(x) > 0$ when

$\eta(x) > 0$. Combining this with Lemma S6 and $\mathbb{P}(\eta(X) = 0) = 0$, we obtain that $f^\star_{w,V(\cdot)}(X) > 0$ almost surely for any solution $f^\star_{w,V(\cdot)}$ to (10).

Similarly, for the case that $w = 1$ and $\mathbb{P}(\eta(X) = 1) = 0$, we have that $\Gamma_1(x) = \arg\min_{a\in\mathbb{R}}(1 - \eta(x))V(-a)$. By using the fact that $H_V(0) < H_V^-(0)$, we can see $a_1^\star < 0$ for any $a_1^\star \in \Gamma_1(x)$ when $\eta(x) < 1$, which further implies for any $f' \in \mathcal{F}_0^\star$, $f'(x) > 0$ when $\eta(x) < 0$. Combining this with Lemma S6 and $\mathbb{P}(\eta(X) = 1) = 0$, we obtain that $f^\star_{w,V(\cdot)}(X) < 0$ almost surely for any solution $f^\star_{w,V(\cdot)}$ to (10).

Lastly, for the case that $w \in (0,1)$ and $\mathbb{P}(\eta(X) = w) = 0$, note $(1 - w)\eta(x) + w(1 - \eta(x)) > 0$ for any $x \in \mathbb{R}^p$, we have that

$$
\begin{aligned}
\Gamma_w(x) &= \arg\min_{a\in\mathbb{R}}(1 - w)\eta(x)V(a) + w(1 - \eta(x))V(-a) \\
&= \arg\min_{a\in\mathbb{R}}\frac{(1 - w)\eta(x)}{\eta(x) + w - 2w\eta(x)}V(a) + \frac{w(1 - \eta(x))}{\eta(x) + w - 2w\eta(x)}V(-a)
\end{aligned}
$$

By definition of classification-calibration, for any $a(x) \in \Gamma_w(x)$ with $\eta(x) \neq w$, we must have that $a(x)(\eta(x) - w) > 0$. This together with Lemma S6 and $\mathbb{P}(\eta(X) = w) = 0$, implies that $f^\star_{w,V(\cdot)}(X)(\eta(X) - w) > 0$ almost surely for any solution $f^\star_{w,V(\cdot)}$ to (10). This completes the proof of proposition 2.

**Proof of Theorem S1.** We shall present an example in which (S3) holds. In particular, we consider the same quadratic discriminant analysis (QDA) setup and choose the same model space $\mathcal{F}$ as in the proof of Theorem S2.

We first present an approach to approximating $\gamma_{W,V(\cdot)}(\mathcal{F})$ for the weighted method, where we use logistic loss $V(x) = \log(1 + e^{-x})$ as the surrogate loss. Let $(X_i, Y_i), i = 1, 2\ldots, n$ be a random sample generated from the QDA setting specified in proof of Theorem S2. For any pre-specified weight $w \in [0,1]$, we first compute the optimal solution $(\hat{\beta}(w), \hat{\beta}_0(w))$ of the following empirical risk minimization problems:

$$
\underset{\beta\in\mathbb{R}^p,\beta_0\in\mathbb{R}}{\text{minimize}} \sum_{i=1}^{n} \left((1 - w)\mathbb{I}(Y_i = 1) + w\mathbb{I}(Y_i = -1)\right)\log(1 + e^{-Y_i(X_i^\top\beta+\beta_0)}),
$$

which is just a weighted logistic regression problem. Then, we consider classifier $\hat{h}_{w,V(\cdot)}(x) = \text{Sign}(x^\top \hat{\beta}(w) + \hat{\beta}_0(w))$ by calculating the empirical false positive rate and empirical false negative rate over an independent random sample $(\tilde{X}_i, \tilde{Y}_i), i = 1, 2 \ldots, N$, which is generated from the same distribution. Lastly, we plot those empirical FPRs and empirical TPRs to approximate $\gamma_{W,V(\cdot)}(\mathcal{F})$. Throughout simulations, we set $n = 2000$ and $N = 10000$.

Next, to approximate FPR-TPR pairs generated by linear classifiers, we adopt the same scheme introduced in the proof of Theorem S2 and denote by $\mathcal{A}$ the set of generated pairs. We plots all these results in Figure S4, where the grey area is the Monte Carlo approximation of $\mathcal{A}$ and blue cure is the ROC curve for weighted method under logistic loss. It is obvious from Figure S4 both statements in (S3) are true in this setting. This completes the proof of Theorem S1.

**Proof of Theorem 4.** By definition of correct specification defined in Definition 4, we have that, up to a $\mathbb{P}_X$-null set, a solution to (17) is exactly $f^\star_{w,V(\cdot)}(x) = M(\eta(x))$ for some strictly increasing function $M(\cdot)$. Since $\eta(X)$ is a continuous random variable, we have that

$$
\begin{aligned}
\gamma_{T,V(\cdot)}(\mathcal{H}, w) &= \{(\text{FPR}(\text{Sign}(f^\star_{w,V(\cdot)}(x) - \delta)), \text{TPR}(\text{Sign}(f^\star_{w,V(\cdot)}(x) - \delta))) \mid \delta \in \mathbb{R}\} \\
&= \{(\text{FPR}(\text{Sign}(M(\eta(x)) - \delta)), \text{TPR}(\text{Sign}(M(\eta(x)) - \delta))) \mid \delta \in \mathbb{R}\} \\
&= \{(\text{FPR}(\text{Sign}(\eta(x) - w)), \text{TPR}(\text{Sign}(\eta(x) - w))) \mid w \in [0, 1]\}.
\end{aligned}
$$

Finally, by using Theorem 1 and the fact that $\eta(X)$ is a continuous random variable, we obtain that $\gamma_{T,V(\cdot)}(\mathcal{H}, w) = \gamma(\mathcal{H}_a)$. This completes the proof.

**Proof of Theorem 5.** We first rewrite optimization problem in (17) into

$$
\underset{f \in \mathcal{F}_a}{\text{minimize}} \ \mathbb{E}\left((1 - w)\eta(X)V(f(X)) + w(1 - \eta(X))V(-f(X))\right). \tag{S55}
$$

Since $V(\cdot)$ is proper and strictly convex, for any $x \in \mathcal{X}$ the solution to the following problem is unique:

$$
\underset{a \in \mathbb{R}}{\text{minimize}} \ (1 - w)\eta V(a) + w(1 - \eta)V(-a). \tag{S56}
$$

Denote by $a_w^\star(\eta)$ the unique solution to above problem. By using optimality of $a_w^\star(\eta)$, we have that any solution $f_w^\star$ to (S55) satisfies $f_w^\star(X) = a_w^\star(\eta(X))$ up to a $\mathbb{P}_X$ null set. Therefore, by definition 4 it suffices to show $a_w^\star(\eta)$ is a strictly increasing function.

To this end, we first show for any $\eta_1 \neq \eta_2$, $a_w^\star(\eta_1) \neq a_w^\star(\eta_2)$. We show this by contradiction. Suppose $\eta_1 \neq \eta_2$ but $a_w^\star(\eta_1) = a_w^\star(\eta_2)$. By optimality of $^\star(\eta_1)$ and $a_w^\star(\eta_2)$, we have that

$$(1-w)\eta_1 V'(a_w^\star(\eta_1)) - w(1-\eta_1)V'(-a_w^\star(\eta_1)) = 0\,,$$
$$(1-w)\eta_2 V'(a_w^\star(\eta_2)) - w(1-\eta_2)V'(-a_w^\star(\eta_2)) = 0\,.$$

Note that $V'(x) < 0$, since $V(\cdot)$ is strictly decreasing. Hence, we must have either $w = 0$, $\eta_1 = \eta_2 = 0$ or $w = 1$, $\eta_1 = \eta_2 = 1$. Both contradict with the assumption that $\eta_1 \neq \eta_2$.

Next, we show $a_w^\star(\eta)$ is strictly increasing with respect to $\eta$. Let $\eta_1 < \eta_2$, we have that $a_w^\star(\eta_1) \neq a_w^\star(\eta_2)$. Moreover, by using the optimality of $a_w^\star(\eta_1)$ and $a_w^\star(\eta_2)$, we obtain that

$$(1-w)\eta_1 V(a_w^\star(\eta_1)) + w(1-\eta_1)V(-a_w^\star(\eta_1)) \;<\; (1-w)\eta_1 V(a_w^\star(\eta_2)) + w(1-\eta_1)V(-a_w^\star(\eta_2))\,,$$
$$(1-w)\eta_2 V(a_w^\star(\eta_2)) + w(1-\eta_2)V(-a_w^\star(\eta_2)) \;<\; (1-w)\eta_2 V(a_w^\star(\eta_1)) + w(1-\eta_2)V(-a_w^\star(\eta_1))\,.$$

Combining these two inequalities, we get

$$(\eta_1 - \eta_2)\left\{(1-w)V(a_w^\star(\eta_1)) - wV(-a_w^\star(\eta_1))\right\} < (\eta_1 - \eta_2)\left\{(1-w)V(a_w^\star(\eta_2)) - wV(-a_w^\star(\eta_2))\right\}\,,$$

or equivalently

$$(\eta_1 - \eta_2)\left\{(1-w)(V(a_w^\star(\eta_1)) - V(a_w^\star(\eta_2))) - w(V(-a_w^\star(\eta_1)) - V(-a_w^\star(\eta_2)))\right\} < 0\,,$$

which further implies $a_w^\star(\eta_1) < a_w^\star(\eta_2)$ by using fact that $V(\cdot)$ is strictly decreasing. This completes the proof of Theorem 5.

**Proof of Theorem S2.** We present a numerical example to demonstrate that the ROC curve

generated by the cutoff method can be dominated by the optimal ROC curve over certain class of model space. In particular, we assume that $\mathbb{P}(Y = 1) = 1/2$, $\mathbb{P}(Y = -1) = 1/2$, and conditioned on $Y$, $X \sim N(Y\boldsymbol{\mu}, \Sigma_Y)$, where $\boldsymbol{\mu} = (1, 1)^\top \in \mathbb{R}^2$, and

$$\Sigma_1 = \begin{pmatrix} 8 & 5 \\ 5 & 4 \end{pmatrix}, \ \Sigma_{-1} = \begin{pmatrix} 1 & -3 \\ -3 & 16 \end{pmatrix}. \tag{S57}$$

Moreover, the model space we choose is $\mathcal{F} = \{f(x) = x^\top \beta + \beta_0 \mid \beta \in \mathbb{R}^2 \text{ and } \beta_0 \in \mathbb{R}\}$.

We first generate FPR-TPR pairs $(\mathbb{P}(f(X) > 0 \mid Y = -1), \mathbb{P}(f(X) > 0 \mid Y = 1))$. Note that the FPR-TPR pair generated by $f \in \mathcal{F}$ only depends on its sign $\text{Sign}(f)$. Denote by $\mathcal{H} = \{\text{Sign}(f) \mid f \in \mathcal{F}\}$ the set of classifier induced by $\mathcal{F}$. Let $\mathcal{F}_0 = \{f(x) = x^\top \beta \mid \beta \in \mathbb{R}^2\}$, $\mathcal{F}_1 = \{f(x) = x^\top \beta + 1 \mid \beta \in \mathbb{R}^2\}$, and $\mathcal{F}_2 = \{f(x) = x^\top \beta - 1 \mid \beta \in \mathbb{R}^2\}$. Note for any $f \in \mathcal{F}$ and $k > 0$, $\text{Sign}(kf) = \text{Sign}(f)$. Hence, we have $\mathcal{H} = \{\text{Sign}(f) \mid f \in \mathcal{F}_0 \cup \mathcal{F}_1 \cup \mathcal{F}_2\}$. This allows us to consider only $\beta \in \mathbb{R}^2$ when generating FPR-TPR pairs.

To proceed, let $\{r_k > 0 \mid k = 1, 2 \ldots, n_r\}$ be a sequence of prespecified radius. For each $r_k$, we sample $N$ points $\{(s_i^{(k)}, t_i^{(k)}) \mid i = 1, 2 \ldots, N\}$ uniformly on the 2-dimensional unit sphere. For each combination of $r_k$ and $(s_i^{(k)}, t_i^{(k)})$, we set $\beta_i^{(k)} = (r_k s_i^{(k)}, r_k t_i^{(k)})$, and generate three FPR-TPR pairs:

$$(\mathbb{P}(X^\top \beta_i^{(k)} > 1 \mid Y = -1), \mathbb{P}(X^\top \beta_i^{(k)} > 1 \mid Y = 1))$$
$$(\mathbb{P}(X^\top \beta_i^{(k)} > -1 \mid Y = -1), \mathbb{P}(X^\top \beta_i^{(k)} > -1 \mid Y = 1))$$
$$(\mathbb{P}(X^\top \beta_i^{(k)} > 0 \mid Y = -1), \mathbb{P}(X^\top \beta_i^{(k)} > 0 \mid Y = 1)).$$

Let $\mathcal{A}$ be the set of pairs generated by the above scheme. Next, we find a FPR-TPR pair $(u_w^\star, v_w^\star)$ such that any other FPR-TPR pairs are under the line $wp^-(u - u_w^\star) - (1 - w)p^+(v - v_w^\star) = 0$ by using a grid search method on $\mathcal{A}$. Denote by $f_w^\star \in \mathcal{F}$ the discriminant function corresponding to $(u_w^\star, v_w^\star)$. We calculate the ROC curve generated by the cutoff method accordingly by evaluating $(\text{FPR}(f_\omega^\star - \delta), \text{TPR}(f_\omega^\star - \delta))$ for $\delta \in \mathbb{R}$. All these results are shown in Figure S5, in which the grey area is a Monte Carlo approximation of $\mathcal{A}$, and the blue curve is the ROC curve for the cutoff

method. Clearly, the ROC curve generated by cutoff method is dominated by population ROC curve. This completes the proof of Theorem S2.

**Proof of Theorem S3.** We first prove (S6). For any measurable classifier $h$, define $R_h := \{x : h(x) = 1\}$. Then, we write the objective function of (S4) in terms of $R_h$ as:

$$
\begin{aligned}
&(1-w)\mathbb{P}(h(X) = -1, Y = 1) + w\mathbb{P}(h(X) = 1, Y = -1) \\
=\ & (1-w)\mathbb{E}((1 - \mathbb{I}(X \in R_h))\eta(X)) + w\mathbb{E}(\mathbb{I}(X \in R_h)(1 - \eta(X))) \\
=\ & (1-w)p^+ - \mathbb{E}(\mathbb{I}(X \in R_h)(\eta(X) - w)) \,.
\end{aligned}
$$

It is easy to see that the minimizer $h_w^\star$ must be of the form $\{x : h_w^\star(x) = 1\} = \{x : \eta(x) > w\} \cup \mathcal{N}_w$ up to a $\mathbb{P}_X$-null set, where $\mathcal{N}_w \subseteq \{\eta(X) = w\}$. This completes the proof of (S6).

Next, we prove (S7). Note that $\gamma_W(\mathcal{H}_a)$ can be decomposed as

$$
\begin{aligned}
\gamma_W(\mathcal{H}_a) = \\
\{(\mathrm{FPR}(h_w^\star), \mathrm{TPR}(h_w^\star)) \mid w = 0 \text{ or } w = 1\} \cup \{(\mathrm{FPR}(h_w^\star), \mathrm{TPR}(h_w^\star)) \mid w \in (0,1)\}. \quad \text{(S58)}
\end{aligned}
$$

We first show that

$$
\{(\mathrm{FPR}(h_w^\star), \mathrm{TPR}(h_w^\star)) \mid w = 0 \text{ or } w = 1\} = \{(0, s_0)\} \cup \{(s_1, 1)\}, \quad \text{(S59)}
$$

for some $s_0 \in [0, \mathbb{P}(\eta(X) = 1)/p^+]$ and $s_1 \in [1 - \mathbb{P}(\eta(X) = 0)/p^-, 1]$. Using (S6), we have that when $w = 1$,

$$
\begin{aligned}
\mathrm{FPR}(h_w^\star) &= \mathbb{P}(h_w^\star(X) = 1 \mid Y = -1) = \frac{\mathbb{E}(1 - \eta(X))(\mathbb{I}(\eta(X) > 1) + \mathbb{I}(X \in \mathcal{N}_w))}{p^-} = 0 \,, \\
\mathrm{TPR}(h_w^\star) &= \mathbb{P}(h_w^\star(X) = 1 \mid Y = 1) = \frac{\mathbb{E}\eta(X)(\mathbb{I}(\eta(X) > 1) + \mathbb{I}(X \in \mathcal{N}_w))}{p^+} \in [0, \mathbb{P}(\eta(X) = 1)/p^+] \,.
\end{aligned}
$$

When $w = 0$, we have that

$$\mathrm{FPR}(h_w^\star) = \mathbb{P}(h_w^\star(X) = 1 \mid Y = -1) \in \left[1 - \mathbb{P}(\eta(X) = 0)/p^-, 1\right],$$

$$\mathrm{TPR}(h_w^\star) = \mathbb{P}(h_w^\star(X) = 1 \mid Y = 1) = 1.$$

This proves (S59).

Next, note for any $w \in (0, 1)$,

$$\mathrm{FPR}(h_w^\star) = \frac{\mathbb{E}(1 - \eta(X))(\mathbb{I}(\eta(X) > w) + \mathbb{I}(X \in \mathcal{N}_w))}{p^-} \in [0, \bar{\alpha}], \tag{S60}$$

where $\bar{\alpha} = 1 - \mathbb{P}(\eta(X) = 0)/p^-$. Thus, we can further decompose the second set in the RHS of (S58) as

$$\{(\mathrm{FPR}(h_w^\star), \mathrm{TPR}(h_w^\star)) \mid w \in (0, 1)\} =$$
$$\{(\mathrm{FPR}(h_w^\star), \mathrm{TPR}(h_w^\star)) \mid w \in (0, 1), \mathrm{FPR}(h_w^\star) = 0 \text{ or } \bar{\alpha}\} \cup$$
$$\{(\mathrm{FPR}(h_w^\star), \mathrm{TPR}(h_w^\star)) \mid w \in (0, 1), \mathrm{FPR}(h_w^\star) \in (0, \bar{\alpha})\} .$$

We next show that

$$\{(\mathrm{FPR}(h_w^\star), \mathrm{TPR}(h_w^\star)) \mid w \in (0, 1), \mathrm{FPR}(h_w^\star) = 0 \text{ or } \bar{\alpha}\} = S_0 \cup S_1, \tag{S61}$$

where $S_0$ and $S_1$ are defined in (S8) and (S9). To this end, we give the necessary and sufficient conditions for $(\alpha, s(\alpha)) \in \{(\mathrm{FPR}(h_w^\star), \mathrm{TPR}(h_w^\star)) \mid w \in (0, 1)\}$ at the two boundary points $\alpha = 0$ and $\alpha = \bar{\alpha}$. In particular, we shall show that (i) $(0, s(0)) \in \{(\mathrm{FPR}(h_w^\star), \mathrm{TPR}(h_w^\star)) \mid w \in (0, 1)\}$ if and only if $c(0) < 1$; and (ii) $(\bar{\alpha}, s(\bar{\alpha})) \in \{(\mathrm{FPR}(h_w^\star), \mathrm{TPR}(h_w^\star)) \mid w \in (0, 1)\}$ if and only if $\lim_{\alpha \to (1 - \mathbb{P}(\eta(X) = 0)/p^-)^-} c(\alpha) > 0$.

For (i), by the proof of Proposition S2, we have that $s'_+(0) < \infty$ if and only if $c(0) < 1$. When $s'_+(0) < \infty$, for any $w \in (0, 1)$ such that $wp^-/((1 - w)p^+) > s'_+(0)$, we have $w > c(0)$ by (S92). By

using (ii) in Lemma S3, we have that $g(c(0)) = \mathbb{E}(1-\eta(X))\mathbb{I}(\eta(X) > c(0)) = 0$, which further implies

$\mathbb{P}(w \leq \eta(X) < 1) = 0$. Therefore we obtain $\text{FPR}(h_w^\star) = 0$ and $\text{TPR}(h_w^\star) = \mathbb{P}(\eta(X) = 1)/p^+ = s(0)$.

When $s'_+(0) = \infty$, for any $w \in (0,1)$, we can find $\alpha > 0$ such that $(s(\alpha)-s(0))/\alpha > wp^-/((1-w)p^+)$, which implies

$$(1-w)p^+(1-s(\alpha)) + wp^-\alpha < (1-w)p^+(1-s(0)).$$

Therefore $(0, s(0))$ can not be the FPR-TPR pairs calculated from $h_w^\star$. This completes the proof of (i).

Similarly, by the proof of Proposition S2, we have $s'_-(\bar{\alpha}) = 0$ if and only if $\bar{c} = \lim_{\alpha \to \bar{\alpha}} c(\alpha) = 0$. When $s'_-(\bar{\alpha}) > 0$, since $\bar{c} > 0$ and $c(\bar{\alpha}) = 0$, we have that $c(\alpha)$ is not continuous at $\alpha = \bar{\alpha}$. By using (iii) in Lemma S3, there exists $c \in (0,1)$ such that $\mathbb{P}(0 < \eta(X) < c) = 0$. Let $w \in (0,c)$, we have that

$$\{x : h_w^\star(x) = 1\} = \{x : \eta(x) > w\} \cup \mathcal{N}_w = \{x : \eta(x) > 0\}$$

up to $\mathbb{P}_X$ null set. Therefore we obtain $\text{FPR}(h_w^\star) = 1 - \mathbb{P}(\eta(X) = 0)/p^-$ and $\text{TPR}(h_w^\star) = 1$. When $s'_-(\bar{\alpha}) = 0$, for any $w \in (0,1)$, we can find $\alpha < \bar{\alpha}$ such that $(s(\bar{\alpha})-s(\alpha))/(\bar{\alpha}-\alpha) > wp^-/((1-w)p^+)$, which implies

$$(1-w)p^+(1-s(\alpha)) + wp^-\alpha < (1-w)p^+(1-s(\bar{\alpha})) + wp^-\bar{\alpha}.$$

Therefore $(\bar{\alpha}, s(\bar{\alpha}))$ can not be the FPR-TPR pairs calculated from $h_w^\star$. This completes the proof of (ii). This proves (S61).

Lastly, it remains to show that

$$\{(\text{FPR}(h_w^\star), \text{TPR}(h_w^\star)) \mid w \in (0,1), \text{FPR}(h_w^\star) \in (0,\bar{\alpha})\} = B_1 \cup B_2 \cup B_3, \tag{S62}$$

where

$$B_1 = \{(\alpha, s(\alpha)) : \mathbb{P}(\eta(X) = c(\alpha)) = 0, \alpha \in (0, \bar{\alpha})\}\,,$$

$$B_2 = \{(\alpha, s(\alpha)) : s(\alpha) \text{ is non-differentiable at } \alpha, \alpha \in (0, \bar{\alpha})\}\,,$$

$$B_3 = \{(\alpha, s(\alpha)) : \mathbb{P}(\eta(X) = c(\alpha)) > 0, \alpha = \text{FPR}(h_w^\star) \in (0, \bar{\alpha})\}\,. \tag{S63}$$

To this end, we write the LHS of (S62) as $A_1 \cup A_2$, where

$$A_1 = \{(\text{FPR}(h_w^\star), \text{TPR}(h_w^\star)) \mid w \in (0, 1), \text{FPR}(h_w^\star) \in (0, \bar{\alpha}), \mathbb{P}(\eta(X) = w) = 0\}\,,$$

$$A_2 = \{(\text{FPR}(h_w^\star), \text{TPR}(h_w^\star)) \mid w \in (0, 1), \text{FPR}(h_w^\star) \in (0, \bar{\alpha}), \mathbb{P}(\eta(X) = w) > 0\}\,. \tag{S64}$$

To prove (S62), our plan is to show that

$$A_1 = B_1 \cup B_2\,, \ A_2 \subseteq B_2 \cup B_3\,, \ B_3 \subseteq A_2 \cup B_2\,. \tag{S65}$$

The second and third fact in (S65) means that $A_2 \setminus B_2 \subseteq B_3 \setminus B_2$ and $B_3 \setminus B_2 \subseteq A_2 \setminus B_2$, which further implies that $A_2 \setminus B_2 = B_3 \setminus B_2$, or equivalently,

$$A_2 \cup B_2 = B_3 \cup B_2\,. \tag{S66}$$

By combining (S66) with the first fact in (S65), we get (S62).

We first prove $A_1 \subseteq B_1 \cup B_2$. To show this, we prove that for any $(\text{FPR}(h_w^\star), \text{TPR}(h_w^\star)) \in A_1$, we have either $(\text{FPR}(h_w^\star), \text{TPR}(h_w^\star)) \in B_1$ or $(\text{FPR}(h_w^\star), \text{TPR}(h_w^\star)) \in B_2$. By applying Lemma S4, we know that there must exist $\alpha_w$ such that $(\text{FPR}(h_w^\star), \text{TPR}(h_w^\star)) = (\alpha_w, s(\alpha_w))$ with $c(\alpha_w) \leq w$. When $c(\alpha_w) = w$, we have $\mathbb{P}(\eta(X) = c(\alpha_w)) = \mathbb{P}(\eta(X) = w) = 0$, which implies that $(\text{FPR}(h_w^\star), \text{TPR}(h_w^\star)) = (\alpha_w, s(\alpha_w)) \in B_1$. When $c(\alpha_w) < w$, by using (S17), we have $\mathbb{P}(c(\alpha_w) < \eta(X) < w) = 0$ and $\mathbb{E}(1 - \eta(X))\mathbb{I}(\eta(X) > c(\alpha_w)) = \alpha_w p^-$. Combining this with Proposition S2 and part (iii) of Lemma S3, it follows that $s(\alpha)$ is non-differentiable at $\alpha = \alpha_w$. This proves that

$(\mathrm{FPR}(h_w^\star), \mathrm{TPR}(h_w^\star)) = (\alpha_w, s(\alpha_w)) \in B_2$. This completes the proof of $A_1 \subseteq B_1 \cup B_2$.

Next we prove that $B_1 \cup B_2 \subseteq A_1$. We first show that $B_1 \subseteq A_1$. By definition of $B_1$, we need to show that for any $\alpha \in (0, \bar{\alpha})$ with $\mathbb{P}(\eta(X) = c(\alpha)) = 0$, we must have $(\alpha, s(\alpha)) \in A_1$. By part (ii) of Lemma S3, we have that

$$\mathbb{E}(1 - \eta(X))\mathbb{I}(\eta(X) > c(\alpha)) \leq \alpha p^- \leq \mathbb{E}(1 - \eta(X))\mathbb{I}(\eta(X) \geq c(\alpha)).$$

This, combined with $\mathbb{P}(\eta(X) = c(\alpha)) = 0$, implies $\mathbb{E}(1 - \eta(X))\mathbb{I}(\eta(X) > c(\alpha)) = \alpha p^-$. Let $w = c(\alpha)$. By using (S6), we have

$$\mathrm{FPR}(h_w^\star) = \frac{\mathbb{E}(1 - \eta(X))\mathbb{I}(\eta(X) > w)}{p^-} = \frac{\mathbb{E}(1 - \eta(X))\mathbb{I}(\eta(X) > c(\alpha))}{p^-} = \alpha.$$

By Lemma S4, we have $\mathrm{TPR}(h_w^\star) = s(\alpha)$. Therefore, it follows that $(\alpha, s(\alpha)) \in A_1$, which completes the proof of $B_1 \subseteq A_1$.

We next show that $B_2 \subseteq A_1$. By definition of $B_2$, it suffices to show that for any $\alpha \in (0, \bar{\alpha})$ at which $s(\alpha)$ is non-differentiable, we must have $(\alpha, s(\alpha)) \in A_1$. By Proposition S2 and part (iii) of Lemma S3, we have that $\mathbb{E}(1 - \eta(X))\mathbb{I}(\eta(X) > c(\alpha)) = \alpha p^-$ and there exists $c \in (c(\alpha), 1)$ such that $\mathbb{P}(c(\alpha) < \eta(X) < c) = 0$. Then, for any $w \in (c(\alpha), c)$, we can see that $\mathbb{P}(c(\alpha) < \eta(X) \leq w) = 0$. This, combined with (S6), implies that

$$\mathrm{FPR}(h_w^\star) = \frac{\mathbb{E}(1 - \eta(X))\mathbb{I}(\eta(X) > w)}{p^-} = \frac{\mathbb{E}(1 - \eta(X))\mathbb{I}(\eta(X) > c(\alpha))}{p^-} = \alpha.$$

Again by applying Lemma S4, we have that $\mathrm{TPR}(h_w^\star) = s(\alpha)$. This implies that $(\alpha, s(\alpha)) \in A_1$, which completes the proof of $B_2 \subseteq A_1$. This finishes the proof of $A_1 = B_1 \cup B_2$.

Next, we prove the second facts in (S65), which is $A_2 \subseteq B_2 \cup B_3$. To show this, we prove that for any $(\mathrm{FPR}(h_w^\star), \mathrm{TPR}(h_w^\star)) \in A_2$, we have either $(\mathrm{FPR}(h_w^\star), \mathrm{TPR}(h_w^\star)) \in B_2$ or $(\mathrm{FPR}(h_w^\star), \mathrm{TPR}(h_w^\star)) \in B_3$. By Lemma S4, there must exist $\alpha_w$ such that $(\mathrm{FPR}(h_w^\star), \mathrm{TPR}(h_w^\star)) = (\alpha_w, s(\alpha_w))$ with $c(\alpha_w) \leq w$. When $c(\alpha_w) = w$, we have $\mathbb{P}(\eta(X) = c(\alpha_w)) = \mathbb{P}(\eta(x) = w) > 0$, which implies

$(\text{FPR}(h_w^\star), \text{TPR}(h_w^\star)) = (\alpha_w, s(\alpha_w)) \in B_3$. When $c(\alpha_w) < w$, by a similar argument used in the proof of $A_1 \subseteq B_1 \cup B_2$, we have $s(\alpha)$ is non-differentiable at $\alpha_w$. This completes the proof of $A_2 \subseteq B_2 \cup B_3$.

It then remains to prove the last fact in (S65), which is $B_3 \subseteq A_2 \cup B_2$. To this end, we show that for any $\alpha_w = \text{FPR}(h_w^\star) \in (0, \bar{\alpha})$ with $\mathbb{P}(\eta(X) = c(\alpha_w)) > 0$, we have either $(\alpha_w, s(\alpha_w)) \in A_2$ or $s(\alpha)$ is non-differentiable at $\alpha_w$. Since $\alpha_w \in (0, \bar{\alpha})$, we must have $w \in (0, 1)$. By using Lemma S4, we have $c(\alpha_w) \leq w$ and $s(\alpha_w) = \text{TPR}(h_w^\star)$. When $c(\alpha_w) = w$, then we have $\mathbb{P}(\eta(X) = w) = \mathbb{P}(\eta(X) = c(\alpha_w)) > 0$, which further implies $(\alpha_w, s(\alpha_w)) = (\text{FPR}(h_w^\star), \text{TPR}(h_w^\star)) \in A_2$. When $c(\alpha_w) < w$, again by a similar argument used in the proof of $A_1 \subseteq B_1 \cup B_2$, we have $s(\alpha)$ is non-differentiable at $\alpha_w$. Therefore, we have $B_3 \subseteq A_2 \cup B_2$. This completes the proof of (S62).

Finally, (S7) follow trivially from (S59), (S61) and (S62). This completes the proof of Theorem S3.

**Proof of Corollary S3.** Denote $\bar{\alpha} = 1 - \mathbb{P}(\eta(X) = 0)/p^-$. We start by proving that

$$\{(\alpha, s(\alpha)) \mid \alpha \in [0, \bar{\alpha}] \setminus \mathcal{I}\} = \text{cl}(A_1) \cup S_0 \cup S_1, \tag{S67}$$

where $A_1$ is defined in (S64). To this end, we first show

$$\{(\alpha, s(\alpha)) \mid \alpha \in [0, \bar{\alpha}] \setminus \mathcal{I}\} \subseteq \text{cl}(A_1) \cup S_0 \cup S_1. \tag{S68}$$

To this end, we start by showing that

$$\{(0, s(0))\} \in \text{cl}(A_1) \cup S_0 \text{ and } \{(\bar{\alpha}, s(\bar{\alpha}))\} \in \text{cl}(A_1) \cup S_1 \tag{S69}$$

For the first claim in (S69), when $c(0) < 1$, we have $\{(0, s(0))\} = S_0 \subseteq \text{cl}(A_1) \cup S_0$. When $c(0) = 1$, by using definition of $c(0)$, we have $\mathbb{P}(c_0 < \eta(X) < 1) > 0$ for any $c_0 < 1$. Also note the set $\{w \in (0, 1) \mid \mathbb{P}(\eta(X) = w) > 0\}$ is countable, which implies that we can find $w_n \in (0, 1)$ with $\mathbb{P}(\eta(X) = w_n) = 0$ such that $w_n \to 1$ as $n \to \infty$. Combining these with (S6), we obtain that

$\alpha_{w_n} = \mathrm{FPR}(h^\star_{w_n}) = \mathbb{E}(1 - \eta(X))\mathbb{I}(\eta(X) > w_n)/p^- > 0$, $(\alpha_{w_n}, s(\alpha_{w_n})) \in A_1$ and

$$\alpha_{w_n} = \frac{1}{p^-}\mathbb{E}(1 - \eta(X))\mathbb{I}(\eta(X) > w_n) \to \frac{1}{p^-}\mathbb{E}(1 - \eta(X))\mathbb{I}(\eta(X) = 1) = 0\,.$$

Then using the continuity of $s(\alpha)$ (c.f., Theorem 1), we have that $\lim_{n\to\infty} s(\alpha_{w_n}) = s(0)$. Therefore, $\{(0, s(0))\} \in \mathrm{cl}(A_1) \subseteq \mathrm{cl}(A_1) \cup S_0$. This completes the proof of the first claim.

For the second claim in (S69), when $\lim_{\alpha \to \bar{\alpha}^-} c(\alpha) > 0$, then obviously $\{(\bar{\alpha}, s(\bar{\alpha}))\} \in \mathrm{cl}(A_1) \cup S_1$ since $\{(\bar{\alpha}, s(\bar{\alpha}))\} = S_1$. When $\lim_{\alpha \to \bar{\alpha}^-} c(\alpha) = 0$, note that part (iii) in Lemma S3 implies that $\mathbb{P}(0 < \eta(X) < c_1) > 0$ for any $c_1 > 0$. Therefore, we can find $0 < w'_n < c_1$ with $\mathbb{P}(\eta(X) = w'_n) = 0$ such that $w'_n \to 0$ as $n \to \infty$. By combining these with (S6), we obtain that $\alpha_{w'_n} = \mathrm{FPR}(h^\star_{w'_n}) = \mathbb{E}(1 - \eta(X))\mathbb{I}(\eta(X) > w'_n)/p^- < \bar{\alpha}$, $(\alpha_{w'_n}, s(\alpha_{w'_n})) \in A_1$ and

$$\alpha_{w'_n} = \frac{1}{p^-}\mathbb{E}(1 - \eta(X))\mathbb{I}(\eta(X) > w'_n) \to \frac{1}{p^-}\mathbb{E}(1 - \eta(X))\mathbb{I}(\eta(X) > 0) = \bar{\alpha}\,.$$

Again by using continuity of $s(\alpha)$, we have that $\lim_{n\to\infty} s(\alpha_{w'_n}) = s(\bar{\alpha}) = 1$. Therefore it follows that $\{(\bar{\alpha}, s(\bar{\alpha}))\} \in \mathrm{cl}(A_1) \subseteq \mathrm{cl}(A_1) \cup S_1$. This proves the second claim, and therefore completes the proof of (S69).

Next, we prove that

$$\{(\alpha, s(\alpha)) \mid \alpha \in (0, \bar{\alpha}) \setminus \mathcal{I}\} \subseteq \mathrm{cl}(A_1)\,. \tag{S70}$$

Since $B_1 \subseteq A_1$ (c.f. (S65)), by Proposition S1 we have that

$$\{(\alpha, s(\alpha)) \mid \alpha \in (0, \bar{\alpha}) \setminus \mathrm{cl}(\mathcal{I})\} \subseteq A_1\,.$$

Therefore it suffices to show $(\alpha_l^-, s(\alpha_l^-)) \in \mathrm{cl}(A_1)$ and $(\alpha_l^+, s(\alpha_l^+)) \in \mathrm{cl}(A_1)$ for any $\alpha_l^-, \alpha_l^+ \in (0, \bar{\alpha})$. Again using Proposition S1, for any $\alpha \in (\alpha_l^-, \alpha_l^+)$, we have $\mathbb{P}(\eta(X) = c(\alpha)) > 0$, $\alpha_l^- = \mathbb{E}(1 - \eta(X))\mathbb{I}(\eta(X) > c(\alpha))/p^-$ and $\alpha_l^+ = \mathbb{E}(1 - \eta(X))\mathbb{I}(\eta(X) \geq c(\alpha))/p^-$.

We first show $(\alpha_l^-, s(\alpha_l^-)) \in \mathrm{cl}(A_1)$. Since $\alpha_l^- > 0$, we can find $c(\alpha) < w_n < 1$ with $\mathbb{P}(\eta(X) =$

$w_n) = 0$ and $\alpha_{w_n} = \text{FPR}(h^\star_{w_n}) = \mathbb{E}(1 - \eta(X))\mathbb{I}(\eta(X) > w_n)/p^- > 0$ such that $w_n \to c(\alpha)$. Therefore we have that $\alpha_{w_n} \le \alpha_l^-$, $(\alpha_{w_n}, s(\alpha_{w_n})) \in A_1$ and

$$\alpha_{w_n} = \frac{1}{p^-}\mathbb{E}(1 - \eta(X))\mathbb{I}(\eta(X) > w_n) \to \frac{1}{p^-}\mathbb{E}(1 - \eta(X))\mathbb{I}(\eta(X) > c(\alpha)) = \alpha_l^-,$$

which further implies $s(\alpha_{w_n}) \to s(\alpha_l^-)$ by the continuity of $s(\alpha)$. This completes the proof of $(\alpha_l^-, s(\alpha_l^-)) \in \text{cl}(A_1)$.

We next show that $(\alpha_l^+, s(\alpha_l^+)) \in \text{cl}(A_1)$. Since $\alpha_l^+ < \bar\alpha$, we can find $0 < w_n' < c(\alpha)$ with $\mathbb{P}(\eta(X) = w_n') = 0$ and $\alpha_{w_n'} = \text{FPR}(h^\star_{w_n'}) = \mathbb{E}(1 - \eta(X))\mathbb{I}(\eta(X) > w_n')/p^- < \bar\alpha$ such that $w_n \to c(\alpha)$. This implies $\alpha_{w_n'} \ge \alpha_l^-$, $(\alpha_{w_n'}, s(\alpha_{w_n'})) \in A_1$ and

$$\alpha_{w_n'} = \frac{1}{p^-}\mathbb{E}(1 - \eta(X))\mathbb{I}(\eta(X) > w_n') \to \frac{1}{p^-}\mathbb{E}(1 - \eta(X))\mathbb{I}(\eta(X) \ge c(\alpha)) = \alpha_l^+.$$

Therefore we have $s(\alpha_{w_n'}) \to s(\alpha_l^+)$, which further means $(\alpha_l^+, s(\alpha_l^+)) \in \text{cl}(A_1)$. This finishes the proof of (S70).

Now we have that

$$\{(\alpha, s(\alpha)) \mid \alpha \in (0, \bar\alpha) \setminus \mathcal{I}\} \cup \{(0, s(0))\} \cup \{(\bar\alpha, s(\bar\alpha))\} \cup S_0 \cup S_1$$
$$\subseteq \text{cl}(A_1) \cup \{(0, s(0))\} \cup \{(\bar\alpha, s(\bar\alpha))\} \cup S_0 \cup S_1,$$

by using (S70). Combining this with facts that $S_0 \subseteq \{(0, s(0))\}$, $S_1 \subseteq \{(\bar\alpha, s(\bar\alpha))\}$ and the two claims in (S69), we finish the proof of (S68).

Our next step is to prove

$$\text{cl}(A_1) \cup S_0 \cup S_1 \subseteq \{(\alpha, s(\alpha)) \mid \alpha \in [0, \bar\alpha] \setminus \mathcal{I}\}. \tag{S71}$$

By using (S69), we have

$$\{(0, s(0))\} \cup \{(\bar{\alpha}, s(\bar{\alpha}))\} \subseteq \mathrm{cl}(A_1) \cup S_0 \cup S_1 \,.$$

Therefore to get (S71), it suffices to prove

$$\mathrm{cl}(A_1) \setminus (\{(0, s(0))\} \cup \{(\bar{\alpha}, s(\bar{\alpha}))\}) \subseteq \{(\alpha, s(\alpha)) \mid \alpha \in (0, \bar{\alpha}) \setminus \mathcal{I}\} \,. \tag{S72}$$

By using (S70), this is equivalent to show $(\alpha, s(\alpha)) \notin \mathrm{cl}(A_1)$ for any $\alpha \in (\alpha_l^-, \alpha_l^+)$. We show this claim by contradiction. Suppose $(\alpha, s(\alpha)) \in \mathrm{cl}(A_1)$ for some $\alpha \in (\alpha_l^-, \alpha_l^+)$. There must exist $w_n \in (0, 1)$ with $\alpha_n = \mathrm{FPR}(h_{w_n}^\star) \in (0, \bar{\alpha})$ and $\mathbb{P}(\eta(X) = w_n) = 0$, such that $\alpha_n \to \alpha$. Without loss of generality, we can assume $w_n \to w^\star$. Since $\alpha_n = \mathrm{FPR}(h_{w_n}^\star) = \mathbb{E}(1 - \eta(X))\mathbb{I}(\eta(X) > w_n)/p^-$, we have either $\alpha = \mathbb{E}(1 - \eta(X))\mathbb{I}(\eta(X) > w^\star)/p^-$ or $\alpha = \mathbb{E}(1 - \eta(X))\mathbb{I}(\eta(X) \geq w^\star)/p^-$. Moreover, by using Proposition S1, we have that $\mathbb{P}(\eta(X) = c(\alpha)) > 0$ and

$$\mathbb{E}(1 - \eta(X))\mathbb{I}(\eta(X) > c(\alpha)) < \alpha p^- < \mathbb{E}(1 - \eta(X))\mathbb{I}(\eta(X) \geq c(\alpha)) \,. \tag{S73}$$

When $\alpha = \mathbb{E}(1 - \eta(X))\mathbb{I}(\eta(X) > w^\star)/p^-$, the first inequality in (S73) implies $c(\alpha) > w^\star$, contradicts the second inequality since $c(\alpha) < 1$ (c.f. part (i) in Lemma S3) and $\mathbb{P}(\eta(X) = c(\alpha)) > 0$. Similar contradictions can be obtained for $\alpha = \mathbb{E}(1 - \eta(X))\mathbb{I}(\eta(X) \geq w^\star)/p^-$. This completes the proof of (S71). By combining (S68) and (S71), we finish the proof of (S67).

Now we are ready to prove (S10). By using Proposition S1, for any $(\alpha, s(\alpha)) \in B_3$ (c.f. (S63)), we have $\alpha \in [\alpha_l^-, \alpha_l^+]$ for some index $l$. Therefore we can represent $B_3$ with

$$B_3 = \bigcup_l \{(\alpha_l, s(\alpha_l))\} \tag{S74}$$

for some $\alpha_l \in [\alpha_l^-, \alpha_l^+]$. Moreover, by using (S7), we have that

$$
\begin{aligned}
\mathrm{cl}(\gamma_W(\mathcal{H}_a)) &= \mathrm{cl}(B_1 \cup B_2) \cup \mathrm{cl}(B_3) \cup \{(0, s_0)\} \cup \{(s_1, 1)\} \cup S_0 \cup S_1 \\
&= \mathrm{cl}(A_1) \cup S_0 \cup S_1 \cup \mathrm{cl}(B_3) \cup \{(0, s_0)\} \cup \{(s_1, 1)\} \\
&= \{(\alpha, s(\alpha)) \mid \alpha \in [0, \bar{\alpha}] \setminus \mathcal{I}\} \cup B_3 \cup \{(0, s_0)\} \cup \{(s_1, 1)\} \\
&= \{(\alpha, s(\alpha)) \mid \alpha \in [0, \bar{\alpha}] \setminus \mathcal{I}\} \cup (\bigcup_l \{(\alpha_l, s(\alpha_l))\}) \cup \{(0, s_0)\} \cup \{(s_1, 1)\}
\end{aligned}
$$

where the third equality uses (S67) and the fact that limiting points of $B_3$ must lie in $\{(\alpha, s(\alpha)) \mid \alpha \in [0, \bar{\alpha}] \setminus \mathcal{I}\}$. Note $(0, s_0) \not\succ (0, s(0))$ and $(s_1, 1) \not\succ (\bar{\alpha}, 1)$, we obtain that

$$
\gamma_W^\star(\mathcal{H}_a) = \mathrm{PF}(\mathrm{cl}(\gamma_W(\mathcal{H}_a))) = \{(\alpha, s(\alpha)) \mid \alpha \in [0, \bar{\alpha}] \setminus \mathcal{I}\} \cup (\bigcup_l \{(\alpha_l, s(\alpha_l))\})
$$

This completes the proof of Corollary S3.

**Proof of Corollary S4.** First by Theorem S3, we have that $\gamma_W(\mathcal{H}_a) \subseteq S(\mathcal{H}_a)$. Next, by using Corollary S3 and the definition of $\gamma_W^\star(\mathcal{H}_a)$, we get $\{(\alpha, s(\alpha)) \mid \alpha \in [0, \bar{\alpha}] \setminus \mathcal{I}\} \subseteq \gamma_W^\star(\mathcal{H}_a) \subseteq \mathrm{cl}(\gamma_W(\mathcal{H}_a))$. These combined with Lemma S5 implies $\mathrm{PF}(\overline{\mathrm{conv}}(\gamma_W(\mathcal{H}_a))) = \gamma(\mathcal{H}_a)$. Therefore we finish the proof of Corollary S4.

**Proof of Theorem S4.** We first prove

$$
\gamma_W(\mathcal{H}) \setminus \{(\mathrm{FPR}(h_{w,\mathcal{H}}^\star), \mathrm{TPR}(h_{w,\mathcal{H}}^\star)) : w = 0 \text{ or } w = 1\} \subseteq \gamma(\mathcal{H}). \tag{S75}
$$

For any $w \in [0, 1]$, let $h_{w,\mathcal{H}}^\star$ be a solution of (S4) and denote $(u_w, v_w) = (\mathrm{FPR}(h_{w,\mathcal{H}}^\star), \mathrm{TPR}(h_{w,\mathcal{H}}^\star))$. It suffices to show for any $w \in (0, 1)$, the solution of (S4) can not be dominated by any other point in $S(\mathcal{H})$. We show this by contradiction. Suppose that there exists $h \in S(\mathcal{H})$ such that $h \succ h_w$. Let $(u, v) = (\mathrm{FPR}(h), \mathrm{TPR}(h))$. Then we have that $u \leq u_w$, $v \geq v_w$ and $(u, v) \neq (u_w, v_w)$. This, together with $w \in (0, 1)$, implies that

$$
(1 - w)p^+(1 - v_w) + wp^- u_w > (1 - w)p^+(1 - v) + wp^- u,
$$

which contradicts with the optimality of $(u_w, v_w)$ for (S4) as

$$(1-w)p^+(1-v_w) + wp^- u_w \leq (1-w)p^+(1-v) + wp^- u.$$

This proves (S75).

Next we prove that $\gamma_W^\star(\mathcal{H}) \subseteq \gamma(\mathcal{H})$. Denote by $\tilde{\gamma}_W(\mathcal{H})$ the LHS of (S75). By definition of $\gamma_W^\star(\mathcal{H})$ we have that

$$\gamma_W^\star(\mathcal{H}) = \mathrm{PF}(\mathrm{cl}(\gamma_W(\mathcal{H}))) = \mathrm{PF}(\mathrm{cl}(\tilde{\gamma}_W(\mathcal{H})) \cup \{(u_0, v_0)\} \cup \{(u_1, v_1)\}).$$

To show $\gamma_W^\star(\mathcal{H}) \subseteq \gamma(\mathcal{H})$, by using (S75) it suffices to prove that if $(u_w, v_w) \notin \mathrm{cl}(\tilde{\gamma}_W(\mathcal{H}))$ for $w = 0$ or $1$, then there exists $h \in \mathcal{H}$ such that $(\mathrm{FPR}(h), \mathrm{TPR}(h)) \in \mathrm{cl}(\tilde{\gamma}_W(\mathcal{H}))$ and $h$ dominates $h_{w,\mathcal{H}}^\star$. First, we consider the case where $w = 0$. Assume $(u_{w_n}, v_{w_n})$ is a converging sequence for some $w_n \in (0, 1)$ such that $w_n \to 0$, and denote its limit as $(\tilde{u}_0, \tilde{v}_0) \in \mathrm{cl}(\gamma_W(\mathcal{H}))$. Note $(u_0, v_0) \notin \mathrm{cl}(\gamma_W(\mathcal{H}))$, we have that $(\tilde{u}_0, \tilde{v}_0) \neq (u_0, v_0)$. Since $S(\mathcal{H})$ is closed, there exists $h \in \mathcal{H}$ such that $(\mathrm{FPR}(h), \mathrm{TPR}(h)) = (\tilde{u}_0, \tilde{v}_0)$. We shall then show $h$ dominates $h_{w,\mathcal{H}}^\star$. On one hand, by the optimality of $(u_{w_n}, v_{w_n})$, we have that $(1-w_n)p^+(1-v_{w_n}) + w_n p^- u_{w_n} \leq (1-w_n)p^+(1-v_0) + w_n p^- u_0$, which implies

$$(u_0 - u_{w_n})w_n p^- \geq (1-w_n)p^+(v_0 - v_{w_n}). \tag{S76}$$

By letting $w_n \to 0$, we obtain that $\tilde{v}_0 \geq v_0$. On the other hand, by the optimality of $(u_0, v_0)$, we have that $v_{w_n} \leq v_0$. By combining this with (S76), we obtain that $(u_0 - u_{w_n})w_n p^- \geq (1-w_n)p^+(v_0 - v_{w_n}) \geq 0$, which implies that $u_{w_n} \leq u_0$. By letting $w_n \to 0$, we get $\tilde{u}_0 \leq u_0$. Therefore we show that $\tilde{u}_0 \leq u_0$ and $\tilde{v}_0 \geq v_0$, which together with $(\tilde{u}_0, \tilde{v}_0) \neq (u_0, v_0)$ implies that $(\tilde{u}_0, \tilde{v}_0)$ dominates $(\tilde{u}_0, \tilde{v}_0)$. This finishes the proof of the case that $w = 0$.

Next, we consider the case where $w = 1$. Similarly, assume $(u_{w_n}, v_{w_n})$ is a converging sequence for some $w_n \in (0, 1)$ such that $w_n \to 1$, and we denote its limit as $(\tilde{u}_1, \tilde{v}_1) \in \mathrm{cl}(\gamma_W(\mathcal{H}))$. Since $(u_1, v_1) \notin \mathrm{cl}(\gamma_W(\mathcal{H}))$, we have that $(\tilde{u}_1, \tilde{v}_1) \neq (u_1, v_1)$. By closedness of $S(\mathcal{H})$, there must exist $h \in \mathcal{H}$

such that $(\mathrm{FPR}(h), \mathrm{TPR}(h)) = (\tilde{u}_1, \tilde{v}_1)$. We next show that $h$ dominates $h^\star_{w,\mathcal{H}}$. On one hand, by the optimality of $(u_{w_n}, v_{w_n})$, we have that $(1-w_n)p^+(1-v_{w_n}) + w_n p^- u_{w_n} \leq (1-w_n)p^+(1-v_1) + w_n p^- u_1$, which further implies that

$$u_1 - u_{w_n} \geq \frac{(1-w_n)p^+}{w_n p^-}(v_1 - v_{w_n}). \tag{S77}$$

Letting $w_n \to 1$, we obtain that $\tilde{u}_1 \leq u_1$. On the other hand, by optimality of $(u_1, v_1)$, we have that $u_{w_n} \geq u_1$. Combining with (S77), we obtain that $v_1 \leq v_{w_n}$. Letting $w_n \to 1$, it follows that $\tilde{v}_1 \geq v_1$. Hence we have shown that $\tilde{u}_1 \leq u_1$ and $\tilde{v}_1 \geq v_1$, which, together with $(\tilde{u}_1, \tilde{v}_1) \neq (u_1, v_1)$, implies that $(\tilde{u}_1, \tilde{v}_1)$ dominates $(u_1, v_1)$. This finishes the proof for $w = 1$.

Next, we present an example to show that it is indeed possible to have $\gamma_W(\mathcal{H}) \neq \gamma(\mathcal{H})$ when the model is mis-specified. In particular, we consider a standard quadratic discriminant analysis (QDA) setup, in which we have

$$Y \sim \mathrm{Bernoulli}(1/2) \text{ and } X = \begin{cases} N(\mu_-, \Sigma_-) & \text{if } Y = -1 \\ N(\mu_+, \Sigma_+) & \text{if } Y = 1 \end{cases}$$

where $\mu_- = (0, 0, 0)^\top$, $\mu_+ = (1, 1, 1)^\top$, $\Sigma_- = \mathrm{Diag}\{1, 9, 16\}$ and $\Sigma_+ = \mathrm{Diag}\{9, 16, 1\}$. We consider the model space to be $\mathcal{H} = \{\mathrm{Sign}(f) \mid f \in \mathcal{F}\}$ with $\mathcal{F} = \{x^\top \beta - 1 \mid \beta_i \in [0, 1], i = 1, 2, 3\}$.

First, we generate FPR-TPR pairs $(\mathbb{P}(X^\top \beta > 1 \mid Y = -1), \mathbb{P}(X^\top \beta > 1 \mid Y = 1))$ using a grid search over $[0, 1]^3$ for $\beta$. All the FPR-TPR pairs are shown in both panels of Figure S3 as an approximation to $S(\mathcal{H})$. Next, we highlight all pairs on the upper boundary in the top panel of Figure S3, which corresponds to the population ROC curve of the constrained method $\gamma_C(\mathcal{H})$. We can see that the pareto frontier of population constrained ROC curve $\gamma_C^\star(\mathcal{H})$ coincides with $\gamma_C(\mathcal{H})$, which is also the same as the optimal ROC curve. For the weighted method, finding a solution to (S4) is equivalent to finding a FPR-TPR pair $(u_w^\star, v_w^\star)$ such that any other FPR-TPR pairs are under the line $wp^-(u - u_w^\star) - (1-w)p^+(v - v_w^\star) = 0$. The population ROC curve of the weighted method $\gamma_W(\mathcal{H})$ is the solid curve in the bottom panel of Figure S3. Clearly, we have $\gamma_W(\mathcal{H}) = \gamma_W^\star(\mathcal{H})$, and it is a strict subset of $\gamma(\mathcal{H})$. Moreover, we have $\mathrm{PF}(\overline{\mathrm{conv}}(\gamma_W(\mathcal{H})))$ does not contain $\gamma(\mathcal{H}) \setminus \gamma_W(\mathcal{H})$.

Therefore we have

$$\gamma(\mathcal{H}) \setminus \gamma_W^\star(\mathcal{H}) \neq \emptyset \text{ and } \gamma(\mathcal{H}) \setminus \mathrm{PF}(\overline{\mathrm{conv}}(\gamma_W(\mathcal{H}))) \neq \emptyset \,.$$

This completes the proof of Theorem S4.

**Proof of Proposition S1.** We first show that $c(\alpha) = c(\tilde{\alpha})$ for any $\alpha \in I(\tilde{\alpha})$. To prove this, we only need to show that $g(c(\tilde{\alpha})) \leq \alpha p^-$ and $g(c) > \alpha p^-$ for any $c < c(\tilde{\alpha})$. For the former claim, note that for any $\alpha \in I(\tilde{\alpha})$,

$$g(c(\tilde{\alpha})) = \mathbb{E}(1 - \eta(X))\mathbb{I}(\eta(X) > c(\tilde{\alpha})) \leq \alpha p^- \,. \tag{S78}$$

For the later claim, note that for any $c < c(\tilde{\alpha})$,

$$g(c) \geq g(c(\tilde{\alpha})) + (1 - c(\tilde{\alpha}))\mathbb{P}(\eta(X) = c(\tilde{\alpha})) = \mathbb{E}(1 - \eta(X))\mathbb{I}(\eta(X) \geq c(\tilde{\alpha})) > \alpha p^- \,. \tag{S79}$$

Now we are ready to prove linearity of $s(\alpha)$ when $\alpha \in I(\tilde{\alpha})$. Without loss of generality, we may assume that $\alpha > 0$. By using (S22) and the fact that

$$\mathbb{P}(X \in \mathcal{N}_\alpha) = \frac{\mathbb{E}(1 - \eta(X))(\alpha - \mathbb{I}(\eta(X) > c(\alpha)))}{1 - c(\alpha)} = \frac{\alpha p^- - \mathbb{E}(1 - \eta(X))\mathbb{I}(\eta(X) > c(\alpha))}{1 - c(\alpha)}, \tag{S80}$$

we have that

$$\begin{aligned}
s(\alpha) &= \frac{\mathbb{P}(R_\alpha^\star) - \alpha p^-}{p^+} = \frac{\mathbb{P}(\eta(X) > c(\alpha)) + \mathbb{P}(\mathcal{N}_\alpha)}{p^+} - \frac{\alpha p^-}{p^+} \\
&= \frac{\alpha p^- - \mathbb{E}(1 - \eta(X))\mathbb{I}(\eta(X) > c(\alpha))}{p^+(1 - c(\alpha))} + \frac{\mathbb{P}(\eta(X) > c(\alpha)) - \alpha p^-}{p^+} \\
&= \frac{c(\tilde{\alpha})p^-}{p^+(1 - c(\tilde{\alpha}))}\alpha + \frac{\mathbb{E}(\eta(X) - c(\tilde{\alpha}))\mathbb{I}(\eta(X) > c(\tilde{\alpha}))}{p^+(1 - c(\tilde{\alpha}))},
\end{aligned} \tag{S81}$$

where we have used the fact that $c(\alpha) = c(\tilde{\alpha})$ for any $\alpha \in I(\tilde{\alpha})$. This proves the linearity of $s(\alpha)$ over $I(\tilde{\alpha})$ when $\mathbb{P}(\eta(X) = c(\tilde{\alpha})) > 0$ for some $c(\tilde{\alpha})) < 1$.

Conversely, suppose that $s(\alpha)$ is linear in $\delta_{\tilde{\alpha}}$. Then, $s(\alpha)$ must be differentiable. Moreover, by Proposition S2, it follows that its derivative over $\delta_{\tilde{\alpha}}$ is $s'(\alpha) = s'_+(\alpha) = \frac{p^- c(\alpha)}{p^+(1-c(\alpha))}$, which must be a constant function over the interval $\delta_{\tilde{\alpha}}$. Hence, $c(\alpha)$ is a constant over the interval $\delta_{\tilde{\alpha}}$. Then for any $\alpha_1, \alpha_2 \in \delta_{\tilde{\alpha}}$ with $\alpha_1 < \alpha_2$, by using (ii) of Lemma S3, we have that

$$\mathbb{E}(1 - \eta(X))\mathbb{I}(\eta(X) > c(\alpha_1)) \leq \alpha_1 p^- \text{ and } \mathbb{E}(1 - \eta(X))\mathbb{I}(\eta(X) \geq c(\alpha_2)) \geq \alpha_2 p^- . \tag{S82}$$

Combining these two equations with the fact that $c(\alpha_1) = c(\alpha_2) = c(\tilde{\alpha})$, we obtain that

$$
\begin{aligned}
(1 - c(\tilde{\alpha}))\mathbb{P}(\eta(X) = c(\tilde{\alpha})) &= \mathbb{E}(1 - \eta(X))\mathbb{I}(\eta(X) \geq c(\tilde{\alpha})) - \mathbb{E}(1 - \eta(X))\mathbb{I}(\eta(X) > c(\tilde{\alpha})) \\
&= \mathbb{E}(1 - \eta(X))\mathbb{I}(\eta(X) \geq c(\alpha_2)) - \mathbb{E}(1 - \eta(X))\mathbb{I}(\eta(X) > c(\alpha_1)) \\
&\geq (\alpha_2 - \alpha_1)p^- > 0 ,
\end{aligned}
$$

which proves that $\mathbb{P}(\eta(X) = c(\tilde{\alpha})) > 0$. This completes the proof.

**Proof of Proposition S2.** We first show that (i) $s(\alpha)$ is differentiable at $\alpha$ for $0 < \alpha < 1 - \frac{\mathbb{P}(\eta(X)=0)}{p^-}$ if and only if $c(\alpha)$ is continuous at $\alpha$; and (ii) $s(\alpha)$ is left-differentiable at $\alpha = 1 - \frac{\mathbb{P}(\eta(X)=0)}{p^-}$. Then we show that (iii) $s(\alpha)$ is right-differentiable at $\alpha = 0$ if and only if $c(0) < 1$.

To prove (i) and (ii), we first derive the left and right derivative of $s(\alpha)$ over $(0, 1 - \frac{\mathbb{P}(\eta(X)=0)}{p^-}]$, denoted as $s'_+(\alpha)$ and $s'_-(\alpha)$, respectively. First, using (S29), we obtain that for any $1 - \frac{\mathbb{P}(\eta(X)=0)}{p^-} \geq \tilde{\alpha} > \alpha > 0$,

$$(\tilde{\alpha} - \alpha)\frac{p^- c(\tilde{\alpha})}{p^+(1-c(\tilde{\alpha}))} \leq s(\tilde{\alpha}) - s(\alpha) \leq (\tilde{\alpha} - \alpha)\frac{p^- c(\alpha)}{p^+(1 - c(\alpha))} . \tag{S83}$$

Letting $\tilde{\alpha} \to \alpha$ and using the right continuity of $c(\alpha)$ (c.f. Lemma S3), we obtain that for any $\alpha \in (0, 1 - \frac{\mathbb{P}(\eta(X)=0)}{p^-})$, the right derivative of $s(\alpha)$ is

$$s'_+(\alpha) = \frac{p^- c(\alpha)}{p^+(1 - c(\alpha))} \tag{S84}$$

Next, we derive the left derivative of $s(\alpha)$ over $(0, 1 - \frac{\mathbb{P}(\eta(X)=0)}{p^-}]$. From (S29), we know that for

any $0 < \tilde{\alpha} < \alpha$,

$$(\alpha - \tilde{\alpha}) \frac{p^- c(\alpha)}{p^+ (1 - c(\alpha))} \leq s(\alpha) - s(\tilde{\alpha}) \leq (\alpha - \tilde{\alpha}) \frac{p^- c(\tilde{\alpha})}{p^+ (1 - c(\tilde{\alpha}))}.$$

Now we consider two cases: (i) $c(\alpha)$ is continuous at $\alpha$; and (ii) $c(\alpha)$ is discontinuous at $\alpha$. When $c(\alpha)$ is continuous at $\alpha$, we have that $\lim_{\tilde{\alpha} \to \alpha^-} \frac{p^- c(\tilde{\alpha})}{p^+ (1 - c(\tilde{\alpha}))} = \frac{p^- c(\alpha)}{p^+ (1 - c(\alpha))}$ and $s'_-(\alpha) = \frac{p^- c(\alpha)}{p^+ (1 - c(\alpha))}$. Hence, in this case, we have that $s'_-(\alpha)$ exists for $\alpha \in (0, 1 - \frac{\mathbb{P}(\eta(X) = 0)}{p^-}]$ and $c(\alpha)$ is continuous at $\alpha$.

When $c(\alpha)$ is discontinuous at $\alpha$, then by the proof of (iii) in Lemma S3, we know that $g(c(\alpha)) = \alpha p^-$ and $\mathbb{P}(c(\alpha) < \eta(X) < \bar{c}) = 0$, where $\bar{c} = \lim_{\tilde{\alpha} \to \alpha^-} c(\tilde{\alpha})$. This further implies that $\mathbb{P}(X \in \mathcal{N}_\alpha) = \frac{\alpha p^- - g(c(\alpha))}{1 - c(\alpha)} = 0$. By (S22) and Theorem 2, we know that for any $\alpha > 0$, $p^+ s(\alpha) = \mathbb{P}(X \in R^\star_\alpha) - \alpha p^-$ with $R^\star_\alpha = \{x : \eta(x) > c(\alpha)\} \cup \mathcal{N}_\alpha$ and $\mathbb{P}(X \in \mathcal{N}_\alpha) = \frac{\alpha p^- - g(c(\alpha))}{1 - c(\alpha)}$. For any $0 < \tilde{\alpha} < \alpha$, we have that $c(\tilde{\alpha}) \geq \bar{c} > c(\alpha)$ due to the discontinuity of $c(\alpha)$. Therefore,

$$
\begin{aligned}
p^+ (s(\alpha) - s(\tilde{\alpha})) &= \mathbb{P}(X \in R^\star_\alpha) - \alpha p^- - \mathbb{P}(X \in R^\star_{\tilde{\alpha}}) + \tilde{\alpha} p^- \\
&= \mathbb{P}(\eta(X) > c(\alpha)) + \mathbb{P}(X \in \mathcal{N}_\alpha) - \mathbb{P}(\eta(X) > c(\tilde{\alpha})) - \mathbb{P}(X \in \mathcal{N}_{\tilde{\alpha}}) - p^-(\alpha - \tilde{\alpha}) \\
&= \mathbb{P}(c(\alpha) < \eta(X) \leq c(\tilde{\alpha})) - \frac{\tilde{\alpha} p^- - g(c(\tilde{\alpha}))}{1 - c(\tilde{\alpha})} - p^-(\alpha - \tilde{\alpha}) \\
&= \mathbb{P}(\bar{c} \leq \eta(X) \leq c(\tilde{\alpha})) - \frac{\tilde{\alpha} p^- - g(c(\tilde{\alpha}))}{1 - c(\tilde{\alpha})} - p^-(\alpha - \tilde{\alpha}), \quad\quad \text{(S85)}
\end{aligned}
$$

where we have used the fact that $\mathbb{P}(X \in \mathcal{N}_\alpha) = 0$ and $\mathbb{P}(c(\alpha) < \eta(X) < \bar{c}) = 0$. Moreover, note that

$$
\begin{aligned}
g(\bar{c}) &= \mathbb{E}(1 - \eta(X))\mathbb{I}(\eta(X) > \bar{c}) = \mathbb{E}(1 - \eta(X))\mathbb{I}(\eta(X) > c(\alpha)) - (1 - \bar{c})\mathbb{P}(\eta(X) = \bar{c}) \\
&= \alpha p^- - (1 - \bar{c})\mathbb{P}(\eta(X) = \bar{c}). \quad\quad \text{(S86)}
\end{aligned}
$$

where we have used the fact that $\mathbb{E}(1 - \eta(X))\mathbb{I}(\eta(X) > c(\alpha)) = g(c(\alpha)) = \alpha p^-$ and $\mathbb{P}(c(\alpha) < \eta(X) <$

$\bar{c}) = 0$. Using this, we obtain that

$$
\begin{aligned}
g(c(\tilde{\alpha})) &= g(\bar{c}) - (g(\bar{c}) - g(c(\tilde{\alpha}))) \\
&= \alpha p^- - (1 - \bar{c})\mathbb{P}(\eta(X) = \bar{c}) - \mathbb{E}(1 - \eta(X))\mathbb{I}(\bar{c} < \eta(X) \le c(\tilde{\alpha}))
\end{aligned}
$$

Substituting this into (S85), we obtain that

$$
\begin{aligned}
&p^+(s(\alpha) - s(\tilde{\alpha})) \\
&= \frac{(\alpha - \tilde{\alpha})p^- c(\tilde{\alpha})}{1 - c(\tilde{\alpha})} + \frac{\bar{c} - c(\tilde{\alpha})}{1 - c(\tilde{\alpha})}\mathbb{P}(\eta(X) = \bar{c}) + \frac{\mathbb{E}(\eta(X) - c(\tilde{\alpha}))\mathbb{I}(\bar{c} < \eta(X) \le c(\tilde{\alpha}))}{1 - c(\tilde{\alpha})} \\
&= \frac{(\alpha - \tilde{\alpha})p^- c(\tilde{\alpha})}{1 - c(\tilde{\alpha})} + \frac{\bar{c} - c(\tilde{\alpha})}{1 - c(\tilde{\alpha})}\mathbb{P}(\eta(X) = \bar{c}) + \frac{\mathbb{E}(\eta(X) - c(\tilde{\alpha}))\mathbb{I}(\bar{c} < \eta(X) < c(\tilde{\alpha}))}{1 - c(\tilde{\alpha})}, \quad \text{(S87)}
\end{aligned}
$$

Now we derive $s'_-(\alpha)$ by considering two scenarios. First, if $c(\tilde{\alpha}) = \bar{c}$ for some $0 < \tilde{\alpha} < \alpha$, then the last two terms in (S87) vanishes, and we have that $s'_-(\alpha) = \lim_{\tilde{\alpha} \to \alpha^-} \frac{p^- c(\tilde{\alpha})}{p^+(1 - c(\tilde{\alpha}))} = \frac{p^- \bar{c}}{p^+(1 - \bar{c})}$.

Second, if $c(\tilde{\alpha}) > \bar{c}$ for all $0 < \tilde{\alpha} < \alpha$, then by definition of $c(\tilde{\alpha})$, we must have $g(\bar{c}) > \tilde{\alpha}p^-$ for all $\tilde{\alpha} < \alpha$, which implies that $g(\bar{c}) \ge \alpha p^-$. This together with (S86), implies that $\mathbb{P}(\eta(X) = \bar{c}) \le 0$, or equivalently $\mathbb{P}(\eta(X) = \bar{c}) = 0$. This implies that the second term in (S87) vanishes.

Now we bound the third term in (S87). Note that

$$
\left| \frac{\mathbb{E}(\eta(X) - c(\tilde{\alpha}))\mathbb{I}(\bar{c} < \eta(X) < c(\tilde{\alpha}))}{1 - c(\tilde{\alpha})} \right| \le \frac{(c(\tilde{\alpha}) - \bar{c})\mathbb{P}(\bar{c} < \eta(X) < c(\tilde{\alpha}))}{1 - c(\tilde{\alpha})}. \quad \text{(S88)}
$$

Next we bound $\mathbb{P}(\bar{c} < \eta(X) < c(\tilde{\alpha}))$. Using (S86), $\mathbb{P}(\eta(X) = \bar{c}) = 0$ and the definition of $c(\tilde{\alpha})$, we have that $g(\bar{c}) = \alpha p^-$ and $g(c) > \tilde{\alpha}p^-$ for any $c \in (\bar{c}, c(\tilde{\alpha}))$. This implies that $g(c) > \tilde{\alpha}p^- = \alpha p^- + (\tilde{\alpha} - \alpha)p^- = g(\bar{c}) + (\tilde{\alpha} - \alpha)p^-$. Therefore, $(\alpha - \tilde{\alpha})p^- > g(\bar{c}) - g(c) = \mathbb{E}(1 - \eta(X))\mathbb{I}(\bar{c} < \eta(X) \le c) \ge (1 - c)\mathbb{P}(\bar{c} < \eta(X) \le c)$. Letting $c \to c(\tilde{\alpha})$, we obtain that

$$
\mathbb{P}(\bar{c} < \eta(X) < c(\tilde{\alpha})) = \lim_{c \to c(\tilde{\alpha})^-} \mathbb{P}(\bar{c} < \eta(X) \le c) \le \lim_{c \to c(\tilde{\alpha})^-} \frac{(\alpha - \tilde{\alpha})p^-}{1 - c} = \frac{(\alpha - \tilde{\alpha})p^-}{1 - c(\tilde{\alpha})}
$$

Combining this with (S88), we obtain that the absolute value of the third term in (S87) is bounded

by $\frac{(\alpha - \tilde{\alpha})(c(\tilde{\alpha}) - \bar{c})p^-}{(1 - c(\tilde{\alpha}))^2}$, which further implies that

$$\left| s'_-(\alpha) - \frac{p^- \bar{c}}{p^+(1 - \bar{c})} \right| \leq \left| \lim_{\tilde{\alpha} \to \alpha-} \frac{p^- c(\tilde{\alpha})}{p^+(1 - c(\tilde{\alpha}))} - \frac{p^- \bar{c}}{p^+(1 - \bar{c})} \right| + \left| \lim_{\tilde{\alpha} \to \alpha-} \frac{(c(\tilde{\alpha}) - \bar{c})p^-}{(1 - c(\tilde{\alpha}))^2} \right| = 0 \,.$$

Hence, in this case, we have that $s'_-(\alpha) = \frac{p^- \bar{c}}{p^+(1 - \bar{c})}$ for $\alpha \in (0, 1 - \frac{\mathbb{P}(\eta(X)=0)}{p^-}]$ and $c(\alpha)$ is discontinuous at $\alpha$.

Consequently, we have shown that in all scenarios that, for any $\alpha \in (0, 1 - \frac{\mathbb{P}(\eta(X)=0)}{p^-}]$,

$$s'_-(\alpha) = \frac{p^- \bar{c}}{p^+(1 - \bar{c})} = \lim_{\tilde{\alpha} \to \alpha-} \frac{p^- c(\tilde{\alpha})}{p^+(1 - c(\tilde{\alpha}))} \tag{S89}$$

where $\bar{c} = \lim_{\tilde{\alpha} \to \alpha-} c(\tilde{\alpha})$. This completes the proof of case (ii) that $s(\alpha)$ is left-differentiable at $\alpha = 1 - \frac{\mathbb{P}(\eta(X)=0)}{p^-}$. Combining this with (S84) and the fact that $c(\alpha)$ is right continuous, we know that $s(\alpha)$ is differentiable over $(0, 1 - \frac{\mathbb{P}(\eta(X)=0)}{p^-})$, if and only if $s'_-(\alpha) = s'_+(\alpha)$, which is then equivalent to the fact that $c(\alpha)$ is continuous at $\alpha$. This then proves case (i) that $s(\alpha)$ is differentiable if and only if $c(\alpha)$ is continuous for any $0 < \alpha < 1 - \frac{\mathbb{P}(\eta(X)=0)}{p^-}$.

Finally, we prove (iii). Recall that we need to show that $s(\alpha)$ is right-differentiable at $\alpha = 0$ if and only if $c(0) < 1$. By Theorem 2, we have that $R^\star_\alpha = \{x : \eta(x) > c(\alpha)\} \cup \mathcal{N}_\alpha$ when $\alpha > 0$ with $c(\alpha) < 1$, which further implies that $R^\star_0 \subseteq R^\star_\alpha$ for any $\alpha > 0$. Combining this with (7) we obtain that

$$
\begin{aligned}
p^+(s(\alpha) - s(0)) &= \mathbb{E}\eta(X)\mathbb{I}(X \in R^\star_\alpha) - \mathbb{E}\eta(X)\mathbb{I}(X \in R^\star_0) = \mathbb{E}\eta(X)\mathbb{I}(X \in R^\star_\alpha \setminus R^\star_0) \\
&= \mathbb{E}\eta(X)\mathbb{I}(c(\alpha) < \eta(X) < 1) + c(\alpha)\mathbb{P}(X \in \mathcal{N}_\alpha) \\
&= \mathbb{E}\eta(X)\mathbb{I}(c(\alpha) < \eta(X) < 1) + \frac{c(\alpha)}{1 - c(\alpha)}\mathbb{E}(1 - \eta(X))(\alpha - \mathbb{I}(\eta(X) > c(\alpha))) \\
&= \frac{\alpha p^- c(\alpha)}{1 - c(\alpha)} + \frac{\mathbb{E}\eta(X)\mathbb{I}(c(\alpha) < \eta(X) < 1)}{1 - c(\alpha)} - \frac{c(\alpha)}{1 - c(\alpha)}\mathbb{P}(c(\alpha) < \eta(X) < 1) \\
&= \frac{\alpha p^- c(\alpha)}{1 - c(\alpha)} + \frac{\mathbb{E}(\eta(X) - c(\alpha))\mathbb{I}(c(\alpha) < \eta(X) < 1)}{1 - c(\alpha)} \,,
\end{aligned}
$$

which implies that

$$\frac{s(\alpha) - s(0)}{\alpha} = \frac{p^- c(\alpha)}{p^+(1 - c(\alpha))} + \frac{\mathbb{E}(\eta(X) - c(\alpha))\mathbb{I}(c(\alpha) < \eta(X) < 1)}{p^+(1 - c(\alpha))\alpha} \tag{S90}$$

By (iii) in Lemma S3, we have $c(0) = \lim_{\alpha \to 0^+} c(\alpha)$. When $c(0) = 1$, we have that

$$\lim_{\alpha \to 0^+} \frac{s(\alpha) - s(0)}{\alpha} = \infty. \tag{S91}$$

Next, we prove that that when $c(0) < 1$, we have

$$\lim_{\alpha \to 0^+} \frac{s(\alpha) - s(0)}{\alpha} = \frac{p^- c(0)}{p^+(1 - c(0))}. \tag{S92}$$

Sufficiently, in view of (S90), we only need to show $\lim_{\alpha \to 0^+} \frac{1}{\alpha}\mathbb{E}(\eta(X) - c(\alpha))\mathbb{I}(c(\alpha) < \eta(X) < 1) = 0$ when $c(0) < 1$. To this end, we consider two cases: (i) there exists $\tilde{\alpha} > 0$ such that $c(\alpha) = c(0)$ for $0 < \alpha < \tilde{\alpha}$; and (ii) $c(\alpha) < c(0)$, for any $\alpha > 0$. For the first case, by part (ii) of Lemma S3, we have that $g(c(\alpha)) = \mathbb{E}(1 - \eta(X))\mathbb{I}(\eta(X) > c(\alpha)) \le \alpha p^-$ for $\alpha \in [0, 1 - \frac{\mathbb{P}(\eta(X)=0)}{p^-}]$. Note $c(\alpha) = c(0)$ for $0 < \alpha < \tilde{\alpha}$, we obtain $g(c(\alpha)) = g(c(0)) = 0$ for $0 < \alpha < \tilde{\alpha}$. This combining with $c(0) < 1$, implies that $\mathbb{P}(c(\alpha) < \eta(X) < 1) = 0$ for $0 < \alpha < \tilde{\alpha}$. Hence we have that $\frac{1}{\alpha}\mathbb{E}(\eta(X) - c(\alpha))\mathbb{I}(c(\alpha) < \eta(X) < 1) = 0$ for $0 < \alpha < \tilde{\alpha}$. Hence, in the first case, we have $\lim_{\alpha \to 0^+} \frac{1}{\alpha}\mathbb{E}(\eta(X) - c(\alpha))\mathbb{I}(c(\alpha) < \eta(X) < 1) = 0$.

For the second case, using part (ii) in Lemma S3, we know that $g(c(\alpha)) \le \alpha p^-$. Letting $\alpha \to 0$ and using the fact that $\lim_{\alpha \to 0^+} c(\alpha) = c(0)$, we obtain that

$$0 = \lim_{\alpha \to 0} \alpha p^- \ge \lim_{\alpha \to 0^+} g(c(\alpha)) = \mathbb{E}(1 - \eta(X))\mathbb{I}(\eta(X) \ge c(0)) = g(c(0)) + (1 - c(0))\mathbb{P}(\eta(X) = c(0)),$$

which further implies that $g(c(0)) = 0$ and $\mathbb{P}(\eta(X) = c(0)) = 0$, where we have used the fact that

$c(0) < 1$. Note that $g(c(0)) = 0$ also implies that $\mathbb{P}(c(0) < \eta(X) < 1) = 0$. As a result, we have

$$
\begin{aligned}
\mathbb{E}(\eta(X) - c(\alpha))\mathbb{I}(c(\alpha) < \eta(X) < 1) &= \mathbb{E}(\eta(X) - c(\alpha))\mathbb{I}(c(\alpha) < \eta(X) < c(0)) \\
&\leq (c(0) - c(\alpha))\mathbb{P}(c(\alpha) < \eta(X) < c(0)), \qquad \text{(S93)} \\
\alpha p^- \geq g(c(\alpha)) = g(c(\alpha)) - g(c(0)) &= \mathbb{E}(1 - \eta(X))\mathbb{I}(c(\alpha) < \eta(X) \leq c(0)) \\
&\geq (1 - c(0))\mathbb{P}(c(\alpha) < \eta(X) \leq c(0)) \\
&= (1 - c(0))\mathbb{P}(c(\alpha) < \eta(X) < c(0)),
\end{aligned}
$$

which implies that $\mathbb{P}(c(\alpha) < \eta(X) < c(0)) \leq \frac{\alpha p^-}{1 - c(0)}$. Combining this with (S93), we get

$$
\begin{aligned}
0 &\leq \frac{\mathbb{E}(\eta(X) - c(\alpha))\mathbb{I}(c(\alpha) < \eta(X) < 1)}{\alpha} \leq \frac{(c(0) - c(\alpha))\mathbb{P}(c(\alpha) < \eta(X) < c(0))}{\alpha} \\
&\leq \frac{p^-}{1 - c(0)}(c(0) - c(\alpha)),
\end{aligned}
$$

which implies that $\lim_{\alpha \to 0^+} \mathbb{E}(\eta(X) - c(\alpha))\mathbb{I}(c(\alpha) < \eta(X) < 1)/\alpha = 0$. This completes the proof of (S92).

Combining (S91) and (S92), it follows that $s(\alpha)$ is right-differentiable at 0 if and only if $c(0) < 1$, which proves case (iii). This completes the proof.

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

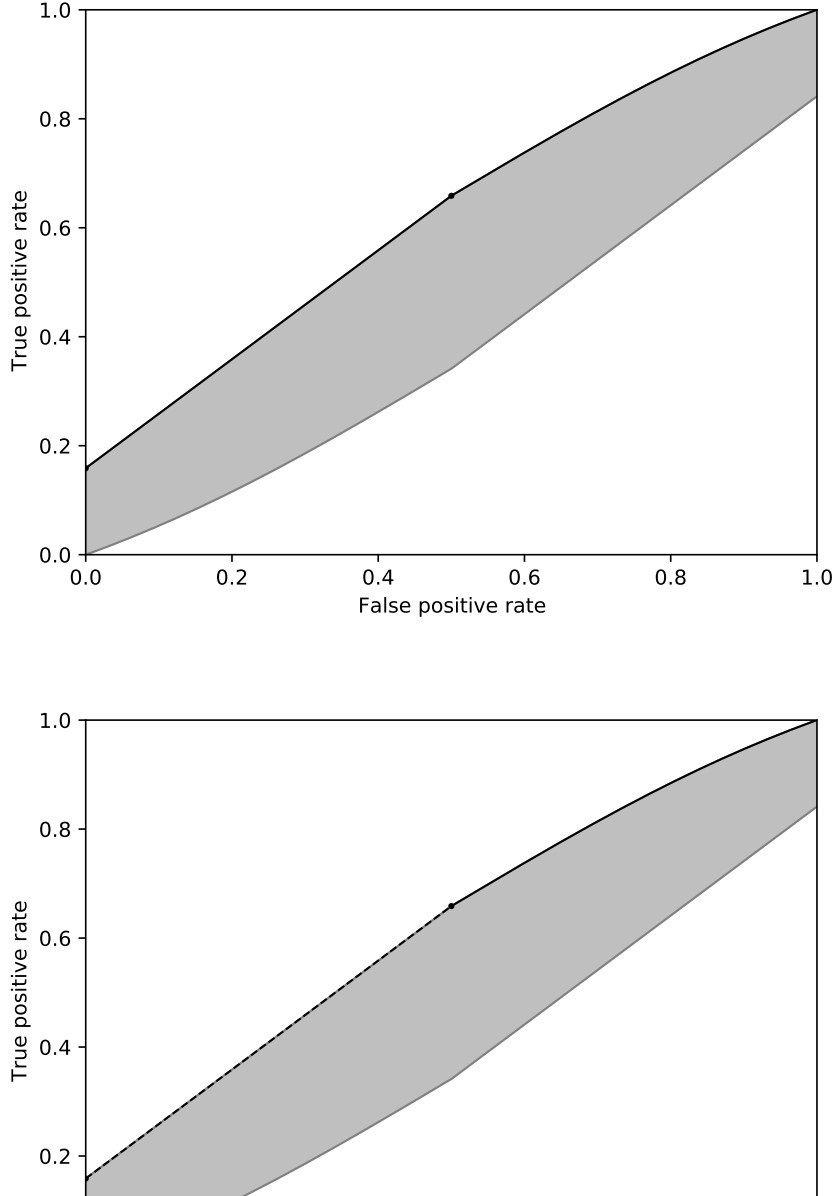

Figure S2: Examples of population ROC curve using weighted method. The solid curve in the top panel is the optimal ROC curve. The dashed straight line in the bottom panel is the linear piece that can not be recovered by the weighted method directly. However, after connecting the points linearly, the optimal ROC curve can be recovered.

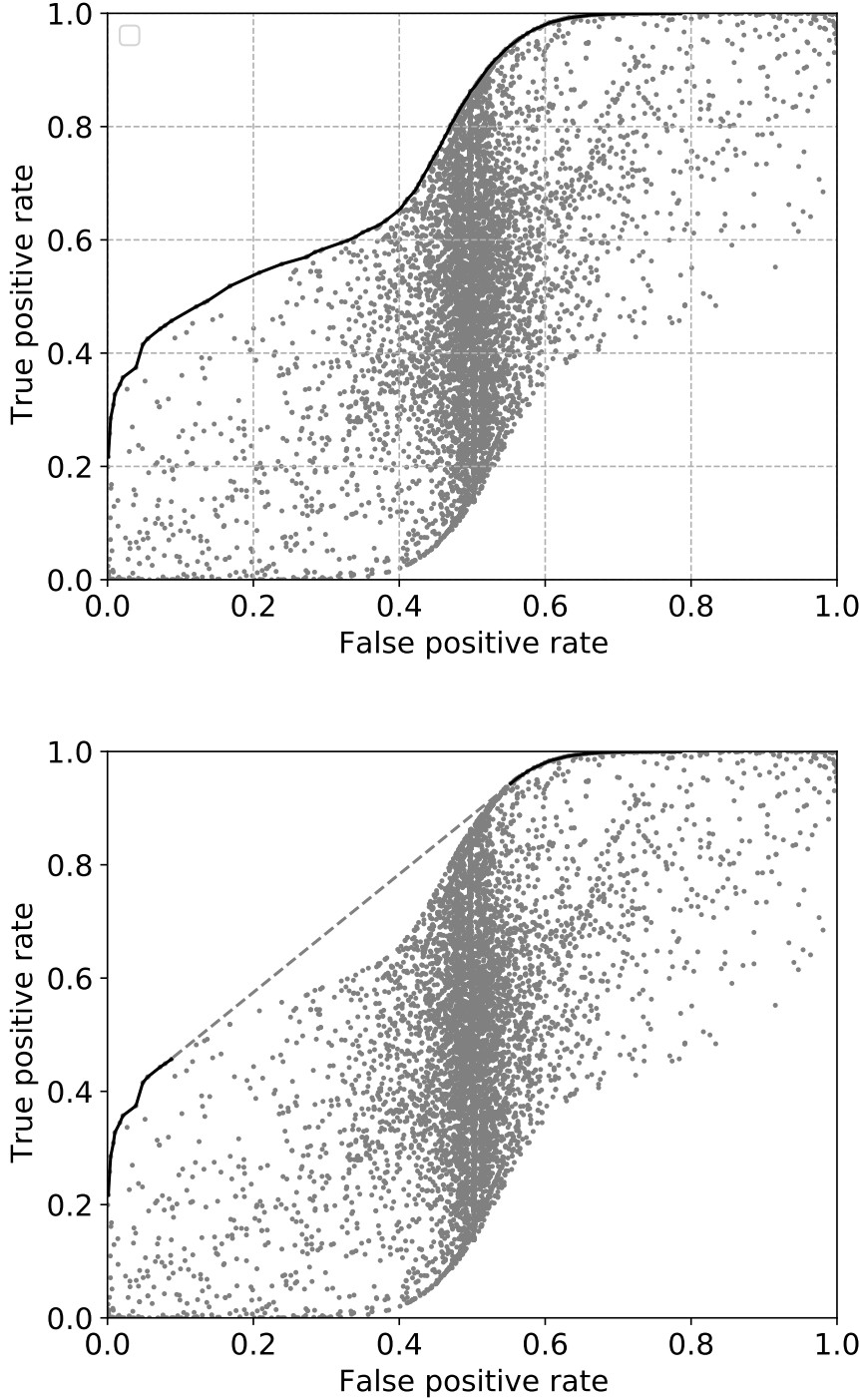

Figure S3: Population ROC curve of the constrained method (solid curve in the top panel) and population ROC curve of the weighted method (solid curve in the bottom panel). The set of grey dots in both panels denote an approximation to the set of all possible FPR-TPR pairs.

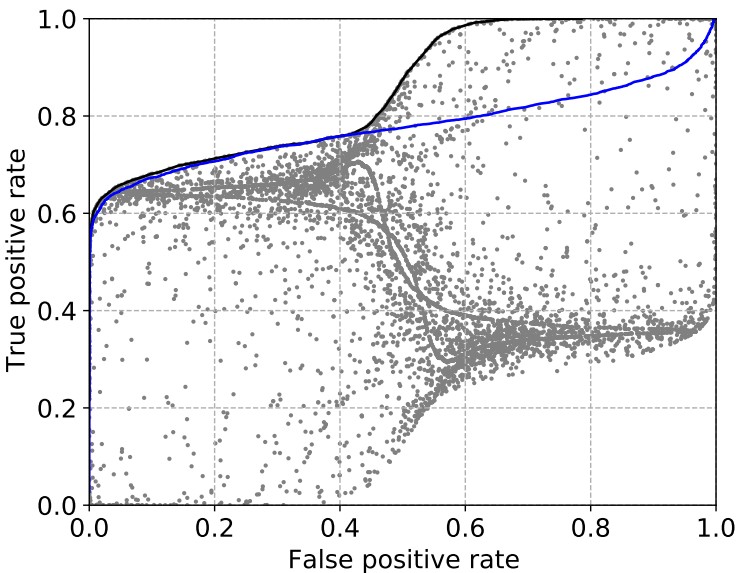

Figure S4: Population ROC curve (blue curve) of the weighted method under surrogate loss. The grey area represents an approximation of all possible FPR-TPR pairs.

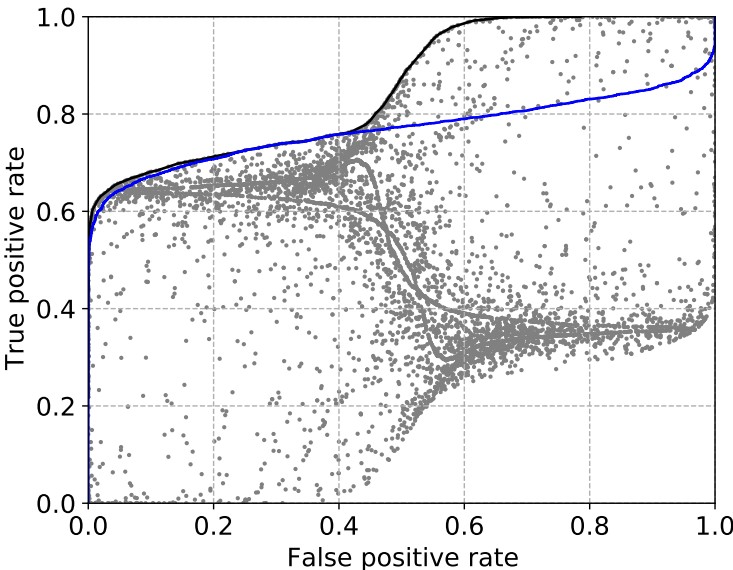

Figure S5: Population ROC curve (blue curve) of the cutoff method. The grey area represents an approximation of all possible FPR-TPR pairs.