# OpenReview forum: "On the consistent estimation of optimal Receiver Operating Characteristic (ROC) curve"
_NeurIPS.cc/2022/Conference — NeurIPS 2022 Accept_

### Official Review · Reviewer_gfsm · 2022-07-08

**Rating:** 6
**Confidence:** 4
**Soundness:** 3 good
**Presentation:** 4 excellent
**Contribution:** 3 good

**Summary:**

This theoretical paper studied three popular ROC curve estimation methods (i.e. Neyman-Pearson classification,  weighted method, and cutoff method) and then analyzed at the population level under both correct and incorrect model specifications.  In particular, the authors showed that all three methods are consistent in the sense that they all target the optimal ROC curve under 0/1 loss or some surrogate loss under correct model specification. Such results can be regarded as an extension of the classical consistency results from binary classification to ROC estimation methods.  The introduced definition of the optimal ROC curve extended the definition in [Scott (2007)] which has much less assumptioin.

**Questions:**

I have no time to go through the proofs step by step. I am wondering what is the main novelty in the proof techniques in addition to the statements about the new results obtained and the relaxed conditions.  For instance, what are the main reasons that you can relax the continuous conditions made in Scott's paper?

**Limitations:**

The authors have not addressed the potential limitation of the work.  it would be nice to see what are the remain questions for future work.

**Strengths And Weaknesses:**

Pros;

1. It seems that this work is the first-ever-known study to investigate theoretical conditions under which the most popular three methods can recover the optimal ROC optimal curve in the statistical decision framework.

2. The proposed definition of the optimal ROC curve using the Pareto frontier is new and has played a central role in the paper.

3.  Some conditions in the literature [Van Trees (2004) and Scott (2007)] are significantly relaxed and extended in the paper.

Cons:

The proofs provided in the supplementary materials are very technical and lengthy. My main question is what is the main novelty in the proof techniques compared to those in the literature.

I have read the authors' reponse and my score remains the same.

---

> ### Author Response · Authors · 2022-08-02
> **Response to Reviewer gfsm**
>
> Thanks for your thorough review and valuable comments.
> For your questions on proof techniques,
> our responses are summarized below.
>
> - The proof techniques in existing literature rely heavily on
> the assumption that $\eta(X)=\mathbb P(Y=1\mid X)$ has a continuous distribution.
> By contrast,
> we only assume that the distribution of $X$ is continuous, which makes
> the old proof techniques no longer applicable in this more general setting.
> For example, under the old assumption that $\eta(X)=\mathbb P(Y=1\mid X)$
>  has continuous distribution,
>  we do not need to address the case that $\mathbb P(\eta(X)=c) > 0$
>  for some $c$.
>  But this could happen under the less stringent assumption that $X$
>  has continuous distribution.
>  The new proof techniques are designed partily to address these challenges.
> - As an example, consider a case where the two conditional distributions of $X$ given $Y=1$ and  $Y=-1$
>     have  different support, that is, there exists a region $D$ such that $f(x|Y=1)/f(x|Y=-1) = 0$ or
>     $\infty$ for $x\in D$. In this case, $\eta(X)$ is no longer continuous
>     but the distribution of $X$ can still be continuous.
>     This situation can happen when the two classes are separable.
>
> Following your suggestion,
> we have also added some potential limitation of our work and gave
> some future directions to explore further.

---

### Official Review · Reviewer_8S33 · 2022-07-11

**Rating:** 6
**Confidence:** 3
**Soundness:** 3 good
**Presentation:** 3 good
**Contribution:** 3 good

**Summary:**

The authors present a definition of 'optimal ROC curve' which is applicable under model misspecification, i.e., when the model class does not contain the target function.  Three different supervised learning approaches for producing a set of fitted functions generating an ROC curve are analyzed in the limit of infinite training data (i.e. at the "population level"): the constrained (Neyman-Pearson) method of maximizing true positive rate subject to an upper bound on the false positive rate, the risk minimization/cost optimization approach in which false positive and false negative errors are weighted differently, and a thresholding (a.k.a. ."cutoff" method) in which one real-valued-output discriminant function is learned and an ROC curve is generated by varying the threshold above which one class is predicted instead of the other. If the model class contains all measurable classifiers, then all 3 methods are shown to be consistent in the sense of converging on the optimal ROC curve, although the cutoff method requires a stronger set of assumptions on the loss function used. If there is model misspecification, however, the authors show that only the constrained method is consistent in estimating the optimal ROC curve. Some simulations are presented where the model class is linear but the target function is either linear (to illustrate the case where the model is correctly specified) or quadratic (to illustrate model misspecification).

**Questions:**

Does the cutoff method analysis cover the case of cross entropy? Maybe it does and I am not seeing it, but it is not clear to me.

Are you sure you want to publish this in NeurIPs rather than a journal? Wouldn't you rather be able to include the proofs as part of the paper itself?

**Limitations:**

I don't see any concerns regarding negative societal impacts.

**Strengths And Weaknesses:**

The results are original to the best of my knowledge.

Assuming correctness, I do believe the result that only the constrained method is consistent under model misspecification is a significant result with practical implications (i.e. that the constrained method should be used unless using nonparametric methods).

However, I have some concerns about whether the cutoff method result is derived under realistic assumptions.  The assumption is made that a discriminant function f(x) is chosen to minimize a weighted expectation of  V( Y f(x) ), where V is a surrogate loss function and Y is the binary class (1 or -1).  It is not clear to me whether this construction covers the most likely real world scenario for binary classification: logistic regression, where f(x) is chosen to minimize cross-entropy (sometimes described as log likelihood). Here f(x) would range between 0 and 1 and represent the probability of class 1 and we would be minimizing -log(f(x)) if Y=1, -log(1-f(x)) if Y=-1.  I might be mistaken, but it is not clear to me whether assuming the loss surrogate V is a function of Y f(x) covers the cross entropy case.

Regarding clarity and presentation, my main concern is that NeurIPs might not be the ideal venue for this work, given that the supplemental contains 59 pages (!!!) of mostly proofs.  I wonder whether it would be better to publish this work in a journal where the reader has access to the full proofs.

I also had one very specific presentation complaint. The indicator function notation used (for instance) on line 157 in s(alpha):

 ...indicator(eta(x) > c(alpha))

...is not the standard notation for indicator function. Maybe this notation is used more than I realize, but I am much more familiar with indicator being represented as a fancy 1 with 2 vertical lines...

https://ewens.caltech.edu/public-code-data/indicator-function-latex

...or simply with I().  I would suggest at least adding a note explaining that the symbol you used represents the indicator function. It took me a while to figure out what that symbol meant, and I had to ask the area chair to confirm for me.

Here are some suggested edits to the writing:

Line 24 empirical risk minimization goes back a lot farther than Elkan 2001 with cost-sensitive learning e.g. Vapnik 1992 empirical risk minimization:
https://proceedings.neurips.cc/paper/1991/file/ff4d5fbbafdf976cfdc032e3bde78de5-Paper.pdf
...and probably further back than that.

Line 27 another type of methods ->. another type of method

31 the set of classifier -> the set of classifiers

Line 34 what is “population level”? I deduced that it is the N->infinity training data limit, but it might be helpful to explain that.

Line 49 all three methods targets -> all three methods target

Line 58 "when simple classifers (linear) are used"….I actually think the result is more broadly applicable...most ML situations will have at least some model misspecification, so suggesting that it's only useful in the linear case is underselling the result …in any practical application the model probably space does not contain the perfect model

Line 63 or mostly focuses - > or mostly focus
Line 134 set of classifier -> set of classifiers
Line 185 of special class -> of special classes
Line 204 the set FPR-TPR pairs -> the set of FPR-TPR pairs

---

> ### Author Response · Authors · 2022-08-02
> **Response to Reviewer 8S33**
>
> Thanks for your valuable comments and suggestions.
> Below are our responses.
>
> - Our cutoff method analysis can cover cross entropy loss.
> Indeed, logistic regression corresponds to the choice of
> surrogate loss $V(Yf(x))=\log(1+\exp(-Yf(X)))$ in our setup.
> To see this more clearly, we write the probability
> of $Y=1$ given feature $X=x$ as
> $\mathbb P(Y=1\mid X=x)=\eta(x)=1/(1+\exp(-f(x)))$,
> where $f(x)$ is a discriminant function (e.g., a linear function).
> Then we have
> $V(f(X))=\log(1+\exp(-f(X)))=-\log(\eta(X))$ if $Y=1$
> and
> $V(-f(X))=\log(1+\exp(f(X)))=-\log(1-\eta(X))$ if $Y=-1$.
> Maybe the confusion is caused by the notations in that in our setup we use
> notation $f(x)$ to refer to a discriminant function in
> $V(Yf(x))$, while in your understanding $f(x)$ is the estimated probability which we will use $\eta(x)$ instead in our setup.
> - Thanks to the suggestions on notation of indicator function,
> we have added a description on line 157.
> - We have added the suggested literature and fixed the typos in the article.

---

> > ### Comment · Reviewer_8S33 · 2022-08-03
> > **Thanks for the responses**
> >
> > I thank the authors for their response. It is reassuring to know that cross-entropy is included in their result.
> >
> > With that said, I choose to keep my score at a 6.  I would not mind seeing the paper accepted, but I don't feel comfortable taking a strong stance in favor of acceptance because it is hard to have very high confidence in the correctness given the 59 pages of proofs, far more than is reasonable to expect a reviewer to double-check in such a short time frame with several other papers to review at the same time.
> >
> > I realize that NeurIPS is very prestigious and an acceptance is highly desirable, so I understand the authors' choice to submit to NeurIPS. However, I still feel it might be better for this work to go through the review process at a journal, where reviewers have more time to double-check the proofs more thoroughly.

---

### Official Review · Reviewer_DSre · 2022-07-11

**Rating:** 7
**Confidence:** 2
**Soundness:** 4 excellent
**Presentation:** 4 excellent
**Contribution:** 4 excellent

**Summary:**

The problem of estimating ROC curves under possible model misspeciffication is
studied. A notion of optimal ROCs under arbitrary model spaces is introduced,
and three existing methods of estimating ROC curves are studied with respect to
this optimal curve. The optimal ROC is shown to coincide with existing
definitions when the model space includes all measurable classifiers. All three
estimation methods are shown to be consistent when correctly specified, however
only one estimation method (constrained/Neyman–Pearson) targets the optimal
curve under model misspecification.


**Questions:**

A significant portion of the theory handles seemingly pathological corner cases,
e.g., non-differentiable parts of the weighted method and pieces dealing
specifically with equalities. Would some mild assumptions allow for meaningful
simplifications of the theory?

There are no real-world experiments, how big a difference is there on real world
data?


**Limitations:**

The limitations of the theory are clear from the stated assumptions. No real
world experiments are a limitation that could be briefly discussed.


**Strengths And Weaknesses:**

The work is novel and the results broadly applicable as ROC analysis is widely
used and forms the basis for many methods. The presentation is good, writing
clear, and the novelty well placed amongst existing literature. No proofs are
presented in the main text, but the interpretation and general presentation of
the theory is excellent.

The proposed framework is shown to coincide with existing results, and indeed
generalises some results as there are no assumptions on the data generating
distribution.

The theory is lightly explored in simulation studies. As expected, the
simulation results align with the theory, but the main results figures are
presented in the supplementary materials as there is a lack of space in the main
text. There are no experiments on real-world data.

---

> ### Author Response · Authors · 2022-08-02
> **Response to Reviewer DSre**
>
> Thanks for your valuable comments and suggestions.
> We summarize our responses below:
> - As you have suggested, we can impose additional assumption that $\eta(X)$
> has continuous distribution, which will avoid the discussion of some scenarios we considered
> in our paper.
> We choose to make our results as general as possible by only assuming that the marginal distribution of $X$ is continuous.
> This allows our result to be applicable to some important scenarios.
> For example, if the conditional distributions of $X$ given $Y=1$ and $Y=-1$ have different supports,
> then $\eta(X)$ is no longer continuous but the distribution of $X$ can still be continuous.
> This could happen when the two class is perfectly separable.
>
> - Thanks for your suggestions on including some real data analysis.
> We have added one real data example and some further discussions on the results in Section 4.

---

### Official Review · Reviewer_hAdd · 2022-07-11

**Rating:** 7
**Confidence:** 3
**Soundness:** 3 good
**Presentation:** 3 good
**Contribution:** 3 good

**Summary:**

The authors present a more general notion of "optimal ROC curve" for binary classification. They show that their definition relies on weaker assumptions than predecessors when considering the set of all possible classifiers, and they extend the definition for general sets of classifiers. They also show sufficient conditions for three common ROC generating procedures to be consistent, under this definition of optimal ROC, when the model is correctly specified. Results on simulated data agree with the theory.

**Questions:**

Questions:

Could you clarify the assumption of continuous data distribution? I notice what appear to be some contradictions. Line 84: "However, our definition is quite general and does not make any assumptions on the data generating distribution," and a similar statement is made on line 150. However, line 104 states "Throughout, we make the assumption that the marginal distribution $X$, denoted as $\mathbb{P}_X$, is continuous..." and in some places you make the similar assumption that $\eta(X)$ is continuous.

Suggestions:

If possible, it would be beneficial to move some of the technical details from sections 2 and 3 to the supplementary material so that (assuming the brevity of other sections was a result of page limit) the motivation, context, and discussion can be improved. For example, many points in section 2 are relatively less significant in the context of prior work and may merit only a mention in the main paper with deeper analysis in the supplementary material; similarly, many key takeaways in section 3 can be stated while moving most of the bulky math to the supplementary material. Some figures in the main paper, both to aid understanding of concepts and to show the main results being discussed, might help too. If it were me, the priority would be:
1. placing the main experiment results in the main paper
2. making the discussion more thorough, including more detailed interpretation of experiment results, potential impacts, limitations resulting from assumptions, practical uses, future work
3. more clearly motivating the problem and its significance (why should someone using ROC curves for model evaluation care about these results?) in the introduction

Understanding would be greatly aided by making sure that all terms and symbols are defined. Some that would aid my understanding:
 - "model misspecification" in this context
 - "at the population level"
 - $\mathbb{I}(...)$ (I assumed this was the indicator function, but see places where I'm not sure that makes sense, such as when there is a random variable with no $\mathbb{P}$ or $\mathbb{E}$)
 - $p^+$ and $p^-$
 - $cl(...)$

Since the synthetic data being used is 2D, include a scatterplot showing the data and its labels. Putting it in the supplementary material is fine.

**Limitations:**

The authors state the assumptions that are necessary for their results, but the realistic limitations implied by these are not always clear. In particular, one of the novel elements of the work is that the "optimal ROC" definition can be extended to general sets of classifiers; however, many of the key results require the assumption that the set contains all possible classifiers, which has important implications in practice that should be discussed.

**Strengths And Weaknesses:**

Overall, I think this paper has a solid core with some weaknesses in the presentation and analysis. Some of the weaknesses are easy to address; if they are addressed, I would improve my score.

EDIT: the authors have largely addressed my concerns, so I am changing my overall score from 4 (borderline reject) to 7 (accept).

Strengths:

I am not deeply familiar with related work, but based on the authors' presentation of it, the work contains original, significant contributions, particularly in section 3. The technical portions are thorough and most sections are clear and easy to read. The problem being addressed is general, fundamental, and broadly relevant in machine learning.


Weaknesses:

The main paper is largely consumed by technical details, and as a result, many higher-level parts (motivation, related work/context, experiment results and analysis, discussion) are not thorough. This makes it difficult to determine the significance and potential impact and limitations of the work.

The originality/significance of section 2 is weak. Most of the results are noted to be only slight generalizations of previous work. The key novelty is the notion of "optimal ROC" for general sets of classifiers, and there is hardly any useful analysis in this context compared to the context of all possible classifiers.

There are some terms and notations throughout the text that are not clearly defined before they are used, and for which I couldn't find definitive information when searching online. These hinder understanding of high-level concepts as well as technical details. This should be easy to fix, though. I've listed some in the Questions/Suggestions section.

The experimental results are very limited and not thoroughly discussed. There are only results for two simple synthetic datasets, the results are relegated to the supplementary material, and the discussion of the results is limited. This would be a good place to test on some real datasets, where assumptions will not necessarily hold, and discuss the practicality and limitations of the paper's contributions.

---

> ### Author Response · Authors · 2022-08-02
> **Response to Reviewer hAdd**
>
> Thanks for your valuable comments and suggestions. Following your suggestions,
> we have made some changes to our submission, which we feel greatly improved the presentation.
> Main changes include:
> - We added some high-level discussions on the practical issues of using three common ROC generating procedures in the Appendix. We will include them in the main file if our paper gets accepted.
> - Following your suggestions, we moved some theoretical results in Section 2 and Section 3 to the Appendix.
> - We carefully proofread our paper, and added missing descriptions to some of the notations and terms as you pointed out in your report.
> We also included more detailed explanations in the Questions/Suggestions section of our rebuttal.
> - We added a real data analysis example in Section 4. We also added some further discussions on the implications of
> our simulations studies and real data examples in the article.
>
> For your questions and suggestions,
> we summarize our response and changes to the main file as follows.
>
> ### Questions
> The basic assumption we use for data distribution is that the marginal distribution $X$, denoted as $\mathbb P_X$, is continuous.
> This is used for all our results in Neyman-Pearson classification (section 3.1)
> and Weighted method (section 3.2).
> The reason why we impose assumption that
> $\eta(X)$ is a continuous random variable in Theorem 4 and Theorem 5 of section 3.3 is that,
> the cutoff method will require stronger assumption on the distribution of $X$ in order to target the optimal ROC curve.
> This stronger assumption also reveals that cutoff method is a less favorable method in general applications.
> To make it more clear,
> we have revised this part to avoid confusion.
>
> To see why our continuity assumption on the marginal distribution $X$ is weaker than
> the assumption that $\eta(X)$ is continuous in literature,
> we could look at the case that the conditional distribution of $X$ given $Y=1$ has
> different support with the conditional distribution of $X$ given $Y=-1$,
> i.e. there exists region $D$ such that $f(x|Y=1)/f(x|Y=-1) = 0$ or $\infty$.
> In this case, $\eta(X)$ is no longer continuous but the distribution
> of $X$ can still be continuous.
> In practice, the situation that these two conditional
> distributions of $X$ have different supports could happen.
> In view of this, our results under weaker assumption are more general and applicable to
> a broader class of problems.
>
> ### Suggestions
> Thanks to your valuable suggestions on improving our paper. Following your suggestions,
> we have made the following changes to our paper:
> 1. We have added a real data example in the main paper. Due to space limit, we put some figures in the Appendix.
>     We will include them in the main text if the paper is accepted for publication.
> 2. We have added more discussions on our experimental results, practical recommedations (in the Appendix) and future work (in the Discussion section).
> 3. We have rewritten parts of the introduction with more clear motivations.
>
> We have also added definitions or descriptions for some of the terms and symbols you pointed out
> that are undefined. These include
> - "Model mis-specification'' means that the optimal classifier is not included in the set of classifiers used for a method.
>     For example, if the optimal classifier has a quadratic decision boundary, then all classifiers with a linear decision boundary is considered to be mis-specified.  (line 188)
> - "at the population level'' refers to the situation when the number of samples goes to infinity. (line 34)
> - $\mathbb I(\cdot)$ denotes the indicator function. (line 157)
> - $p^+=\mathbb P(Y=1)$ and $p^-=\mathbb P(Y=-1)$ are probabilties of observing positive and negative class labels, respectively. (line 108)
> - $\text{cl}(A)$ denotes the closure of set $A$. (line 215)
>
> Moreover, we also included a scatterplot showing the $2$-D
> data and corresponding labels for our simulations in the Appendix.
>
> ### Limitations
> We have added some high-level discussions on the practical issues of using the three common
> method for estimating ROC curves in the Appendix.
> These discussions are not only supported by our results under the
> assumption that the model space includes all possible classifiers
> (e.g. Corollary 1, Corollary S2 and Theorem 4),
> but also based on our results under the assumption that the model space is general and
> may be mis-specified (e.g. Proposition 1, Theorem S1 and Theorem S2).

---

> > ### Comment · Reviewer_hAdd · 2022-08-03
> > **Reply to Authors**
> >
> > Thanks to the authors for your thorough response. The authors have largely addressed my concerns in their reply and revision, so I am raising my overall score from a 4 (borderline reject) to a 7 (accept).
> >
> > A few remaining things to note:
> >  - If I'm understanding correctly, the satement on line 86 "our definition is quite general and does not make any assumptions on the data generating distribution" is incorrect, and that it should state that the definition makes weaker assumptions than prior work (thanks for clarifying these assumptions vs those made in previous work in your reply).
> > - The experiments on real-world data are not very comprehensive as only one dataset is considered, but they do support the theoretical results. It would be nice to see a few more datasets included in the results for Figure S3.
> > - It's somewhat cumbersome that the experiment results are still in the supplementary material only, but I understand the space limit, and the key takeaways are conveyed in the text of the main paper.

---

> > > ### Author Response · Authors · 2022-08-08
> > > **Further response to Reviewer hAdd**
> > >
> > > Thank you for your further suggestions.
> > > We have made some additional edits to the article
> > > based on your suggestions. In particular,
> > > - We have rewritten the statement on line 86.
> > > - We will expand the real data examples in the main file if our paper is accepted.

---

### Comment · Area_Chair_ux1q · 2022-08-03
**Please start author-reviewer discussion**

Hi authors and reviewers,

The discussion phase has begun. Please read the other reviews and the author's response (if the authors choose to submit one) and start discussing them with the other reviewers, the authors, and myself.

Note that by default, the authors can see the discussions posted by the reviewers (and vice versa). Please use the "Readers" field to adjust the audience of your post if so wished.

*Our goal is to contribute to the discussion to reach a consensus on each paper*. There is only one week for the discussion (until **August 9**). So please do not wait and start the discussion immediately. Thank you very much.

Best,\
The AC

---

### Meta-Review · Area_Chair_ux1q · 2022-08-26

**Recommendation:** Accept
**Confidence:** Certain

**Metareview:**

I have read all the reviews and discussions carefully.

The reviewers all praised the novelty and the significance of the work. The major complaint is that the proofs are too long to be carefully checked. The authors have appropriately addressed some comments from the reviewers. Given the unanimous support, I have decided to recommend the acceptance of the paper.

**Award:**

No

---

### Decision · Program_Chairs · 2022-09-14

Accept